# ORDER-OPTIMAL INSTANCE-DEPENDENT BOUNDS FOR OFFLINE REINFORCEMENT LEARNING WITH PREFERENCE FEEDBACK

## ABSTRACT

We consider offline reinforcement learning (RL) with preference feedback in which the implicit reward is a linear function of an unknown parameter. Given an offline dataset, our objective consists in ascertaining the optimal action for each state, with the ultimate goal of minimizing the *simple regret*. We propose an algorithm, $\underline{\text{RL}}$ with $\underline{\text{L}}$ocally $\underline{\text{O}}$ptimal $\underline{\text{W}}$eights or RL-LOW, which yields a simple regret of $\exp(-\Omega(n/H))$ where $n$ is the number of data samples and $H$ denotes an instance-dependent hardness quantity that depends explicitly on the suboptimality gap of each action. Furthermore, we derive a first-of-its-kind instance-dependent lower bound in offline RL with preference feedback. Interestingly, we observe that the lower and upper bounds on the simple regret match order-wise in the exponent, demonstrating order-wise optimality of RL-LOW. In view of privacy considerations in practical applications, we also extend RL-LOW to the setting of $(\varepsilon, \delta)$-differential privacy and show, somewhat surprisingly, that the hardness parameter $H$ is unchanged in the asymptotic regime as $n$ tends to infinity; this underscores the inherent efficiency of RL-LOW in terms of preserving the privacy of the observed rewards. Given our focus on establishing instance-dependent bounds, our work stands in stark contrast to previous works that focus on establishing worst-case regrets for offline RL with preference feedback.

## 1 INTRODUCTION

Reinforcement Learning (RL) (Sutton and Barto, 2018) has been widely recognized for its capacity to facilitate agents in learning a sequence of optimal actions through iterative interactions with their environments. However, RL encounters significant hurdles in environments that are characterized by uncertainty or lacking explicit reward signals. To address these shortcomings, the concept of RL with human feedback (or RLHF) has emerged as a prominent paradigm. Preference-based RL (PbRL) (Christiano et al., 2017; Chen et al., 2022; Ibarz et al., 2018; Palan et al., 2019) has stood out as one of the most widely used frameworks for RLHF. In this regard, preference-based RL has achieved remarkable performances in practical applications, with particular importance lying in its ability to align large language models (LLMs) with human intent, thereby mitigating the output of toxic and dishonest information (Ouyang et al., 2022; Ziegler et al., 2019; Glaese et al., 2022; Bai et al., 2022; Liu et al., 2023), and improving the quality of applying to the specific tasks (Stiennon et al., 2020; Wu et al., 2021; Nakano et al., 2021).

In this work, we tackle the problem of offline RL with preference feedback, wherein the learning mechanism operates solely on pre-existing (or offline) data without dynamically engaging with the environment. Given the high cost associated with human interaction, offline RL has assumed particular importance in the context of incorporating human feedback. The significance of this offline framework has been justified by many previous prominent works (Shin et al., 2023; Ouyang et al., 2022; Zhu et al., 2023; Kim et al., 2023). For instance, within the learning process of InstructGPT (Ouyang et al., 2022) or the training procedure of Ahmadian et al. (2024), a pivotal procedure involves the training of a reward model utilizing pre-trained LLM feature vectors, coupled with the utilization of pre-collected human preference feedback as the training dataset. Conceptually, this procedure can be construed as treating the current prompt context as a state within a certain Markov Decision Process (MDP), while the responses generated by the LLM serve as actions within

this process. Empirical findings presented by Ouyang et al. (2022) demonstrate the efficacy of this offline framework in effectively aligning human intent with the outputs of LLMs.

However, the literature concerning theoretical analyses within the domain of offline PbRL remains rather scant. Previous theoretical analyses (Zhu et al., 2023; Zhan et al., 2024) of offline PbRL predominantly focused on the worst-case (or minimax) regret, often resulting in the derivation of regret upper bounds for their algorithms of the form $\tilde{O}(n^{-1/2})$, where $n$ is the size of the offline dataset. In this work, we adopt a different approach that is centered on instance-dependent guarantees. In other words, we wish to derive performance guarantees that are functions of the specific problem instance, thus elucidating the role of fundamental hardness parameters. This yields complementary insights to the existing worst-case analyses. To this end, we design and analyze RL-LOW, a preference-based RL algorithm. Our analysis of the performance RL-LOW unveils an instance-dependent simple regret bound of $\exp(-\Omega(n/H))$, where $H$ is a hardness parameter. This reveals that the simple regret decays exponentially fast in the size of the dataset $n$ and the exponential rate of convergence has also been identified. Complementarily, by proving an instance-dependent lower bound, we show that any algorithm will suffer from a simple regret of at least $\exp(-O(n/H))$. Thus, the dependence of the problem on $H$ is fundamental and cannot be improved upon, thereby demonstrating the efficacy of RL-LOW and the tightness of our analyses.

## 1.1 RELATED WORKS

**Preference-based RL:** From the empirical viewpoint, Christiano et al. (2017) initially demonstrated that RL systems can effectively address complex tasks like Atari games and simulated robot locomotion by learning from human preferences between trajectory segments. Later, numerous researchers started to employ human preference feedback to enhance the performance of LLMs, e.g., aligning the LLMs' behavior with human intent (Ouyang et al., 2022; Ziegler et al., 2019; Glaese et al., 2022; Bai et al., 2022; Liu et al., 2023), and enhancing the efficacy of application to specific tasks (Stiennon et al., 2020; Wu et al., 2021; Nakano et al., 2021).

From the theoretical perspective, the existing literature remains sparse in offline RL with preference feedback. Zhu et al. (2023) elucidated the failure of the maximum likelihood estimation (MLE) procedure in some scenarios. Motivated by this, they theoretically prove the (near) minimax optimality of the PESSIMISTIC MLE approach with a high probability guarantee. In addition, Zhan et al. (2024) introduced a novel paradigm for general reward functions, and they introduce $\varepsilon$-bracket approximations for reward models, accompanied by a rigorous theoretical analysis delineating sample complexity in terms of approximation error $\varepsilon$ and the high-probability parameter $\delta$.

We observe that the above theoretical investigations, while invaluable, are not instance-dependent. Typically, the above minimax or worst-case guarantees yield upper bounds in the form of $\tilde{O}(n^{-1/2})$ and do not depend on any problem-specific factors (such as suboptimality gaps). Our research stands out as a pioneering attempt in offering an instance-dependent examination for offline RL with preference feedback, thereby bridging a critical gap in the existing literature.

**Label-Differential Privacy:** In our study, we also consider the notion of *label privacy*, acknowledging that the labels in our offline dataset originate from users, thus highlighting the imperative to protect user privacy. Chaudhuri and Hsu (2011) were among the pioneers in exploring the concept of *label privacy* within the context of supervised learning for binary classification. Their foundational work posits that the sensitive information primarily resides in the labels, while considering the unlabeled attributes as non-sensitive. Later, the concept of label privacy has been investigated across various machine learning paradigms, including but not limited to PAC learning (Beimel et al., 2013) and deep learning frameworks (Ghazi et al., 2021). This broadened examination underscores the significance and relevance of label privacy considerations across diverse areas of machine learning research and applications.

More recently, Chowdhury et al. (2024) investigated the use of label differential privacy to protect the privacy of human labelers in the process of estimating rewards from preference-based feedback. Chowdhury et al. (2024) derive an upper bound for their proposed algorithm on the estimation error. They show that it also decays as $O(n^{-1/2})$ and the implied constant here depends on $(\varepsilon, \delta)$, the parameters that define differential privacy. This bound only applies in the scenario of estimating the reward value and is not applicable if we want to understand how it depends on the simple regret of a specific instance. In our work, we consider the effect of $(\varepsilon, \delta)$-DP on the simple regret.

### 1.2 OUR CONTRIBUTIONS

We summarize our main contributions as follows:

1. We establish the first-of-its-kind instance-dependent lower bound characterized by suboptimality gaps for a given problem instance. Our analysis reveals that this lower bound takes the form $\exp(-O(n/H))$, where $H$ is a hardness parameter that is an explicit function of the suboptimality gaps. This finding furnishes a novel, and possibly generalizable, analytical approach for assessing algorithmic performance within the realm of preference-based RL.

2. We design a simple algorithm RL-LOW based on the novel concept of *locally optimal weights*. Our analysis demonstrates that its expected simple regret matches the aforementioned instance-dependent lower bound (in the exponential decay rate of the simple regret), thus revealing our algorithm's achievement of instance-dependent optimality.

3. We extend RL-LOW to be applicable to the $(\varepsilon, \delta)$-differential privacy with labels by combining the Gaussian mechanism with the aforementioned locally optimal weights. Our analysis demonstrates that, for large datasets, this combination enables our algorithm to achieve differential privacy without weakening the bound on the simple regret, underscoring the superiority of the design and analysis of RL-LOW.

4. As a by-product of our analyses, we show that RL-LOW achieves a worst-case bound of the form $O(n^{-1/2})$. If we translate the high-probability upper bound in Zhu et al. (2023) to the same worst-case setting, we obtain a bound of the form $O(\sqrt{n^{-1}\log n})$. Thus, our work provides a noticeable (albeit small) improvement over the state-of-the-art theoretical result in Zhu et al. (2023).

## 2 PRELIMINARIES AND PROBLEM SETUP

Let $\mathcal{S} = \{1, \ldots, S\}$ denote the state space, and $\mathcal{A} = \{1, \ldots, A\}$ denote the action set. We assume that there is an unknown non-degenerate distribution $\rho = (\rho_1, \ldots, \rho_S)$ over the states, i.e., $\rho_k > 0$ for all $k \in \mathcal{S}$. The $i$-th action of state $k$ is associated with the feature vector $\phi(k, i) \in \mathbb{R}^d$, and its associated (unknown) reward is

$$r_{k,i} = \langle \phi(k, i), \theta \rangle, \tag{1}$$

where $\theta \in \mathbb{R}^d$ is an unknown parameter vector. The collection of all feature vectors is denoted as $\phi = \{\phi(k, i)\}_{k \in \mathcal{S}, i \in \mathcal{A}}$. For all $k \in \mathcal{S}$, we denote the suboptimaliy gap of action $i \in \mathcal{A}$ as $\Delta_{k,i} = \max_{j \in \mathcal{A}} r_{k,j} - r_{k,i}$. Let $(a^{(0)}, a^{(1)}) \in \mathcal{A}^2$ be a pair of comparisons and let $s \in \mathcal{S}$ be a state. Then, we define a stochastic label $\sigma \in \{0, 1\}$, following the Bradley–Terry–Luce (BTL) model as

$$\mathbb{P}(\sigma = 1 \mid a^{(0)}, a^{(1)}, s) = \frac{\exp(r_{s,a^{(1)}})}{\exp(r_{s,a^{(0)}}) + \exp(r_{s,a^{(1)}})}. \tag{2}$$

Given this model, we assume throughout that we have access to an *offline dataset*, which we denote as $\mathcal{D} = \{(s_i, a_i^{(0)}, a_i^{(1)}, \sigma_i)\}_{i=1}^n$. Note that this dataset consists of $n$ tupies of states, pairs of actions for comparison, and stochastic labels. Without loss of generality, we assume that the comparisons are arranged such that $a_i^{(0)} < a_i^{(1)}$ for all $i = 1, \ldots, n$, and $a_i^{(0)} < a_j^{(0)}$ (or $a_i^{(1)} \le a_j^{(1)}$ if $a_i^{(0)} = a_j^{(0)}$) for all $i < j$. For simplicity, we assume that the feature vectors satisfy $\phi(k, i) \neq \phi(k, j)$ for all states $k \in \mathcal{S}$ and all actions $i \neq j$. In addition, we assume that for each state $k$, the best action $i_k^* = \arg\max_{j \in \mathcal{A}} r_{k,j}$ is unique. Broadly speaking, our objective is to use the offline dataset $\mathcal{D}$ to estimate the best action $i_k^*$ for each state $k \in \mathcal{S}$. Following Zhu et al. (2023), we aim to design a (possibly randomised) algorithm $\Pi$ that uses the dataset $\mathcal{D}$ to output a set of actions $\{\hat{i}_k\}_{k \in \mathcal{S}}$ that minimizes the *simple regret*[1], defined as

$$R_n = \mathbb{E}_{k \sim \rho}[r_{k,i_k^*} - r_{k,\hat{i}_k}]. \tag{3}$$

We also consider a generalized version of the regret that is amenable to the MDP setting in Section 6. Let $N \in \mathbb{R}^{S \times A \times A}$ be a tensor that collects the proportions of each comparison in the dataset $\mathcal{D}$. In particular, $N_{k,i,j} := \frac{1}{n} \sum_{\iota=1}^n \mathbb{1}\{s_\iota = k, a_\iota^{(1)} = i, a_\iota^{(2)} = j\}$ is the proportion of the number of times actions $i$ and $j$ have been compared under state $k$. A problem instance, denoted as $v$, is completely

---

[1]The term "simple regret" is referred to as "performance gap" in some existing works (e.g., Zhu et al. (2023)).

characterised by the tuple $(\rho, \mathcal{S}, \mathcal{A}, \phi, N, \theta)$. In the following, we index instance-specific parameters with the instance $v$ to indicate their dependence on $v$; this will be omitted when the instance is clear from the context. In addition, we write $\mathbb{P}_v^\Pi$ (resp. $\mathbb{E}_v^\Pi$) to denote the probability measure (reps. the expectation) induced under algorithm $\Pi$ and under the instance $v$. Given an instance $v$, we assume $nN_{k,i,j} \in \mathbb{N}$ is an integer [2]. Other assumptions are as follows.

**Assumption 2.1.** (Bounded reward) There exists a finite and known constant $L$ such that for any $k \in \mathcal{S}$ and $i \in \mathcal{A}$, it holds that $|\langle \phi(k,i), \theta \rangle| \leq L$.

In previous works (Zhu et al., 2023), the authors assume that the norms of the feature vectors $\phi(k,i)$ and parameter vector $\theta$ are separately bounded. This clearly implies that Assumption 2.1 is satisfied, but Assumption 2.1 is weaker as it is a bound on the rewards.

**Definition 2.2.** (Consistent instance) A problem instance $v = (\rho, \mathcal{S}, \mathcal{A}, \phi, N, \theta)$ is *consistent* if for all $(k,i,j) \in \mathcal{S} \times \mathcal{A}^2$, it holds that $\phi(k,i) - \phi(k,j) \in \mathrm{Span}\{\phi(k',i') - \phi(k',j') : (k',i',j') \in \mathcal{S} \times \mathcal{A}^2 \text{ and } N_{k',i',j'} > 0\}$.

We say an instance $v$ is *inconsistent* if it is not consistent. In the following, we will be only concerned with those instances that are consistent as the following result shows that it is impossible to design a algorithm that achieves vanishing simple regret for inconsistent instances.

**Proposition 2.3.** *(Impossibility result)* For any inconsistent instance $v = (\rho, \mathcal{S}, \mathcal{A}, \phi, N, \theta)$, there exists an instance $v' = (\rho, \mathcal{S}, \mathcal{A}, \phi, N, \theta')$ such that for all algorithms $\Pi$

$$\liminf_{n \to \infty} \left\{ \mathbb{E}_v^\Pi[R_n] + \mathbb{E}_{v'}^\Pi[R_n] \right\} > 0. \tag{4}$$

## 3 THE PROPOSED ALGORITHM: RL-LOW

In this section, we describe our computationally and statistically efficient algorithm for offline RL with preference feedback based on the novel idea of *locally optimal weights* for estimating the relative reward of each pair of actions. This algorithm, called RL-LOW, is simple and is presented formally in Algorithm 1. Before we describe its components, we introduce some notations.

First, we denote $B_{k,i,j}$ as the empirical success rate with the comparison of actions $i$ and $j$, i.e., for $k \in \mathcal{S}$ and $i, j \in \mathcal{A}$ with $N_{k,i,j} > 0$

$$B_{k,i,j} := \frac{1}{nN_{k,i,j}} \sum_{\iota=1}^{n} \sigma_\iota \mathbb{1}\{s_\iota = k, a_\iota^{(1)} = i, a_\iota^{(2)} = j\}, \tag{5}$$

and $B_{k,j,i} := 1 - B_{k,i,j}$. If $N_{k,i,j} = N_{k,j,i} = 0$, we define $B_{k,i,j} = B_{k,j,i} = 0$. Subsequently, certain empirical success rates may exhibit magnitudes that are either excessively large or small. We clip them by means of the following operation:

$$B_{k,i,j}^{\mathrm{CLP}} = \mathrm{CLIP}_L(B_{k,i,j}) \quad \text{where} \quad \mathrm{CLIP}_L(a) = \begin{cases} \frac{\exp(2L)}{1+\exp(2L)} & a > \frac{\exp(2L)}{1+\exp(2L)} \\ \frac{1}{1+\exp(2L)} & a < \frac{1}{1+\exp(2L)} \\ a & \text{otherwise} \end{cases}. \tag{6}$$

In accordance with Assumption 2.1, the implicit rewards are bounded by $L$. Consequently, within our BTL model framework, the success rate of each comparison necessarily falls within the interval $\left[\frac{1}{1+\exp(2L)}, \frac{\exp(2L)}{1+\exp(2L)}\right]$. We exploit this in Eqn. (6) to ensure that the implementation of our clip operation is consistent with the model's dynamics. We are now ready to introduce the notion of *locally optimal* weights, which plays a central role in the estimation of the rewards.

**Definition 3.1.** (Locally Optimal Weight) For an consistent instance $v$, let $\mathcal{U}_{k,i,j} = \{u \in \mathbb{R}^{S \times A \times A} : \phi(k,i) - \phi(k,j) = \sum_{k' \in \mathcal{S}, i', j' \in \mathcal{A}} u_{k',i',j'}(\phi(k',i') - \phi(k',j'))$ and $u_{k',i',j'} = 0$ if $N_{k',i',j'} = 0\}$. We say that $w^{(k,i,j)} = (w_{k',i',j'}^{(k,i,j)})_{k' \in \mathcal{S}, i', j' \in \mathcal{A}}$ is a set of *locally optimal weights* for $(k,i,j) \in \mathcal{S} \times \mathcal{A}^2$ with $i \neq j$ if

$$w^{(k,i,j)} \in \operatorname*{argmin}_{u \in \mathcal{U}_{k,i,j}} \left\{ \sum_{k' \in \mathcal{S}, i', j' \in \mathcal{A}: N_{k',i',j'} > 0} \frac{(u_{k',i',j'})^2}{N_{k',i',j'}} \right\}. \tag{7}$$

---

[2] For brevity in notation, we assume that $nN_{k,i,j} \in \mathbb{N}$ is an integer. To be more precise, the sample count for $(k,i,j)$ should be written as $\lceil nN_{k,i,j} \rceil$.

---

**Algorithm 1** Reinforcement Learning with Locally Optimal Weights (RL-LOW)

---

**Input:** Dataset $\mathcal{D} = \{(s_i, a_i^{(1)}, a_i^{(2)}, \sigma_i)\}_{i=1}^n$ and feature maps $\phi = \{\phi(k, i)\}_{k \in \mathcal{S}, i \in \mathcal{A}}$ .

**Output:** The estimated best action $\hat{i}_k \in \mathcal{A}$ of each state $k \in \mathcal{S}$.

1: Compute the sample proportions $N_{k,i,j} \leftarrow \frac{1}{n} \sum_{\iota=1}^n \mathbb{1}\{s_\iota = k, a_\iota^{(1)} = i, a_\iota^{(2)} = j\}$.

2: For $k \in \mathcal{S}$ and $i, j \in \mathcal{A}$ such that $N_{k,i,j} > 0$, compute the success rate of each pair of comparisons using Eqn. (5).

3: Clip the success rate using the knowledge of $L$ and using Eqn. (6).

4: For each state $k \in \mathcal{S}$ and distinct actions $i, j \in \mathcal{A}$ with $i < j$, compute the locally optimal weights $(w_{k',i',j'}^{(k,i,j)})_{k' \in \mathcal{S}, i', j' \in \mathcal{A}}$ using Eqn. (7).

5: Compute the empirical relative reward $\hat{r}_{k,i,j}$ for each $k \in \mathcal{S}, i, j \in \mathcal{A}$ using Eqn. (8).

6: **return** for any $k \in \mathcal{S}$, let $\hat{i}_k \in \{i \in \mathcal{A} : \hat{r}_{k,i,j} \geq 0, \forall j \neq i\}$; resolve ties uniformly.

---

The weights in Eqn. (7) are described as "locally optimal" because they are customized to each $(k, i, j)$ tuple. Hence, $w^{(k,i,j)}$ is *local* to $(k, i, j)$. This is a novelty in the design of our algorithm.

By the definition of the consistency of an instance (cf. Definition 2.2), there exists a subset $\beta \subset \mathcal{S} \times \mathcal{A}^2$ such that $\phi(k, i) - \phi(k, j) \in \text{Span}\{\phi(k', i') - \phi(k', j') : (k', i', j') \in \beta\}$. Hence, there exists a locally optimal weight for every pair of actions given a consistent instance. In addition, $w^{(k,i,j)}$ can be calculated efficiently by its analytic form (see details in Appendix E).

Equipped with the definition of locally-optimal weights, we now provide an estimate of the relative reward for state $k \in \mathcal{S}$ and pair of action $(i, j) \in \mathcal{A}^2$ with $i \neq j$ as follows:

$$\hat{r}_{k,i,j} = \sum_{k' \in \mathcal{S}, i' \in \mathcal{A}, j' \in \mathcal{A}} w_{k',i',j'}^{(k,i,j)} \log\left(\frac{B_{k',i',j'}^{\text{CLP}}}{1 - B_{k',i',j'}^{\text{CLP}}}\right), \tag{8}$$

and we define $\hat{r}_{k,i,i} = 0$ for all $k \in \mathcal{S}$ and $i \in \mathcal{A}$. The term $\frac{(u_{k',i',j'})^2}{N_{k',i',j'}}$ in Eqn. (7) is a proxy for the variance introduced by the pair of actions $(i', j')$ in state $k'$ when associated with the coefficient $u_{k',i',j'}$ in the linear combination of the definition of $\mathcal{U}_{k,i,j}$. Our objective is to minimize the cumulative variance proxy for $(k, i, j)$, thus enhancing the precision of the estimate of the relative reward for $(k, i, j)$ for the purposes of establishing the tightest possible concentration result for subGaussian random variables (see Appendix G).

Finally, for any $k \in \mathcal{S}$, let $\hat{i}_k \in \hat{\mathcal{I}}_k := \{i \in \mathcal{A} : \hat{r}_{k,i,j} \geq 0, \forall j \neq i\}$ be any estimate of the best action under state $k$. It is natural to wonder whether $\hat{i}_k$ exists, i.e., whether the set $\hat{\mathcal{I}}_k$ is empty. The following proposition answers this in the affirmative.

**Proposition 3.2.** *For any consistent instance $v$ and using estimate of the best action $\hat{i}_k$ under each state $k$ as prescribed by* RL-LOW*, we have $|\hat{\mathcal{I}}_k| \geq 1$ and*

$$\underset{i \in \mathcal{A}}{\arg\max} \, \hat{r}_{k,i,j_1} = \underset{i \in \mathcal{A}}{\arg\max} \, \hat{r}_{k,i,j_2} = \hat{\mathcal{I}}_k \quad \text{for any} \quad j_1, j_2 \in \mathcal{A}. \tag{9}$$

**Discussion of Computational Complexity:** Proposition 3.2 obviates the need to compute all values of $\hat{r}_{k,i,j}$ for each $(k, i, j) \in \mathcal{S} \times \mathcal{A}^2$. We demonstrate that the RL-LOW algorithm can be efficiently implemented with a computational complexity of $\mathcal{O}(SAd + nd^2 + d^3)$, as the term $SAd$ corresponds to the natural process of scanning the feature vectors for all state-action pairs. The terms $nd^2 + d^3$ are typical in scenarios involving a linear reward structure, such as in linear regression. It is worth noting that the term $SAd$ can be removed if we do not need to output $\hat{i}_k$ for each $k \in \mathcal{S}$, but rather a parametric function $\hat{i}(k; \vartheta)$ is to be learned; see details in Appendix E.2.1.

### 3.1 UPPER BOUND OF RL-LOW

In this section, we provide an instance-dependent upper bound of the simple regret for Rl-LOW. In addition, we also provide a worst-case upper bound as a by-product. First, we define an instance-dependent hardness parameter $H(v)$. Let

$$H(v) := \max_{k \in \mathcal{S}, i \in \mathcal{A}: i \neq i_k^*} \frac{\gamma_{k,i}}{\Delta_{k,i}^2} \quad \text{where} \quad \gamma_{k,i} := \sum_{k' \in \mathcal{S}, i', j' \in \mathcal{A}: N_{k',i',j'} > 0} \frac{(w_{k',i',j'}^{(k,i,i_k^*)})^2}{N_{k',i',j'}}. \tag{10}$$

The parameter $\gamma_{k,i}$ exhibits a positive correlation with the variance proxy of the relative empirical reward $\hat{r}_{k,i,i_k^*}$. Consequently, the ratio $\frac{\gamma_{k,i}}{\Delta_{k,i}^2}$ in the definition of $H(v)$ serves as a quantitative measure of the difficulty that the empirical reward of a suboptimal action $i$ surpasses that of $i_k^*$ in state $k$; see more intuitive explanations in Appendix B.

**Theorem 3.3.** *(Instance-Dependent Upper Bound) For any consistent instance $v$, under* RL-LOW, *we have for all sufficiently large $n$,*

$$\mathbb{E}_v^{\text{RL-LOW}}[R_n] \leq \exp\left(-\frac{n}{C_{\text{up}} \cdot H(v)}\right), \tag{11}$$

*where $C_{\text{up}}$ is a universal constant.*[3]

From Theorem 3.3, it is evident that the upper bound decays exponentially fast and the exponent is a function of an instance-dependent hardness term $H(v)$. This is the first instance-dependent analysis in offline reinforcement learning with preference feedback.

It is natural to wonder why we do not devise an instance-dependent analysis of or modification to PESSIMISTIC MLE which was developed by Zhu et al. (2023). Note that PESSIMISTIC MLE is designed to perform well *with high probability* and not necessarily *in expectation*. In particular, the regret bound of PESSIMISTIC MLE holds with probability at least $1 - \delta$. Hence, to ensure the regret is less than $\exp(-\Omega(n/H(v)))$, one should set the failure probability $\delta$ to be $\exp(-\Theta(n/H(v)))$, which is not possible as $H(v)$ is unknown to the algorithm (since $\theta$ is also unknown).

We further provide a worst-case upper bound for RL-LOW as follows.

**Proposition 3.4.** *(Worst-Case Upper Bound) For any consistent instance $v$ and for all $n \geq 1$,*

$$\mathbb{E}_v^{\text{RL-LOW}}[R_n] \leq \frac{\sum\limits_{k,i:i \neq i_k^*} \rho_k(\sqrt{\gamma_{k,i}} + \tilde{\gamma}_{k,i})}{C_{\text{wup}}\sqrt{n}} \quad \text{where } \tilde{\gamma}_{k,i} = \sum\limits_{k',i',j':N_{k',i',j'}>0} \frac{|w_{k',i',j'}^{(k,i,i_k^*)}|}{\sqrt{N_{k',i',j'}}} \tag{12}$$

*and $C_{\text{wup}} > 0$ is a universal constant.*

We note that in Zhu et al. (2023), the high probability upper bound is of the form $O\left(\sqrt{n^{-1}\log(1/\delta)}\right)$ for the dependency of $n$ and $\delta$. Hence, if we desire a bound in expectation, we obtain, through the law of total probability, a bound of the form $\mathbb{E}[R_n] = O\left(\sqrt{n^{-1}\log(1/\delta)} + \delta\right)$. Minimizing this bound over $\delta$ yields $\mathbb{E}[R_n] = O\left(\sqrt{n^{-1}\log n}\right)$. In terms of the dependence on $n$, it exhibits a performance that is slightly inferior to our established upper bound of $O(n^{-1/2})$.

## 4 INSTANCE-DEPENDENT LOWER BOUND

In this section, we derive the first-of-its-kind instance-dependent lower bound on offline RL with preference feedback. Before we present our bound, we present some auxiliary lemmas that are potentially instrumental in deriving lower bounds on other preference-based RL problems.

Given any instance $v$, we let $P_v^{(n)}$ denote the joint distribution of the associated labels $\{\sigma_i\}_{i=1}^n$. The following lemma provides an estimate of the Kullback–Leibler (KL) divergence between instances $v$ and $v'$ that share the same parameters except for the latent vector $\theta$ that defines the reward in (1).

**Lemma 4.1.** *For any instance $v = (\rho, \mathcal{S}, \mathcal{A}, \phi, N, \theta)$ and $v' = (\rho, \mathcal{S}, \mathcal{A}, \phi, N, \theta')$, it holds that*

$$2n \exp(-4R_{\max}) \leq \frac{D_{\text{KL}}(P_v^{(n)} \| P_{v'}^{(n)})}{\tilde{D}(v, v')} \leq 2n \exp(2R_{\max}) \tag{13}$$

*where*

$$\tilde{D}(v, v') = \sum_{k \in \mathcal{S}, i,j \in \mathcal{A}} N_{k,i,j}(\langle \phi(k,i) - \phi(k,j), \theta - \theta' \rangle)^2, \tag{14}$$

*and where $R_{\max} = \max_{k \in \mathcal{S}, i \in \mathcal{A}} \max\{|\langle \phi(k,i), \theta \rangle|, |\langle \phi(k,i), \theta' \rangle|\}$ is the maximum absolute reward in these two instances.*

---

[3]In this paper, our universal constants depend on $L$, which is known and fixed throughout.

This lemma demonstrates that when the rewards are bounded, the weighted sum of squared differences of the relative rewards can be used to approximate the KL divergence between the distributions of two instances. The approximation is precisely $\tilde{D}(v, v')$ defined in (14). Furthermore, for any $\mathbf{z} \in \mathbb{R}^d$, $\eta \in \mathbb{R}$ and consistent instance $v = (\rho, \mathcal{S}, \mathcal{A}, \phi, N, \theta)$, we let $\mathrm{Alt}(v, \mathbf{z}, \eta)$ be the set of instances that share the same instance parameters except for $\theta$ and satisfies $\langle \mathbf{z}, \theta' - \theta \rangle = \eta$ for all $v' = (\rho, \mathcal{S}, \mathcal{A}, \phi, N, \theta') \in \mathrm{Alt}(v, \mathbf{z}, \eta)$. The following lemma states a useful property that relates the Alt set to the "approximate KL divergence" $\tilde{D}$.

**Lemma 4.2.** *Let $\mathcal{G}$ be an arbitrary orthonormal basis of* $\mathrm{Span}\{\phi(k', i') - \phi(k', j') : (k', i', j') \in \mathcal{S} \times \mathcal{A}^2 \text{ and } N_{k',i',j'} > 0\}$. *Also let $[\mathbf{w}]_\mathcal{G}$ denote the column vector that represents $\mathbf{w}$ under basis $\mathcal{G}$ (Meyer, 2000, Chapter 4). Define the matrix*

$$V := \sum_{k \in \mathcal{S}, i,j \in \mathcal{A}} N_{k,i,j} [\phi(k,i) - \phi(k,j)]_\mathcal{G} [\phi(k,i) - \phi(k,j)]_\mathcal{G}^\top. \tag{15}$$

*Then for any consistent instance $v = (\rho, \mathcal{S}, \mathcal{A}, \phi, N, \theta)$, $\eta \in \mathbb{R}$, and $\mathbf{z} \in \mathrm{Span}\{\phi(k', i') - \phi(k', j') : (k', i', j') \in \mathcal{S} \times \mathcal{A}^2 \text{ and } N_{k',i',j'} > 0\}$,*

$$\min_{v' \in \mathrm{Alt}(v, \mathbf{z}, \eta)} \tilde{D}(v, v') = \frac{\eta^2}{\|[\mathbf{z}]_\mathcal{G}\|_{V^{-1}}^2}. \tag{16}$$

Lemma 4.2 provides an estimate of the KL divergence between instance $v$ and $v' \in \mathrm{Alt}(v, \mathbf{z}, \eta)$. This, in turn, provides a convenient means to apply the ubiquitous *change of measure* technique to derive the lower bound.

In addition, let $(\bar{i}, \bar{k})$ be the state-action pair that attains maximum in the definition of hardness in Eqn. (10). Define the subset of instances

$$\mathcal{Q} = \left\{ v \text{ consistent} : \frac{\gamma_{\bar{k}, \bar{i}}}{\Delta_{\bar{k}, \bar{i}}^2} \geq 4 \frac{\gamma_{k,i}}{\Delta_{k,i}^2} \ \forall (k, i) \neq (\bar{k}, \bar{i}) \text{ and } i \neq i_k^* \right\}. \tag{17}$$

We are now ready to state our lower bound.

**Theorem 4.3.** *(Instance-Dependent Lower Bound) For any instance $v = (\rho, \mathcal{S}, \mathcal{A}, \phi, N, \theta) \in \mathcal{Q}$, there exists another instance $v' = (\rho, \mathcal{S}, \mathcal{A}, \phi, N, \theta')$ with $H(v) \leq H(v') \leq 8H(v)$ such that for all sufficiently large $n$,*

$$\inf_\Pi \left\{ \mathbb{E}_v^\Pi[R_n] + \mathbb{E}_{v'}^\Pi[R_n] \right\} \geq \exp \left( -\frac{n}{C_{\mathrm{lo}} \cdot H(v)} \right),$$

*where $C_{\mathrm{lo}} > 0$ is a universal constant.*

The alternative instance $v'$ that appears in Theorem 4.3 is judiciously chosen to be $v' \in \mathrm{Alt}(v, \phi(\bar{k}(v), \bar{i}(v)) - \phi(\bar{k}(v), i_{\bar{k}}^*(v)), 2\Delta_{\bar{k}(v), \bar{i}(v)}(v))$. In particular, it is designed so that the optimal action $i_{\bar{k}}^*$ under state $k$ of instance $v$ will become suboptimal under instance $v'$, and its suboptimality gap is at least $\Delta_{\bar{k}(v), \bar{i}(v)}(v)$ under $v'$.

Theorem 4.3 is an instance-dependent lower bound for all instances in the set $\mathcal{Q}$. The condition that defines $\mathcal{Q}$ in Eqn. (17) ensures that the hardness quantities $H(v)$ and $H(v')$ have the same order. Since instances in $\mathcal{Q}$ cover all possible hardness values $H(v)$ (i.e., for every hardness values $h > 0$, there exists an instance in $\mathcal{Q}$ of hardness $h$), we conclude that for any (small) $\epsilon \in (0, 1)$, there does not exist any algorithm $\Pi$ that achieves

$$\mathbb{E}_v^\Pi[R_n] = \exp \left( -\Omega \left( \frac{n}{H(v)^{1-\epsilon}} \right) \right) \text{ for all consistent instance } v. \tag{18}$$

In this sense, the exponential decay rate of the simple regret of RL-LOW presented in Theorem 3.3 is asymptotically tight (or optimal) and the exponential dependence on the hardness parameter $H(v)$ is necessary, fundamental, and cannot be improved upon.

## 5 EXTENSION TO $(\varepsilon, \delta)$-DIFFERENTIAL PRIVACY (DP)

In this section, we extend our algorithm RL-LOW to be amenable to $(\varepsilon, \delta)$-differential privacy with labels. To formalize our results, we provide the definition of $(\varepsilon, \delta)$-DP, following Dwork et al. (2014). We say that two sets of preference labels, $\sigma := \{\sigma_i\}_{i=1}^n$ and $\sigma' := \{\sigma_i'\}_{i=1}^n$ are *neighboring* if there exists $s \in [n]$ such that $\sigma_s \neq \sigma_s'$ and $\sigma_j = \sigma_j'$ for all $j \neq s$.

**Definition 5.1.** (Differential Privacy with labels) Fix any label-free dataset $\{(s_i, a_i^{(1)}, a_i^{(2)})\}_{i=1}^n$. A (randomized) algorithm $\mathcal{M} : \{0,1\}^n \to \mathcal{A}^S$ (that takes as inputs a set of labels and outputs a set of actions, one for each state) satisfies $(\varepsilon, \delta)$-DP if for all neighboring labels $\sigma := \{\sigma_i\}_{i=1}^n$ and $\sigma' := \{\sigma_i'\}_{i=1}^n$,

$$\mathbb{P}(\mathcal{M}(\sigma) \in \mathcal{Z}) \le e^\varepsilon \, \mathbb{P}(\mathcal{M}(\sigma') \in \mathcal{Z}) + \delta \qquad \forall \mathcal{Z} \subset \mathcal{A}^S. \tag{19}$$

Note that Definition 5.1 primarily concerns protecting the privacy of users' *labels*. In particular, the DP mechanism guarantees that any alteration in a user's label must not substantially affect the output of our algorithm. Otherwise, there exists a risk that a user's label might be inferred through the algorithm's output. Our definition of differential privacy (DP) aligns with that of Chowdhury et al. (2024).

We now adapt our RL-LOW to $(\varepsilon, \delta)$-DP by using the Gaussian mechanism (Dwork et al., 2014). Firstly, we introduce the private version of the empirical success rate (analogous to $B_{k,i,j}$ in (5)) which we denote as

$$\tilde{B}_{k,i,j} := \frac{1}{nN_{k,i,j}} \sum_{\iota=1}^n \sigma_\iota \mathbb{1}\{s_\iota = k, a_\iota^{(1)} = i, a_\iota^{(2)} = j\} + \tilde{\xi}_{k,i,j} \ \forall k \in \mathcal{S}, (i,j) \in \mathcal{A}^2 \text{ with } N_{k,i,j} > 0,$$

where $\tilde{\xi}_{k,i,j}$ is an independent (across $k$, $i$, and $j$) Gaussian noise with zero mean and variance $\frac{2\log(1.25/\delta)}{(\varepsilon n N_{k,i,j})^2}$, and we let $\tilde{B}_{k,j,i} := 1 - \tilde{B}_{k,i,j}$. If $N_{k,i,j} = N_{k,j,i} = 0$, we define $\tilde{B}_{k,i,j} = \tilde{B}_{k,j,i} = 0$. Again, analogously to the operation in (6), we clip $\tilde{B}_{k,i,j}$ to form

$$\tilde{B}_{k,i,j}^{\mathrm{CLP}} = \mathrm{CLIP}_L(\tilde{B}_{k,i,j}) \tag{20}$$

Similarly to Eqn. (8), the perturbed estimated relative rewards are given as follows

$$\tilde{r}_{k,i,j} = \sum_{k' \in \mathcal{S}, (i',j') \in \mathcal{A}^2} w_{k',i',j'}^{(k,i,j)} \log\left(\frac{\tilde{B}_{k',i',j'}^{\mathrm{CLP}}}{1 - \tilde{B}_{k',i',j'}^{\mathrm{CLP}}}\right), \tag{21}$$

where $w^{(k,i,j)}$ is defined in Definition 3.1. Finally, the empirical best action is $\hat{i}_k \in \tilde{\mathcal{I}}_k := \{i \in \mathcal{A} : \tilde{r}_{k,i,j} \ge 0, \forall j \ne i\}$. A similar argument as Proposition 3.2 shows that $\hat{i}_k$ exists; see details in Appendix E for the details. The algorithm described above is a differentialy private version of the RL-LOW algorithm and hence, it is named DP-RL-LOW.

DP-RL-LOW with the carefully chosen variance of $\xi_{k,i,j}$ fulfils the requirement of $(\varepsilon, \delta)$-DP.

**Proposition 5.2.** *Given privacy parameters $\varepsilon, \delta \in (0,1)$, DP-RL-LOW satisfies $(\varepsilon, \delta)$-DP.*

The proof of Proposition 5.2 follows exactly along the lines of the proof of Dwork et al. (2014, Theorem A.1) and is omitted. We next upper bound the expected simple regret of DP-RL-LOW.

**Theorem 5.3.** *(Instance-Dependent Upper Bound for DP-RL-LOW) Given any consistent instance $v$, for all sufficiently large $n$,*

$$\mathbb{E}_v^{\mathrm{DP\text{-}RL\text{-}LOW}}[R_n] \le \exp\left(-C_{\mathrm{DP}} \cdot \left(\frac{n}{H(v)} \wedge \left(\frac{n}{H_{\mathrm{DP}}^{(\varepsilon,\delta)}(v)}\right)^2\right)\right), \tag{22}$$

*where $C_{\mathrm{DP}} > 0$ is a universal constant, and*

$$H_{\mathrm{DP}}^{(\varepsilon,\delta)}(v) = \max_{k \in \mathcal{S}, i \in \mathcal{A}: i \ne i_k^*} \frac{\sqrt{\log(\frac{1.25}{\delta}) \gamma_{k,i}^{\mathrm{DP}}}}{\sqrt{\varepsilon} \Delta_{k,i}} \quad and \quad \gamma_{k,i}^{\mathrm{DP}} = \sum_{k',i',j' \in \mathcal{A}: N_{k',i',j'} > 0} \left(\frac{w_{k',i',j'}^{(k,i,i_k^*)}}{N_{k',i',j'}}\right)^2. \tag{23}$$

*Consequently,*

$$\limsup_{n \to \infty} \frac{1}{n} \log \mathbb{E}_v^{\mathrm{DP\text{-}RL\text{-}LOW}}[R_n] \le -\frac{C_{\mathrm{DP}}}{H(v)}. \tag{24}$$

The limiting statement in (24) implies that DP-RL-LOW has the same order of the exponential decay rate as its non-differentially privacy counterpart RL-LOW when $n$ is sufficiently large; in particular, $n > (H_{\mathrm{DP}}^{(\varepsilon,\delta)}(v))^2/H(v)$ suffices to nullify the effect of the privacy requirement. In other words, in the sense of the exponent, privacy comes "for free" for sufficiently large offline datasets. We also compute a worst-case upper bound of DP-RL-LOW in Appendix H.2. It is of the form $O(\frac{1}{\sqrt{n}} + \frac{\sqrt{\log(1.25/\delta)}}{\varepsilon n})$, and resembles that in Qiao and Wang (2024) without preference feedback.

## 6 EXTENSION TO THE MDP SETTING

Similar to Zhu et al. (2023, Section 1), our definition of simple regret is based on the *static* state distribution $\rho$ in the previous sections. In this section, we extend our results to the MDP setting when the transition probabilities $P(k'|k,i)$ for $(k',k,i) \in \mathcal{S}^2 \times \mathcal{A}$ are known (Zhu et al., 2023, Section 5). Given the transition probabilities $P(k'|k,i)$ and an MDP policy $\pi$, we let $d^\pi$ denote the state distribution (Sutton and Barto, 2018, Section 9.2) under $\pi$. Without loss of generality, we assume the MDP policies are deterministic, and we denote $\pi(k) \in \mathcal{A}$ to be the output action of $\pi$ under state $k$. Let $\pi^*$ denote the optimal MDP policy that is assumed to be unique, i.e.,

$$\pi^* = \operatorname*{argmax}_\pi \mathbb{E}_{k \sim d^\pi}\big[r_{k,\pi(k)}\big].$$

Then, we define the *simple regret* of any MDP policy $\pi$ as

$$R^{\mathrm{MDP}}(\pi) = \mathbb{E}_{k \sim d^{\pi^*}}\big[r_{k,\pi^*(k)}\big] - \mathbb{E}_{k \sim d^\pi}\big[r_{k,\pi(k)}\big]. \tag{25}$$

We now adapt our RL-LOW to the MDP setting by redefining the output as an MDP policy:

$$\hat{\pi}_{\mathrm{out}} \in \operatorname*{argmax}_\pi \mathbb{E}_{k \sim d^\pi}[\hat{r}_{k,\pi(k),j^\dagger}],$$

where $j^\dagger \in \mathcal{A}$ is arbitrarily fixed (e.g., $j^\dagger = 1$). We simply call this adaptation RL-LOW-MDP. The upper bound on its simple regret is stated as follows.

**Theorem 6.1.** *(Instance-Dependent Upper Bound for* RL-LOW-MDP*) Given any consistent instance $v$, for all sufficiently large $n$,*

$$\mathbb{E}_v^{\mathrm{RL\text{-}LOW\text{-}MDP}}\big[R^{\mathrm{MDP}}(\hat{\pi}_{\mathrm{out}})\big] \leq \exp\left(-\frac{n}{C_{\mathrm{MDP}} \cdot H_{\mathrm{MDP}}(v)}\right) \tag{26}$$

*where $C_{\mathrm{MDP}} > 0$ is a universal constant,*

$$H_{\mathrm{MDP}}(v) := \max_{\pi \neq \pi^*} \frac{\gamma^{\mathrm{MDP}}(\pi)}{(\mathbb{E}_{k \sim d^{\pi^*}}\big[r_{k,\pi^*(k)}\big] - \mathbb{E}_{k \sim d^\pi}\big[r_{k,\pi(k)}\big])^2},$$

*and*

$$\gamma^{\mathrm{MDP}}(\pi) := \max_{k:\pi(k)\neq\pi^*(k)} \sum_{k',i',j':N_{k',i',j'}>0} \frac{(w_{k',i',j'}^{(k,\pi(k),\pi^*(k))})^2}{N_{k',i',j'}}.$$

In the presence of the MDP, $H_{\mathrm{MDP}}(v)$, which is a generalization of $H(v)$ in Eqn. (10), turns out to be the instance-dependence hardness parameter of the problem. The proof of Theorem 6.1 is provided in Appendix G. It is important to observe that there exist MDPs (e.g., $P(k|k,i) = 1$ or $S = 1$) such that Theorem 6.1 particularizes to Theorem 3.3. Moreover, the lower bound in Theorem 4.3 is also applicable to the present more general MDP setting when the transition probability kernel $P(k'|k,i)$ is independent of $(k,i)$ and $k'$ follows the distribution $\rho$. Admittedly, the complexity of the problem increases substantially when the transition probabilities are unknown; this aspect warrants further investigation in future studies. Our findings serve as an initial step in exploring instance-dependent bounds in the context of offline RLHF.

## 7 NUMERICAL SIMULATIONS

In this section, we present some numerical simulations of our algorithm RL-LOW and and its differentially private counterpart DP-RL-LOW. We compare them to the state-of-the-art (non-private algorithm) PESSIMISTIC MLE developed by Zhu et al. (2023). We conduct the experiments on a synthetic dataset. Specifically, we set the number of states $S = 2$, the number of actions $A = 10$, the dimensionality of the data $d = 5$, the unknown parameter vector $\theta = [1,1,1,1,1]^\top$, and the state distribution $\rho = [0.4, 0.6]$. The feature vector of each action is generated as follows: For the $i$-th action of state $k \in \{1,2\}$, we first uniformly generate a $d$-dimentional vector $\phi'(k,i)$ with all non-negative elements and $\|\phi'(k,i)\|_1 = 1$. Then, for each state $k \in \{1,2\}$, we set the feature vector of $i$-th action as $\phi(k,i) = \phi'(k,i) - 0.01(i-1)\theta$. That is, in both state 1 and 2, the best action is the first action, and the suboptimality gap of the $i$-th action is $0.05i$. In addition, for both states $k \in \{1,2\}$ and $i < j$, we set $N_{k,i,j} = \frac{1}{A(A-1)}$, i.e., the proportions of comparisons for this

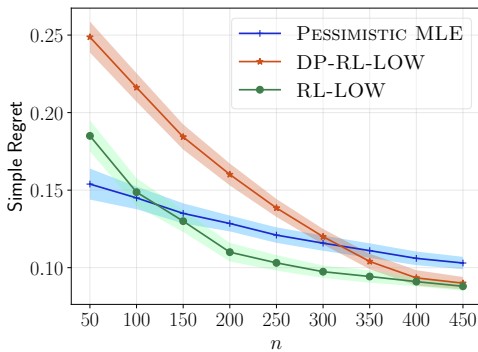 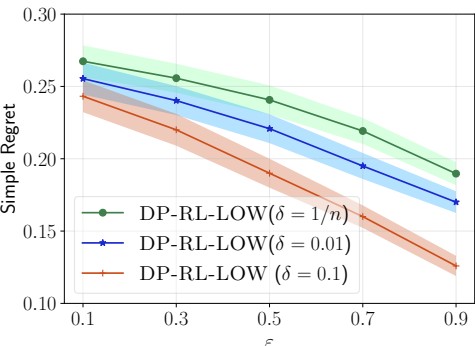

Figure 1: Comparison of RL-LOW and DP-RL-LOW to PESSIMISTIC MLE on average simple regret and standard deviation (shaded area). In the left figure, we set $\delta = 0.2$ and $\varepsilon = 0.9$ for DP-RL-LOW. In the right figure, we set $n = 400$ for all policies.

instance are uniform. In the simulation, we use $\lceil N_{k,i,j} n \rceil$ as the number of samples involved in the comparison between actions $i$ and $j$ under state $k$.

As for the hyperparameters of PESSIMISTIC MLE, we follow the default setting of Zhu et al. (2023, Section 3). In addition, PESSIMISTIC MLE only works under the assumption that $\langle 1, \theta \rangle = 0$ (Zhu et al., 2023, Assumption 2.1). Therefore, when running the experiments of PESSIMISTIC MLE, we further set $d = 6$, $\theta = [1, 1, 1, 1, 1, -5]$ and the 6-th element of each feature vector is set to 0. Then, this new instance is mathematically equivalent to the original instance and additionally satisfies Assumption 2.1 of Zhu et al. (2023) which is needed for PESSIMISTIC MLE.

The simulation results are shown in Figure 1. We run each experiment 200 times, and report the average and standard deviation. From Figure 1 (left), we observe that RL-LOW is inferior to PESSIMISTIC MLE for small $n$. However, since RL-LOW is instance-dependent optimal in the exponential decay rate and in its dependence on the hardness parameter $H(v)$, the experimental findings depicted in Figure 1 (left) corroborate the empirical superiority of our proposed RL-LOW algorithm over PESSIMISTIC MLE for $n$ sufficiently large ($n > 150$ suffices). This observation underscores the efficacy of our novel algorithmic design based on locally optimal weights. Furthermore, from Figure 1 (left), we also observe that as the sample size $n$ increases, the performance of DP-RL-LOW converges to that of RL-LOW, consistent with our theoretical findings in Theorem 5.3.

Lastly, as shown in Figure 1 (right) and the curve of DP-RL-LOW of 1 (left), it is evident that achieving comparable performance between RL-LOW and DP-RL-LOW may necessitate substantially larger sample sizes $n$ when considering small privacy parameters of $\varepsilon$ and $\delta$. This observation is again consistent with our theoretical findings in Theorem 5.3.

## 8 CONCLUDING REMARKS

This paper addresses the problem of offline RL with preference feedback, aiming to determine the optimal action for each state to minimize the simple regret. We introduced a novel algorithm, RL-LOW, which achieves a simple regret of $\exp(-\Omega(n/H(v)))$, where $n$ represents the number of data samples and $H(v)$ characterizes an instance-dependent hardness parameter related to the suboptimality gaps of each action. Additionally, we established a first-of-its-kind instance-dependent lower bound for offline RL with preference feedback, demonstrating the order-wise optimality of RL-LOW (in the exponential decay rate) through the matching of lower and upper bounds on the simple regret. To address privacy concerns, we extended RL-LOW to be amenable to $(\varepsilon, \delta)$-differential privacy, revealing that the hardness parameter $H(v)$ remains unchanged in the asymptotic regime as $n$ tends to infinity. This underscores RL-LOW's effectiveness and robustness in preserving the privacy of observed rewards. Our focus on establishing instance-dependent bounds sets this work apart from previous research that focuses primarily on worst-case regret analyses in offline RL with preference feedback. Some interesting directions for future research include extending our work to incorporate general reward functions (Zhan et al., 2024). In particular, a natural question concerns whether or not there exist an algorithm that is instance-dependent and order-optimal for general reward functions?

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

# Supplementary Material for "Order-Optimal Instance-Dependent Bounds for Offline Reinforcement Learning with Preference Feedback"

## A  ADDITIONAL DETAILS OF INTRODUCTION

### A.1  MORE RELATED WORKS BEYOND PbRL

**Offline RL without preference feedback** The domain of offline RL has been extensively researched over an extended period. Here, We focus on the recent works. Chen and Jiang (2019) revisits and provides theoretical insights into the essential but underexplored assumptions of mild distribution shift and strong representation conditions in value-function approximations, advancing their necessity and applicability. Xie et al. (2021) bridges the gap between online and offline reinforcement learning by introducing the policy finetuning problem, proposing algorithms that leverage a reference policy close to the optimal policy to achieve sample-efficient learning in episodic MDPs. Yin et al. (2022) investigates the statistical limits of offline reinforcement learning using linear models, introducing the variance-aware pessimistic value iteration method to improve learning bounds with offline data. More recently, Wang et al. (2022) enhances the understanding of gap-dependent sample complexity in offline reinforcement learning, demonstrating improved rates under specific policy coverage conditions and providing algorithms nearly matching lower bounds. Similarly, Nguyen-Tang et al. (2023) investigated gap-dependent analysis for offline RL, and they achieved fast rates and zero sub-optimality under specific conditions, and providing both gap-dependent upper and lower bounds for performance with linear function approximation.

Overall, our study identifies a significant oversight in previous research: the absence of preference feedback consideration. Consequently, our work represents the inaugural investigation into instance-dependent bounds within the context of offline reinforcement learning incorporating preference feedback.

**Dueling Bandits** The Dueling Bandits problem was first introduced by Yue and Joachims (2009), sparking a substantial body of subsequent research on the topic. In this section, we highlight some relevant works. Inspired by the classical contextual bandits problem, (Dudík et al., 2015) extend the framework of duel bandits into a contextual setting, and they propose a new concept of von Neumann winner, a game-theoretic solution concept that addresses limitations of the Condorcet winner, along with three efficient algorithms for its online learning and approximation from data. In contrast, (Saha, 2021) explore a distinct aspect of contextual dueling bandits through their proposed Subsetwise-Preference Feedback Model, and the author presents two algorithms for pairwise preferences, achieving near-optimal regret bounds, and extending the analysis to general subsetwise preferences, demonstrating that the fundamental performance limits remain consistent regardless of the subset size. However, this study mainly focuses on the worst-case analysis. More recently, (Di et al., 2024) addressed the contextual dueling bandits with adversarial feedback, proposing a robust algorithm using uncertainty-weighted maximum likelihood estimation. Nonetheless, this work focuses on the adversarial setting, whereas our work examines the stochastic setting.

### A.2  A MOTIVATIONAL EXAMPLE OF LABEL-DP

In the development of question-answering (QA) systems, a common approach for improving response quality involves engaging users in a labeling process where they are asked to provide preference labels. Specifically, users evaluate pairs of system-generated responses to a given question and indicate which response they prefer. This method, often referred to as *pairwise preference labeling*, is integral to training RLHF algorithms that aim to optimize the relevance and utility of answers provided by QA systems.

Given our understanding of the nature of this process, our research emphasizes the importance of protecting the confidentiality of user-submitted preference labels. Without any concerted attempt to protect privacy, these labels, which directly reflect individual opinions or biases toward specific types of responses, can potentially reveal sensitive information, e.g., their preferences for some specific political parties. Therefore, we augment our RL-LOW with a label-DP protection mechanism to mitigate the risk of privacy breaches from the labels..

## B ADDITIONAL EXPLANATION FOR THE HARDNESS PARAMETER $H(v)$

The hardness parameter $H(v)$ is inversely proportional to the square of the suboptimality gaps across various states and actions. Specifically, a smaller suboptimality gap implies an increased hardness parameter, indicating that the instance is more challenging to learn. This relationship underscores the significance of the suboptimality gap as a critical measure in evaluating the complexity and learning difficulty of each instance.

For example, let's suppose $S = 1$, $A = 2$, $r_{1,1} = 0$ and $r_{1,2} = 1$. Given offline data with of size $n$ (i.e., these are $n$ history records for "action 1 wins action 2" or "action 2 wins action 1"), if the learner picks the action with the most winning records (as it does in our algorithm RL-LOW), then by Hoeffding's inequality, it will suffer an upper bound of expected simple regret of $\exp(-C \cdot \frac{n}{(r_{1,2}-r_{1,1})^{-2}})$ for a constant $C$ that does not depend on $r_{1,2} - r_{1,1}$ (in fact this upper bound is also tight in the hardness parameter according to our lower bound). Notice the exponent is $\Theta(\frac{-n}{(r_{1,2}-r_{1,1})^{-2}})$, and the hardness parameter $H(v)$ is exactly $(r_{1,2} - r_{1,1})^{-2}$ under this instance.

## C USEFUL FACTS

In this section, we collate some useful facts that will be used in the subsequent proofs.

**Definition C.1** (SubGaussian norm). A random variable $X$ is *subGassian* if it has a finite subGaussian norm denoted as $\|X\|_{\psi_2}$ and defined as

$$\|X\|_{\psi_2} = \inf \left\{ c > 0 : \mathbb{E}\left[\exp\left(\frac{X^2}{c^2}\right)\right] \leq 2 \right\} < +\infty.$$

**Definition C.2** (Variance proxy). The variance proxy of a subGaussian random variable $X$ is denoted as $\|X\|_{vp}^2$ and defined as

$$\|X\|_{vp}^2 := \inf \left\{ s^2 > 0 : \mathbb{E}\left[\exp((X - \mathbb{E}[X])t)\right] \leq e^{\frac{s^2 t^2}{2}}, \;\; \forall\, t > 0 \right\}.$$

**Lemma C.3** (Linear combination of subGaussian random variables). *Let $X_1, \ldots, X_n$ be independent subGaussian random variables, where the variance proxy of $X_i$ is $\sigma_i^2$. Then, for any $a_1, \ldots, a_n \in \mathbb{R}$, the random variable $\sum_{i=1}^{n} a_i X_i$ is a subGaussian random variable with variance proxy $\sigma^2 = \sum_{i=1}^{n} a_i^2 \sigma_i^2$.*

**Lemma C.4.** *(Tail bound of subGaussian random variables) Suppose $X$ is subGaussian with variance proxy $\sigma^2$. Then, for any $\epsilon > 0$, we have*

$$\Pr(X - \mathbb{E}[X] \geq \epsilon) \leq \exp\left(-\epsilon^2 / \left(2\sigma^2\right)\right),$$

*and*

$$\Pr(X - \mathbb{E}[X] \leq -\epsilon) \leq \exp\left(-\epsilon^2 / \left(2\sigma^2\right)\right),$$

**Lemma C.5.** *(Adapted from (Vershynin, 2018, Proposition 2.5.2)) For any subGaussian random variable $X$,*

$$\|X\|_{vp} \leq C\|X\|_{\psi_2},$$

*where $C \leq 6\sqrt{2e} \cdot (3\sqrt{\log 2} + 1)$. If $\mathbb{E}[X] = 0$, then we have*

$$\|X\|_{\psi_2} \leq \sqrt{6}\|X\|_{vp}.$$

## D PROOF OF PROPOSITION 2.3

**Lemma D.1.** *For any inconsistent instance $v = (\rho, \mathcal{S}, \mathcal{A}, \phi, N, \theta)$, there exists $(k, i) \in \mathcal{S} \times \mathcal{A}$ with $i \neq i_k^*$ such that*

$$\phi(k, i) - \phi(k, i_k^*) \notin \mathrm{Span}\{\phi(k', i') - \phi(k', j') \mid N_{k', i', j'} > 0, (k', i', j') \in \mathcal{S} \times \mathcal{A}^2\}.$$

*Proof.* We prove this result by contradiction. Fix any inconsistent instance $v = (\rho, \mathcal{S}, \mathcal{A}, \phi, N, \theta)$. Assume that for all $(k, i) \in \mathcal{S} \times \mathcal{A}$ with $i \neq i_k^*$, it holds

$$\phi(k, i) - \phi(k, i_k^*) \in \mathrm{Span}\{\phi(k', i') - \phi(k', j') \mid N_{k,i,j} > 0, (k', i', j') \in \mathcal{S} \times \mathcal{A}^2\}. \quad (27)$$

By the fact that $\phi(k,i) - \phi(k,j) = (\phi(k,i) - \phi(k,i_k^*)) - (\phi(k,j) - \phi(k,i_k^*)))$, Eqn. (27) implies that for any $(k,i,j) \in \mathcal{S} \times A \times A$ with $i \neq j$, it holds

$$\phi(k,i) - \phi(k,j) \in \text{Span}\{\phi(k',i') - \phi(k',j') \mid N_{k,i,j} > 0, (k',i',j') \in \mathcal{S} \times \mathcal{A}^2\}, \quad (28)$$

which is a contradiction to that $v$ is inconsistent. This completes the proof of Lemma D.1. $\square$

*Proof of Proposition 2.3.* Fix any inconsistent instance $v = (\rho, \mathcal{S}, \mathcal{A}, \phi, N, \theta)$. By Lemma D.1, there exists $(\underline{k}, \underline{i}) \in \mathcal{S} \times \mathcal{A}$ with $\underline{i} \neq i_{\underline{k}}^*$ such that

$$\phi(\underline{k}, \underline{i}) - \phi(\underline{k}, i_{\underline{k}}^*) \notin \text{Span}\{\phi(k',i') - \phi(k',j') \mid N_{k',i',j'} > 0, (k',i',j') \in \mathcal{S} \times \mathcal{A}^2\}.$$

That is, there exists a vector $\mathbf{z} \in \mathbb{R}^d$ such that

$$\langle \mathbf{z}, \phi(\underline{k}, \underline{i}) - \phi(\underline{k}, i_{\underline{k}}^*) \rangle = -2\langle \theta, \phi(\underline{k}, \underline{i}) - \phi(\underline{k}, i_{\underline{k}}^*) \rangle$$

and for all $(k',i',j') \in \mathcal{S} \times \mathcal{A}^2$ with $N_{k',i',j'} > 0$,

$$\langle \mathbf{z}, \phi(k',i') - \phi(k',j') \rangle = 0.$$

Finally, we let $\theta' = \theta + \mathbf{z}$, and instance $v' = (\rho, \mathcal{S}, \mathcal{A}, \phi, N, \theta')$. By the fact that $\langle \theta, \phi(k',i') - \phi(k',j') \rangle = \langle \theta', \phi(k',i') - \phi(k',j') \rangle$ for all $(k',i',j') \in \mathcal{S} \times \mathcal{A}^2$ with $N_{k',i',j'} > 0$ we get that for all $n \geq 1$

$$D_{\text{KL}}(P_v^{(n)}, P_{v'}^{(n)}) = 0.$$

Therefore, we get that $P_v^{(n)}$ is equal to $P_{v'}^{(n)}$. In addition, by definition of $R_n$, we get that

$$\mathbb{E}_v^\pi[R_n] + \mathbb{E}_{v'}^\pi[R_n]$$

$$= \mathbb{E}_v^\pi \left[ \sum_{k \in \mathcal{S}} \rho_k \left( \max_{j \in \mathcal{A}} \langle \phi(k,j) - \phi(k, \hat{i}_k), \theta \rangle \right) \right] + \mathbb{E}_{v'}^\pi \left[ \sum_{k \in \mathcal{S}} \rho_k \left( \max_{j \in \mathcal{A}} \langle \phi(k,j) - \phi(k, \hat{i}_k), \theta' \rangle \right) \right]$$

$$\overset{(a)}{=} \mathbb{E}_v^\pi \left[ \sum_{k \in \mathcal{S}} \rho_k \left( \max_{j \in \mathcal{A}} \langle \phi(k,j) - \phi(k, \hat{i}_k), \theta \rangle + \max_{j \in \mathcal{A}} \langle \phi(k,j) - \phi(k, \hat{i}_k), \theta' \rangle \right) \right] \quad (29)$$

$$\geq \rho_{\underline{k}} \min_{i \in \mathcal{A}} \left[ \max_{j \in \mathcal{A}} \langle \phi(\underline{k}, j) - \phi(\underline{k}, i), \theta \rangle + \max_{j \in \mathcal{A}} \langle \phi(\underline{k}, j) - \phi(\underline{k}, i), \theta' \rangle \right],$$

where (a) follows from the fact that $P_v^{(n)}$ is equivalent with $P_{v'}^{(n)}$.

Further, by the definition of $v$ and $v'$, we get that

$$\min_{i \in \mathcal{A}} \left[ \max_{j \in \mathcal{A}} \langle \phi(\underline{k}, j) - \phi(\underline{k}, i), \theta \rangle + \max_{j \in \mathcal{A}} \langle \phi(\underline{k}, j) - \phi(\underline{k}, i), \theta' \rangle \right] > 0,$$

and recall that $\rho_{\underline{k}} > 0$. This completes the proof of Proposition 2.3

$\square$

# E  PROOF OF PROPOSITION 3.2 AND MORE DETAILS ON COMPUTATIONAL COMPLEXITY

## E.1  PROOF OF PROPOSITION 3.2

We first provide analytical solutions of $w^{(k,i,j)}$ and $\gamma_{k,i,i_k^*}$.

**Lemma E.1.** *Fix any consistent instance $v = (\rho, \mathcal{S}, \mathcal{A}, \phi, N, \theta)$. Recall the definitions of $w^{(k,i,j)}$ and $\gamma_{k,i,i_k^*}$ in Eqn. (7) and Eqn. (10), respectively. Then, for any $(k,i) \in \mathcal{S} \times \mathcal{A}$ with $i \neq i_k^*$,*

$$\gamma_{k,i,i_k^*} = \|[\phi(k, i_k^* - \phi(k,i)]_{\mathcal{G}}\|_{V^{-1}}^2 \quad (30)$$

*and for any $(k,i,j) \in \mathcal{S} \times \mathcal{A}^2$ with $i \neq j$,*

$$w_{k',i',j'}^{(k,i,j)} = N_{k',i',j'}[\phi(k',i') - \phi(k',j')]_{\mathcal{G}}^\top V^{-1}[\phi(k,i) - \phi(k,j)]_{\mathcal{G}} \quad (31)$$

*where $V$ and $\mathcal{G}$ are as defined in Lemma 4.2.*

*Proof of Lemma E.1.* Fix any $(k, i, j) \in \mathcal{S} \times \mathcal{A}^2$ with $i \neq j$. By definition, the optimization problem of Eqn. (7) is equivalent to

$$\min_{u \in \mathbb{R}^{S \times A \times A}} \sum_{k' \in \mathcal{S}, i', j' \in \mathcal{A} : N_{k',i',j'} > 0} \frac{(u_{k',i',j'})^2}{N_{k',i',j'}} \tag{32}$$

subject to

$$[\phi(k, i) - \phi(k, j)]_{\mathcal{G}} = \sum_{k' \in \mathcal{S}, i', j' \in \mathcal{A} : N_{k',i',j'} > 0} u_{k',i',j'} [\phi(k', i') - \phi(k', j')]_{\mathcal{G}}.$$

The Lagrangian of the above constrained optimization problem is

$$L(u, \lambda) = \sum_{k' \in \mathcal{S}, i', j' \in \mathcal{A} : N_{k',i',j'} > 0} \frac{(u_{k',i',j'})^2}{N_{k',i',j'}} + \lambda^\top \Bigg( [\phi(k, i) - \phi(k, j)]_{\mathcal{G}}$$

$$- \sum_{k' \in \mathcal{S}, i', j' \in \mathcal{A} : N_{k',i',j'} > 0} u_{k',i',j'} [\phi(k', i') - \phi(k', j')]_{\mathcal{G}} \Bigg),$$

for $u \in \mathbb{R}^{S \times A \times A}$ and $\lambda \in \mathbb{R}^{|\mathcal{G}|}$. Then, by solving $\frac{\mathrm{d}L}{\mathrm{d}\lambda} = 0$ and $\frac{\mathrm{d}L}{\mathrm{d}u_{k',i',j'}} = 0$ for all $(k', i', j') \in \mathcal{S} \times \mathcal{A}^2$ with $N_{k',i',j'} > 0$, we obtain that the minimum of (32) is

$$\lambda = -2V^{-1}[\phi(k, i) - \phi(k, j)]_{\mathcal{G}}$$

and

$$u_{k',i',j'} = N_{k',i',j'} [\phi(k', i') - \phi(k', j')]_{\mathcal{G}}^\top V^{-1} [\phi(k, i) - \phi(k, j)]_{\mathcal{G}} \tag{33}$$

for all $(k', i', j') \in \mathcal{S} \times \mathcal{A}^2$ with $N_{k',i',j'} > 0$. That is, for any $(k, i) \in \mathcal{S} \times \mathcal{A}$ with $i \neq i_k^*$,

$$\gamma_{k,i,i_k^*} = \| [\phi(k, i_k^*) - \phi(k, i)]_{\mathcal{G}} \|_{V^{-1}}^2 \tag{34}$$

This completes the desired proof. $\qquad\square$

Then, we are ready to prove Proposition 3.2

*Proof of Propostion 3.2.* By Lemma E.1, we get that under RL-LOW, for any $(k, i, j, j_2) \in \mathcal{S} \times \mathcal{A}^3$,

$$\hat{r}_{k,i,j} + \hat{r}_{k,j,j_2}$$

$$= \sum_{(k',i',j') \in \mathcal{S} \times \mathcal{A}^2} N_{k',i',j'} [\phi(k', i') - \phi(k', j')]_{\mathcal{G}}^\top V^{-1} [\phi(k, i) - \phi(k, j)]_{\mathcal{G}} \log \left( \frac{B_{k',i',j'}^{\mathrm{CLP}}}{1 - B_{k',i',j'}^{\mathrm{CLP}}} \right)$$

$$+ \sum_{(k',i',j') \in \mathcal{S} \times \mathcal{A}^2} N_{k',i',j'} [\phi(k', i') - \phi(k', j')]_{\mathcal{G}}^\top V^{-1} [\phi(k, j) - \phi(k, j_2)]_{\mathcal{G}} \log \left( \frac{B_{k',i',j'}^{\mathrm{CLP}}}{1 - B_{k',i',j'}^{\mathrm{CLP}}} \right)$$

$$= \sum_{(k',i',j') \in \mathcal{S} \times \mathcal{A}^2} N_{k',i',j'} [\phi(k', i') - \phi(k', j')]_{\mathcal{G}}^\top V^{-1} [\phi(k, i) - \phi(k, j_2)]_{\mathcal{G}} \log \left( \frac{B_{k',i',j'}^{\mathrm{CLP}}}{1 - B_{k',i',j'}^{\mathrm{CLP}}} \right)$$

$$= \hat{r}_{k,i,j_2}, \tag{35}$$

which implies that $|\hat{\mathcal{I}}_k| \geq 1$ and

$$\operatorname*{argmax}_{i \in \mathcal{A}} \hat{r}_{k,i,j_1} = \operatorname*{argmax}_{i \in \mathcal{A}} \hat{r}_{k,i,j_2} = \hat{\mathcal{I}}_k \quad \text{for any} \quad j_1, j_2 \in \mathcal{A}. \tag{36}$$

This completes the proof of Proposition 3.2. $\qquad\square$

Following the same lines as the proof of Propostion 3.2, we get the corollary below.

**Corollary E.2.** *For any consistent instance $v$ and using estimate of the best action under each state $k$ as prescribed by* DP-RL-LOW, *we have $|\tilde{\mathcal{I}}_k| \geq 1$ and*

$$\operatorname*{argmax}_{i \in \mathcal{A}} \tilde{r}_{k,i,j_1} = \operatorname*{argmax}_{i \in \mathcal{A}} \tilde{r}_{k,i,j_2} = \tilde{\mathcal{I}}_k \quad \text{for any} \quad j_1, j_2 \in \mathcal{A}. \tag{37}$$

### E.2 COMPUTATIONAL COMPLEXITY

#### E.2.1 COMPUTATIONAL COMPLEXITY OF RL-LOW AND DP-RL-LOW

With Proposition 3.2, it is not necessary to compute all values of $\hat{r}_{k,i,j}$ for each $(k,i,j) \in \mathcal{S} \times \mathcal{A}^2$. Instead, we only need to compute $\hat{r}_{k,i,j^\dagger}$ for each $(k,i) \in \mathcal{S} \times \mathcal{A}$ where $j^\dagger \in \mathcal{A}$ is (arbitrarily) fixed (e.g., $j^\dagger = 1$), and let $\hat{i}_k \in \arg\max_{i \in \mathcal{A}} \hat{r}_{k,i,j^\dagger}$.

Note that by Lemma E.1, we have

$$\hat{r}_{k,i,j^\dagger} = \sum_{(k',i',j') \in \mathcal{S} \times \mathcal{A}^2} N_{k',i',j'}[\phi(k',i') - \phi(k',j')]_\mathcal{G}^\top V^{-1}[\phi(k,i) - \phi(k,j^\dagger)]_\mathcal{G} \log\left(\frac{B_{k',i',j'}^{\mathrm{CLP}}}{1 - B_{k',i',j'}^{\mathrm{CLP}}}\right).$$

Then, if we pre-calculate a *global vector* of $\sum_{(k',i',j') \in \mathcal{S} \times \mathcal{A}^2} N_{k',i',j'}[\phi(k',i') - \phi(k',j')]_\mathcal{G}^\top V^{-1} \log\left(\frac{B_{k',i',j'}^{\mathrm{CLP}}}{1 - B_{k',i',j'}^{\mathrm{CLP}}}\right)$, we can compute each $\hat{r}_{k,i,j^\dagger}$ for $(k,i) \in \mathcal{S} \times \mathcal{A}$ in $\mathcal{O}(d)$ time complexity. Similarly, $\tilde{r}_{k,i,j^\dagger}$ in DP-RL-LOW can be computed through an analogous procedure. Hence, the overall computational complexity of RL-LOW and DP-RL-LOW are $\mathcal{O}(SAd + nd^2 + d^3)$, where the term "$SAd$" corresponds to the natural process of scanning the feature vectors for all state-action pairs, and the terms "$nd^2 + d^3$" corresponds to compute $V^{-1}$ in above.

The computational complexity's dependence on "$SAd$" is inevitable in our current framework, as the generation of the output $\hat{i}_k$ is required for all $k \in \mathcal{S}$. Nonetheless, in an alternative setting in which the set of best actions to be estimated $\{\hat{i}_k \in \mathcal{A}\}_{k \in \mathcal{S}}$ is replaced by a parametric function $\{\hat{i}(k; \vartheta) \in \mathcal{A}\}$ where a parameter $\vartheta$ is to be estimated. In this setting, the previously global vector can be utilized to represent $\vartheta$. Then, in this setting, the overall computational complexity becomes $O(nd^2 + d^3)$, which is required to compute the global vector.

#### E.2.2 COMPUTATIONAL COMPLEXITY OF RL-LOW-MDP

In RL-LOW-MDP, after each $\hat{r}_{k,i,j^\dagger}$ for $(k,i) \in \mathcal{S} \times \mathcal{A}$ is computed, the rest process is the standard RL problem under the condition that the transition and reward functions are known. Hence, the overall computational complexity of RL-LOW-MDP is $\mathcal{O}(SAd + nd^2 + d^3 + g(S, A))$, where the term $g(S, A)$ corresponds to the above standard problem that can be solved by asynchronous dynamic programming or linear programming (Sutton and Barto, 2018). Particularly $g(S, A) = O(\kappa SA)$ by using asynchronous dynamic programming, and $\kappa$ is a hyperparameter that represents the average number of iteration steps, which controls the solution precision.

## F PROOF OF LOWER BOUND

Before we prove the lower bound, we first give a useful corollary. The following corollary is a direct result of Pinsker's inequality.

**Corollary F.1.** *Fix any $C \in (0, \frac{1}{2})$. For any $p, q \in (0,1)$ with $\min(p, 1-p) \geq C$ and $\min(q, 1-q) \geq C$, we have*

$$2(p - q)^2 \leq d_{\mathrm{KL}}(p, q) \leq \frac{2}{C}(p - q)^2$$

*where $d_{\mathrm{KL}}(p, q)$ denotes the KL divergence between the Bernoulli distributions with parameters of $p$ and $q$.*

Then, we prove Lemma 4.1 that reveals the KL divergence between instances. Recall that given any instance $v$, we let $P_v^{(n)}$ denote the distribution of $(\sigma_i)_{i=1}^n$. The following lemma gives an estimation of the KL divergence between instance $v$ and $v'$ that share the same parameters but $\theta$.

**_Lemma 4.1_.** For any instance $v = (\rho, \mathcal{S}, \mathcal{A}, \phi, N, \theta)$ and $v' = (\rho, \mathcal{S}, \mathcal{A}, \phi, N, \theta')$, it holds that

$$2n \exp(-4R_{\max}) \leq \frac{D_{\mathrm{KL}}(P_v^{(n)} \| P_{v'}^{(n)})}{\tilde{D}(v, v')} \leq 2n \exp(2R_{\max}) \tag{38}$$

where

$$\tilde{D}(v, v') = \sum_{k \in \mathcal{S}, i, j \in \mathcal{A}} N_{k,i,j}(\langle \phi(k,i) - \phi(k,j), \theta - \theta' \rangle)^2, \tag{39}$$

and where $R_{\max} = \max_{k \in \mathcal{S}, i \in \mathcal{A}} \max\{|\langle \phi(k,i), \theta \rangle|, |\langle \phi(k,i), \theta' \rangle|\}$ is the maximum absolute reward in these two instances.

*Proof.* Fix any consistent instance $v = (\rho, \mathcal{S}, \mathcal{A}, \phi, N, \theta)$. By the chain rule of the KL divergence, we have

$$D_{\mathrm{KL}}(P_v^{(n)} \| P_{v'}^{(n)})$$

$$= n \sum_{k \in \mathcal{S}, i, j \in \mathcal{A}} N_{k,i,j} d_{\mathrm{KL}} \left( \frac{\exp(\langle \phi(k,i), \theta \rangle)}{\exp(\langle \phi(k,i), \theta \rangle) + \exp(\langle \phi(k,j), \theta \rangle)}, \right.$$

$$\left. \frac{\exp(\langle \phi(k,i), \theta' \rangle)}{\exp(\langle \phi(k,i,\theta') \rangle) + \exp(\langle \phi(k,j), \theta' \rangle)} \right)$$

$$= n \sum_{k \in \mathcal{S}, i, j \in \mathcal{A}} N_{k,i,j} d_{\mathrm{KL}}(\mathrm{Sig}(\langle \phi(k,i) - \phi(k,j), \theta \rangle), \mathrm{Sig}(\langle \phi(k,i) - \phi(k,j), \theta' \rangle)) \tag{40}$$

where $\mathrm{Sig}(\cdot)$ represents the Sigmoid function. By the fact that $-2R_{\max} \le \langle \phi(k,i) - \phi(k,j), \theta \rangle \le 2R_{\max}$ and $-2R_{\max} \le \langle \phi(k,i) - \phi(k,j), \theta' \rangle \le 2R_{\max}$ and that $\frac{\mathrm{d}\mathrm{Sig}(x)}{\mathrm{d}x} = \frac{\exp(-x)}{(\exp(-x)+1)^2}$, we further get that

$$|\mathrm{Sig}(\langle \phi(k,i) - \phi(k,j), \theta \rangle) - \mathrm{Sig}(\langle \phi(k,i) - \phi(k,j), \theta' \rangle)|$$
$$\le |\langle \phi(k,i) - \phi(k,j), \theta - \theta' \rangle| \tag{41}$$

and

$$|\mathrm{Sig}(\langle \phi(k,i) - \phi(k,j), \theta \rangle) - \mathrm{Sig}(\langle \phi(k,i) - \phi(k,j), \theta' \rangle)|$$
$$\ge \exp(-2R_{\max})|\langle \phi(k,i) - \phi(k,j), \theta - \theta' \rangle|. \tag{42}$$

Then, by Corollary F.1, we get that Eqn. (41) and Eqn. (42) imply that

$$d_{\mathrm{KL}}(\mathrm{Sig}(\langle \phi(k,i) - \phi(k,j), \theta \rangle), \mathrm{Sig}(\langle \phi(k,i) - \phi(k,j), \theta' \rangle))$$
$$\le 2\exp(2R_{\max})|\langle \phi(k,i) - \phi(k,j), \theta - \theta' \rangle|^2 \tag{43}$$

and

$$d_{\mathrm{KL}}(\mathrm{Sig}(\langle \phi(k,i) - \phi(k,j), \theta \rangle), \mathrm{Sig}(\langle \phi(k,i) - \phi(k,j), \theta' \rangle))$$
$$\ge 2\exp(-4R_{\max})|\langle \phi(k,i) - \phi(k,j), \theta - \theta' \rangle|^2. \tag{44}$$

Finally, combining Eqn. (40), Eqn. (43) and Eqn. (44), we complete the proof of Lemma 4.1. $\square$

Then, recall that we denote $\tilde{D}(\cdot, \cdot)$ as the approximation of KL divergence between instance $v$ and $v'$ that share the same parameter except $\theta$, i.e., for any $v = (\rho, \mathcal{S}, \mathcal{A}, \phi, N, \theta)$ and $v = (\rho, \mathcal{S}, \mathcal{A}, \phi, N, \theta')$,

$$\tilde{D}(v, v') := \sum_{k \in \mathcal{S}, i, j \in \mathcal{A}} N_{k,i,j}(\langle \phi(k,i) - \phi(k,j), \theta - \theta' \rangle)^2.$$

In addition, recall that for any $\mathbf{z} \in \mathbb{R}^d$, $\eta \in \mathbb{R}$ and consistent instance $v = (\rho, \mathcal{S}, \mathcal{A}, \phi, N, \theta)$, we denote $\mathrm{Alt}(v, \mathbf{z}, \eta)$ as the set of instances that share the same parameter except $\theta$ and satisfy

$$\langle \mathbf{z}, \theta' - \theta \rangle = \eta$$

for all $v' = (\rho, \mathcal{S}, \mathcal{A}, \phi, N, \theta') \in \mathrm{Alt}(v, \mathbf{z}, \eta)$.

We are ready to prove Lemma 4.2 that reveals a useful property for $\mathrm{Alt}(\cdot)$ and $\tilde{D}(\cdot)$

**Lemma 4.2**. Let $\mathcal{G}$ be an arbitrary orthonormal basis of $\text{Span}\{\phi(k',i') - \phi(k',j') : (k',i',j') \in \mathcal{S} \times \mathcal{A}^2 \text{ and } N_{k',i',j'} > 0\}$. Also let $[\mathbf{w}]_{\mathcal{G}}$ denote the column vector that represents $\mathbf{w}$ under basis $\mathcal{G}$ (Meyer, 2000, Chapter 4). Define the matrix

$$V := \sum_{k \in \mathcal{S}, i, j \in \mathcal{A}} N_{k,i,j} [\phi(k,i) - \phi(k,j)]_{\mathcal{G}} [\phi(k,i) - \phi(k,j)]_{\mathcal{G}}^\top.$$

Then for any consistent instance $v = (\rho, \mathcal{S}, \mathcal{A}, \phi, N, \theta)$, $\eta \in \mathbb{R}$, and $\mathbf{z} \in \text{Span}\{\phi(k',i') - \phi(k',j') : (k',i',j') \in \mathcal{S} \times \mathcal{A}^2 \text{ and } N_{k',i',j'} > 0\}$,

$$\min_{v' \in \text{Alt}(v, \mathbf{z}, \eta)} \tilde{D}(v, v') = \frac{\eta^2}{\|[\mathbf{z}]_{\mathcal{G}}\|_{V^{-1}}^2}. \tag{45}$$

and the minimum of (45) is attained in $v' = (\rho, \mathcal{S}, \mathcal{A}, \phi, N, \theta')$ with

$$\theta' = \theta - \frac{\eta}{\|[\mathbf{z}]_{\mathcal{G}}\|_{V^{-1}}^2} V^{-1} \mathbf{z}.$$

*Proof.* By definition, we equivalently write down the optimization problem of Eqn. (45) as follows.

$$\min_{\mathbf{x} \in \mathbb{R}^d} \sum_{k \in \mathcal{S}, i, j \in \mathcal{A}} N_{k,i,j} (\langle [\phi(k,i) - \phi(k,j)]_{\mathcal{G}}, [\mathbf{x}]_{\mathcal{G}} \rangle)^2 \tag{46}$$

subject to

$$\langle \mathbf{x}, \mathbf{z} \rangle = \eta.$$

The Lagrangian of the above constrained optimization problem is,

$$L(\mathbf{x}, \lambda) = \sum_{k \in \mathcal{S}, i, j \in \mathcal{A}} N_{k,i,j} (\langle [\phi(k,i) - \phi(k,j)]_{\mathcal{G}}, [\mathbf{x}]_{\mathcal{G}} \rangle)^2 + \lambda(\langle \mathbf{x}, \mathbf{z} \rangle - \eta).$$

By solving $\frac{\mathrm{d}L}{\mathrm{d}\lambda} = 0$ and $\frac{\mathrm{d}L}{\mathrm{d}\mathbf{x}_i} = 0$ for all $i \in [d]$, we attain the minimum of Eqn. (46) at

$$\begin{cases} \lambda = \frac{2\eta}{\|[\mathbf{z}]_{\mathcal{G}}\|_{V^{-1}}^2} \\ \mathbf{x} = -\frac{\eta}{\|[\mathbf{z}]_{\mathcal{G}}\|_{V^{-1}}^2} V^{-1} \mathbf{z} + \mathbf{g}, \end{cases}$$

where $\mathbf{g}$ is any vector that is orthogonal with vector space $\text{Span}\{\phi(k',i') - \phi(k',j') : (k',i',j') \in \mathcal{S} \times \mathcal{A}^2 \text{ and } N_{k',i',j'} > 0\}$, which implies

$$L(\mathbf{z}, \lambda) = \frac{\eta^2}{\|[\mathbf{z}]_{\mathcal{G}}\|_{V^{-1}}^2}.$$

This completes the desired proof. $\qquad \square$

**Lemma F.2.** *Fix any* $v = (\rho, \mathcal{S}, \mathcal{A}, \phi, N, \theta) \in \mathcal{Q}$. *Let*

$$v' = (\rho, \mathcal{S}, \mathcal{A}, \phi, N, \theta') \in \underset{u \in \text{Alt}(v, \phi(\bar{k}(v), \bar{i}(v)) - \phi(\bar{k}(v), i^*_{\bar{k}(v)}), 2\Delta_{\bar{k}(v), \bar{i}(v)}(v))}{\text{argmin}} \tilde{D}(v, u).$$

*Then, it holds*

$$\begin{cases} \langle \phi(k, i^*_k(v)) - \phi(k, i), \theta - \theta' \rangle = 2\Delta_{k,i}(v) & \textit{for } (k,i) = (\bar{k}(v), \bar{i}(v)) \\ |\langle \phi(k, i^*_k(v)) - \phi(k, i), \theta - \theta' \rangle| \leq \frac{1}{2}\Delta_{k,i}(v) & \forall i \neq i^*_k(v) \textit{ and } (k,i) \neq (\bar{k}(v), \bar{i}(v)) \end{cases} \tag{47}$$

*Consequently,*

$$\begin{cases} i^*_k(v) = i^*_k(v') & \forall k \neq \bar{k}(v) \\ \bar{k}(v) = \bar{k}(v') \\ i^*_{\bar{k}(v')}(v') = \bar{i}(v) \end{cases} \tag{48}$$

*Proof.* Fix any $v$ and $v'$ as in Lemma F.2. By the definition of $v'$, we can immediately obtain the first inequality of Eqn. (47), i.e.,

$$\langle \phi(k, i_k^*(v)) - \phi(k, i), \theta - \theta' \rangle = 2\Delta_{k,i}(v) \text{ for } (k, i) = (\bar{k}(v), \bar{i}(v)). \tag{49}$$

Then, we will use proof by contradiction to prove the second inequality of Eqn. (47).

Assume that there exists $(\tilde{k}, \tilde{i}) \in \mathcal{S} \times \mathcal{A}$ with for all $\tilde{i} \neq i_{\tilde{k}}^*$ and $(\tilde{k}, \tilde{i}) \neq (\bar{k}(v), \bar{i}(v))$ such that

$$|\langle \phi(\tilde{k}, i_{\tilde{k}}^*) - \phi(\tilde{k}, \tilde{i}), \theta - \theta' \rangle| > \frac{\Delta_{\tilde{k}, \tilde{i}}(v)}{2} \tag{50}$$

Then, we let $v'' = (\rho, \mathcal{S}, \mathcal{A}, \phi, N, \theta'')$, where

$$\theta'' = \theta + \frac{(\theta - \theta') 2\Delta_{\tilde{k}, \tilde{i}}(v)}{\langle \phi(\tilde{k}, i_{\tilde{k}}^*) - \phi(\tilde{k}, \tilde{i}), \theta - \theta' \rangle},$$

which implies that $v'' \in \text{Alt}(v, \phi(\tilde{k}, \tilde{i}) - \phi(\tilde{k}, i_{\tilde{k}}^*), 2\Delta_{\tilde{k}, \tilde{i}}(v))$. Note that from Lemma 4.2, we have

$$\tilde{D}(v, v') = \frac{4\Delta_{\bar{k}(v), \bar{i}(v)}(v)^2}{\|[\phi(\bar{k}(v), \bar{i}(v)) - \phi(\bar{k}(v), i_{\bar{k}(v)}^*)]_{\mathcal{G}}\|_{V^{-1}}^2}, \tag{51}$$

where $V$ and $\mathcal{G}$ are as defined in Lemma 4.2. In addition, by definition, we also can get that

$$\begin{aligned} \tilde{D}(v, v'') &= \left( \frac{2\Delta_{\tilde{k}, \tilde{i}}(v)}{\langle \phi(\tilde{k}, i_{\tilde{k}}^*) - \phi(\tilde{k}, \tilde{i}), \theta - \theta' \rangle} \right)^2 \cdot \tilde{D}(v, v') \\ &\overset{(a)}{<} 16\tilde{D}(v, v') \\ &\overset{(b)}{<} \frac{16\Delta_{\bar{k}(v), \bar{i}(v)}(v)^2}{\|[\phi(\bar{k}(v), \bar{i}(v)) - \phi(\bar{k}(v), i_{\bar{k}(v)}^*)]_{\mathcal{G}}\|_{V^{-1}}^2} \\ &\overset{(c)}{<} \frac{64}{\gamma_{\bar{k}(v), \bar{i}(v)}(v)} \end{aligned} \tag{52}$$

where (a) follows from Eqn. (50) and (b) follows from Eqn. (51), and (c) follows from Lemma E.1.

Similarly, from Lemma 4.2, we get that

$$\min_{u \in \text{Alt}(v, \phi(\tilde{k}, \tilde{i}) - \phi(\tilde{k}, i_{\tilde{k}}^*), 2\Delta_{\tilde{k}, \tilde{i}}(v))} \tilde{D}(v, u) = \frac{4\Delta_{\tilde{k}, \tilde{i}}(v)^2}{\|[\phi(\tilde{k}, \tilde{i}) - \phi(\tilde{k}, i_{\tilde{k}}^*)]_{\mathcal{G}}\|_{V^{-1}}^2}$$
$$= \frac{4}{\gamma_{\tilde{k}, \tilde{i}}(v)} \tag{53}$$

By the fact that $v'' \in \text{Alt}(v, \phi(\tilde{k}, \tilde{i}) - \phi(\tilde{k}, i_{\tilde{k}}^*))$ as well as Eqn. (27) and Eqn. (28), we further get that

$$\frac{4}{\gamma_{\tilde{k}, \tilde{i}}(v)} < \frac{64}{\gamma_{\bar{k}(v), \bar{i}(v)}(v)}.$$

That is, $16\gamma_{\tilde{k}, \tilde{i}}(v) > \gamma_{\bar{k}(v), \bar{i}(v)}(v)$, which contradicts the fact that $v \in \mathcal{Q}$.

Hence, for all $(k, i) \in \mathcal{S} \times \mathcal{A}$ with $\forall \tilde{i} \neq i_{\tilde{k}}^*$ and $(k, i) \neq (\bar{k}(v), \bar{i}(v))$, we have

$$|\langle \phi(k, i_k^*(v)) - \phi(\bar{k}(v), \bar{i}(v)), \theta - \theta' \rangle| \geq 2\Delta_{k,i}(v)$$

Consequently, we have

$$\begin{cases} i_k^*(v) = i_k^*(v') & \forall k \neq \bar{k}(v) \\ \bar{k}(v) = \bar{k}(v') \\ i_{\bar{k}(v')}^*(v') = \bar{i}(v) \end{cases}$$

This completes the proof of Lemma F.2. $\qquad\square$

**Lemma F.3.** *Fix any $v = (\rho, \mathcal{S}, \mathcal{A}, \phi, N, \theta) \in \mathcal{Q}$. Let*

$$v' = (\rho, \mathcal{S}, \mathcal{A}, \phi, N, \theta') \in \underset{u \in \mathrm{Alt}(v, \phi(\bar{k}(v), \bar{i}(v)) - \phi(\bar{k}(v), i^*_{\bar{k}(v)}), 2\Delta_{\bar{k}(v), \bar{i}(v)}(v))}{\mathrm{argmin}} \tilde{D}(v, u).$$

*Then, it holds*

$$H(v) \leq H(v') \leq 8H(v).$$

*Proof.* Fix any $v$ and $v'$ as in Lemma F.2. Note that from Lemma F.2, we get that instance $v$ and instance $v'$ share the same optimal action for the state $k \neq \bar{k}$ and the optimal action of state $\bar{k}(v)$ is $\bar{i}(v)$ under the instance $v'$, i.e., $i^*_{\bar{k}(v')}(v') = \bar{i}(v)$ and $i^*_k(v) = i^*_k(v')$ for all $k \neq \bar{k}(v)$. Then, by definition, the hardness of instance $v'$ is

$$H(v') = \max_{k \in \mathcal{S}, i \in \mathcal{A}: i \neq i^*_k(v')} \frac{\gamma_{k,i}(v')}{\Delta^2_{k,i}(v')}$$

$$\overset{(a)}{=} \max \left\{ \max_{k \in \mathcal{S}, i \in \mathcal{A}: i \neq i^*_k(v'), k \neq \bar{k}(v)} \frac{\gamma_{k,i}(v)}{\Delta^2_{k,i}(v')}, \max_{i \in \mathcal{A}: i \neq \bar{i}(v), k = \bar{k}(v)} \frac{\gamma_{k,i}(v')}{\Delta^2_{k,i}(v')} \right\}$$

where (a) follows the fact that $i^*_k(v) = i^*_k(v')$ for all $k \neq \bar{k}(v)$ and both instances $v$ and $v'$ share the same parameters of $\phi$ and $N$, which implies $\gamma_{k,i}(v) = \gamma_{k,i}(v')$ for all $k \neq \bar{k}(v)$.

Then, by the fact that $|\langle \phi(k, i^*_k(v)) - \phi(k, i), \theta - \theta' \rangle| \leq \frac{1}{2}\Delta_{k,i}(v)$ for all $i \neq i^*_k(v)$ and $(k, i) \neq (\bar{k}(v), \bar{i}(v))$ from Lemma F.2, we get that

$$\max_{k \in \mathcal{S}, i \in \mathcal{A}: i \neq i^*_k(v'), k \neq \bar{k}(v)} \frac{\gamma_{k,i}(v)}{\Delta^2_{k,i}(v')} \leq \max_{k \in \mathcal{S}, i \in \mathcal{A}: i \neq i^*_k(v'), k \neq \bar{k}(v)} \frac{4\gamma_{k,i}(v)}{\Delta^2_{k,i}(v)}$$

$$\leq 4H(v) \tag{54}$$

Further, by the fact that $\langle \phi(k, i^*_k(v)) - \phi(k, i), \theta - \theta' \rangle = 2\Delta_{k,i}(v)$ for $(k, i) = (\bar{k}(v), \bar{i}(v))$ Lemma F.2, we get that

$$\max_{i \in \mathcal{A}: i \neq \bar{i}(v), k = \bar{k}(v)} \frac{\gamma_{k,i}(v')}{\Delta^2_{k,i}(v')}$$

$$= \max \left\{ \frac{\gamma_{\bar{k}(v), \bar{i}(v)}(v)}{\Delta^2_{\bar{k}(v), \bar{i}(v)}(v)}, \max_{i \in \mathcal{A}: i \neq \bar{i}(v), i \neq i^*_{\bar{k}(v)}(v)} \frac{\gamma_{\bar{k}(v), i}(v')}{\Delta^2_{\bar{k}(v), i}(v')} \right\}$$

$$= \max \left\{ H(v), \max_{i \in \mathcal{A}: i \neq \bar{i}(v), i \neq i^*_{\bar{k}(v)}(v)} \frac{\gamma_{\bar{k}(v), i}(v')}{\Delta^2_{\bar{k}(v), i}(v')} \right\}$$

$$\overset{(a)}{=} \max \left\{ H(v), \max_{i \in \mathcal{A}: i \neq \bar{i}(v), i \neq i^*_{\bar{k}(v)}(v)} \frac{\|[\phi(\bar{k}(v), \bar{i}(v)) - \phi(\bar{k}(v), i)]_{\mathcal{G}}\|^2_{V^{-1}}}{\Delta^2_{\bar{k}(v), i}(v')} \right\} \tag{55}$$

where (a) follows from Lemma E.1, $V$ and $\mathcal{G}$ are as defined in Lemma 4.2. Similarly, by the fact that $|\langle \phi(k, i^*_k(v)) - \phi(k, i), \theta - \theta' \rangle| \leq \frac{1}{2}\Delta_{k,i}(v)$ $\forall i \neq i^*_k(v)$ and $(k, i) \neq (\bar{k}(v), \bar{i}(v))$, we obtain for $i \in \mathcal{A}$ with $i \neq \bar{i}(v), i \neq i^*_{\bar{k}(v)}(v)$

$$\frac{\|[\phi(\bar{k}(v), \bar{i}(v)) - \phi(\bar{k}(v), i)]_{\mathcal{G}}\|^2_{V^{-1}}}{\Delta^2_{\bar{k}(v), i}(v')} \leq \frac{4\|[\phi(\bar{k}(v), \bar{i}(v)) - \phi(\bar{k}(v), i)]_{\mathcal{G}}\|^2_{V^{-1}}}{(\Delta_{\bar{k}(v), i}(v) \vee \Delta_{\bar{k}(v), \bar{i}}(v))^2}$$

$$= \frac{4\|[\phi(\bar{k}(v), \bar{i}(v)) - \phi(\bar{k}(v), i^*_{\bar{k}(v)}(v)) + \phi(\bar{k}(v), i^*_{\bar{k}(v)}(v)) - \phi(\bar{k}(v), i)]_{\mathcal{G}}\|^2_{V^{-1}}}{(\Delta_{\bar{k}(v), i}(v) \vee \Delta_{\bar{k}(v), \bar{i}}(v))^2}$$

$$\leq \frac{4(\|[\phi(\bar{k}(v), \bar{i}(v)) - \phi(\bar{k}(v), i^*_{\bar{k}(v)}(v))]_{\mathcal{G}}\|^2_{V^{-1}} + \|[\phi(\bar{k}(v), i^*_{\bar{k}(v)}(v)) - \phi(\bar{k}(v), i)]_{\mathcal{G}}\|^2_{V^{-1}}}{(\Delta_{\bar{k}(v), i}(v) \vee \Delta_{\bar{k}(v), \bar{i}}(v))^2}$$

$$\leq 8H(v). \tag{56}$$

Finally, combining Eqn. (54), Eqn. (55) and Eqn. (56), we complete the proof of Lemma F.3. $\qquad\square$

With the ingredients of the above lemmas, we are ready to prove Theorem 4.3.

*Proof of Theorem 4.3.* Fix any $v = (\rho, \mathcal{S}, \mathcal{A}, \phi, N, \theta) \in \mathcal{Q}$ and

$$v' = (\rho, \mathcal{S}, \mathcal{A}, \phi, N, \theta') \in \underset{u \in \mathrm{Alt}(v, \phi(\bar{k}(v), \bar{i}(v)) - \phi(\bar{k}(v), i^*_{\bar{k}(v)}(v)), 2\Delta_{\bar{k}(v), \bar{i}(v)}(v))}{\mathrm{argmin}} \tilde{D}(v, u).$$

By Lemma F.3, we get that $H(v') \leq H(v) \leq 8H(v')$. By Lemma 4.2, we get that

$$\tilde{D}(v, v') = \frac{\|[\phi(\bar{k}(v), i^*_{\bar{k}}(v)) - \phi(\bar{k}(v), \bar{i}(v))]_{\mathcal{G}}\|^2_{V^{-1}}}{4\Delta^2_{\bar{k}(v), \bar{i}(v)}(v)}$$

$$= \frac{1}{4H(v)},$$

where $\mathcal{G}$ and $V$ are as defined in Lemma 4.2. Further, by Lemma 4.1 and Lemma F.2, we get that for any $n > 0$

$$D_{\mathrm{KL}}(P^{(n)}_v \| P^{(n)}_{v'}) \leq 2 \exp(2L) \cdot n \tilde{D}(v, v')$$

$$= \frac{n}{2 \exp(-2L) H(v)}. \tag{57}$$

Then, we let $\bar{\Delta}$ be the minimum suboptimality of state $\bar{k}(v)$ in both instances $v$ and $v'$, i.e.,

$$\bar{\Delta} = \min_{i \in \mathcal{A}} \left[ \max_{j \in \mathcal{A}} \langle \phi(\bar{k}(v), j) - \phi(\bar{k}(v), i), \theta \rangle + \max_{j \in \mathcal{A}} \langle \phi(\bar{k}(v), j) - \phi(\bar{k}(v), i), \theta' \rangle \right]$$

By the definitions of $v$ and $v'$, we can obtain that $\bar{\Delta} > 0$. Then, we get that for any algorithm $\Pi$

$$\mathbb{E}^{\Pi}_v[R_n] + \mathbb{E}^{\Pi}_{v'}[R_n] \geq \rho_{\bar{k}(v)} \bar{\Delta}(1 - D_{\mathrm{TV}}(P^{(n)}_v, P^{(n)}_{v'})) \tag{58}$$

$$\overset{(a)}{\geq} \frac{1}{2} \rho_{\bar{k}(v)} \bar{\Delta} \exp(-D_{\mathrm{KL}}(P^{(n)}_v \| P^{(n)}_{v'}))$$

$$\overset{(b)}{\geq} \frac{1}{2} \rho_{\bar{k}(v)} \bar{\Delta} \exp\left(-\frac{n}{2 \exp(-2L) H(v)}\right),$$

where $D_{\mathrm{TV}}(\cdot, \cdot)$ denotes the total variance distance, and (a) follows from Bretagnolle–Huber inequality (Tsybakov, 2009, Lemma 2.6) and (c) follows from Eqn. (57). Finally, for all sufficiently large $n$, we have

$$\mathbb{E}^{\Pi}_v[R_n] + \mathbb{E}^{\Pi}_{v'}[R_n] \geq \exp\left(-\frac{n}{C_{\mathrm{lo}} \cdot H(v)}\right),$$

which completes the proof of Theorem 4.3.

$\square$

# G  PROOF OF UPPER BOUNDS

**Lemma G.1.** *Let $Y_n$ be a random variable sampled from the Binomial distribution with $n$ trials and probability of success $p \in [1 - \beta, \beta]$ for $\beta \in (\frac{1}{2}, 1)$. Then,*

$$|\mathbb{E}(f(X_n)) - f(\mathbb{E}(X_n))| \leq \frac{3}{(1 - \beta)^4 \sqrt{n}}$$

*where $X_n = \frac{Y_n}{n}$ and*

$$f(x) = \begin{cases} \log(\frac{x}{1-x}) & \text{if } x \in (1 - \beta, \beta) \\ \log(\frac{\beta}{1-\beta}) & \text{if } x \geq \beta \\ \log(\frac{1-\beta}{\beta}) & \text{if } x \leq 1 - \beta. \end{cases} \tag{59}$$

*Proof of Lemma G.1.* For simplicity of notation, we let $x_0 := \mathbb{E}(X_n)$, which implies $x_0 = p \in [1 - \beta, \beta]$. Then, by the smoothness of $f(\cdot)$ on $[1 - \beta, \beta]$, we obtain the equivalent expression of $f(\cdot)$ from the Taylor expansion of $f(\cdot)$ on $[1 - \beta, \beta]$,

$$f(x) = \begin{cases} f(x_0) + f'(x_0) \cdot (x - x_0) + \frac{1}{2} f''(\xi_x) \cdot (x - x_0)^2 & \text{if } x \in [1 - \beta, \beta] \\ f(x_0) + f'(x_0) \cdot (\beta - x_0) + \frac{1}{2} f''(\xi_\beta) \cdot (\beta - x_0)^2 & \text{if } x > \beta \\ f(x_0) + f'(x_0) \cdot (1 - \beta - x_0) + \frac{1}{2} f''(\xi_\beta) \cdot (1 - \beta - x_0)^2 & \text{if } x < 1 - \beta, \end{cases} \tag{60}$$

where $\xi_x \in (\min(x, x_0), \max(x, x_0))$ only depends on $x$ and $x_0$ in the Talor expansion.

By using the fact that $f'(x_0) \cdot (\beta - x_0) = f'(x_0) \cdot (\beta - x) + f'(x_0) \cdot (x - x_0)$ and $f'(x_0) \cdot (1 - \beta - x_0) = f'(x_0) \cdot (1 - \beta - x) + f'(x_0) \cdot (x - x_0)$, we obtain from Eqn. (60) that

$$|\mathbb{E}(f(X_n)) - f(\mathbb{E}(X_n))|$$
$$\leq \sup_{x \in (1-\beta, \beta)} |f''(x)| \cdot \text{Var}(X_n) + |f'(x_0)| \cdot \mathbb{E}([X_n - \beta]_+) + |f'(x_0)| \cdot \mathbb{E}([1 - \beta - X_n]_+) \tag{61}$$

where $\text{Var}(\cdot)$ represents the variance of the random variable $\cdot$ and $[x]_+ = \max\{x, 0\}$.

In addition, we note that

$$\mathbb{E}([X_n - \beta]_+) \leq \int_\beta^1 (x - \beta) \mathbb{P}(X_n \geq x) \, dx$$

$$\overset{(a)}{\leq} \int_\beta^1 (x - \beta) \exp(-2n(x - x_0)^2) \, dx$$

$$\overset{(b)}{\leq} \int_\beta^1 (x - \beta) \exp(-n(x - \beta)^2) \, dx$$

$$\leq \int_\beta^1 \frac{1}{\sqrt{n}} \, dx$$

$$= \frac{1 - \beta}{\sqrt{n}}, \tag{62}$$

where (a) follows from the fact $X_n$ is a subGaussian random variable with variance proxy $\frac{1}{4n}$, and (b) follows from the fact that $(x - \beta) \exp(-n(x - \beta)^2) \leq \frac{1}{\sqrt{n}}$. Similarly, we also can get that

$$\mathbb{E}([1 - \beta - X_n]_+) \leq \frac{1 - \beta}{\sqrt{n}}. \tag{63}$$

Finally,

$$|\mathbb{E}(f(X_n)) - f(\mathbb{E}(X_n))|$$

$$\overset{(a)}{\leq} \sup_{x \in (1-\beta, \beta)} |f''(x)| \cdot \text{Var}(X_n) + |f'(x_0)| \cdot \mathbb{E}([X_n - \beta]_+) + |f'(x_0)| \cdot \mathbb{E}([1 - \beta - X_n]_+)$$

$$\overset{(b)}{\leq} \frac{1}{(1 - \beta)^4} \cdot \text{Var}(X_n) + f'(x_0) \cdot \mathbb{E}([X_n - \beta]_+) + f'(x_0) \cdot \mathbb{E}([1 - \beta - X_n]_+)$$

$$\overset{(c)}{\leq} \frac{1}{(1 - \beta)^4} \cdot \frac{1}{n} + f'(x_0) \cdot \mathbb{E}([X_n - \beta]_+) + f'(x_0) \cdot \mathbb{E}([1 - \beta - X_n]_+)$$

$$\overset{(d)}{\leq} \frac{1}{(1 - \beta)^4} \cdot \frac{1}{n} + \frac{1}{(1 - \beta)^2} \cdot \mathbb{E}([X_n - \beta]_+) + \frac{1}{(1 - \beta)^2} \cdot \mathbb{E}([1 - \beta - X_n]_+)$$

$$\overset{(e)}{\leq} \frac{1}{(1 - \beta)^4} \cdot \frac{1}{n} + \frac{1}{(1 - \beta)^2} \cdot \frac{2}{\sqrt{n}}$$

$$\leq \frac{3}{(1 - \beta)^4 \sqrt{n}}, \tag{64}$$

where (a) follows from Eqn. (61), (b) follows from $\text{Var}(X_n) \leq \frac{1}{n}$, (c) follows from $f''(x) = \frac{2x-1}{(x-1)^2 x^2}$ for $x \in (\beta, 1 - \beta)$, (d) follows from $f'(x) = \frac{1}{x - x^2}$ for $x \in (\beta, 1 - \beta)$, and (e) follows from Eqn. (62) and Eqn. (63). $\qquad \square$

**Lemma G.2.** *Let $Y_n$ be a random variable sampled from the binomial distribution with parameters $n > 0$ and $p \in [1 - \beta, \beta]$ for $\beta \in (\frac{1}{2}, 1)$. Let $X_n = \frac{Y_n}{n}$. Then, $f(X_n)$ is subGaussian with*

$$\|f(X_n)\|_{\mathrm{vp}}^2 \leq \frac{C}{n},$$

*where $C \leq (6\sqrt{2e} \cdot (3\sqrt{\log 2} + 1))^2 \cdot \frac{3}{2(1-\beta)^4}$ and $f(\cdot)$ is defined in Eqn. (59).*

*Proof.* By the definition of $X_n$, we get that

$$\|X_n - p\|_{\mathrm{vp}}^2 \leq \frac{1}{4n}.$$

Then, by Lemma C.5, we get that

$$\|X_n - p\|_{\phi_2}^2 \leq \frac{3}{2n}.$$

By the fact that $|f'(x)| \leq \frac{1}{(1-\beta)^2}$ for $x \in (1 - \beta, \beta)$ and $f'(x) = 0$ for $x \in (0, 1 - \beta) \bigcup (\beta, 1)$ , we get that

$$\|f(X_n) - f(p)\|_{\phi_2}^2 \leq \frac{3}{2n(1-\beta)^4}. \tag{65}$$

Similarly, from Lemma C.5, we further get that

$$\|f(X_n) - f(p)\|_{\mathrm{vp}}^2 \leq (6\sqrt{2e} \cdot (3\sqrt{\log 2} + 1))^2 \cdot \frac{3}{2n(1-\beta)^4}. \tag{66}$$

Note that by definition, we have $\|f(X_n) - f(p)\|_{\mathrm{vp}}^2 = \|f(X_n)\|_{\mathrm{vp}}^2$, which completes the proof of Lemma G.2. $\qquad\square$

With the ingredients of the above lemmas, we are ready to prove the upper bound of our algorithms

*Proof of Theorem 3.3.* Fix any consistent instance $v$ under RL-LOW. By Lemma G.2 and Lemma C.3, we get that $\hat{r}_{k,i_k^*,i}$ is subGaussian with variance proxy as

$$\|\hat{r}_{k,i_k^*,i}\|_{\mathrm{vp}}^2 \leq \frac{C\gamma_{k,i}}{n}, \tag{67}$$

where $C \leq (6\sqrt{2e} \cdot (3\sqrt{\log 2} + 1))^2 \cdot \frac{3}{2(1-\beta)^4}$ and $\beta = \frac{\exp(2L)}{1+\exp(2L)}$.

In addition, by Lemma G.1, we get that for any $(k', i', j') \in \mathcal{S} \times \mathcal{A} \times \mathcal{A}$ with $N_{k',i',j'} > 0$,

$$|\mathbb{E}_v^{\mathrm{RL\text{-}LOW}}[f(B_{k',i',j'})] - f(\mathbb{E}_v^{\mathrm{RL\text{-}LOW}}[B_{k',i',j'}])| \leq \frac{3}{(1-\beta)^4\sqrt{n \cdot N_{k',i',j'}}},$$

which implies for any $(k, i) \in \mathcal{S} \times \mathcal{A}$ with $i \neq i_k^*$,

$$\left| \sum_{k' \in \mathcal{S}, i' \in \mathcal{A}, j' \in \mathcal{A}} w_{k',i',j'}^{(k,i_k^*,i)} \mathbb{E}_v^{\mathrm{RL\text{-}LOW}}[f(B_{k',i',j'})] - f(\mathbb{E}_v^{\mathrm{RL\text{-}LOW}}[\hat{r}_{k,i_k^*,i}]) \right|$$

$$\leq \sum_{k' \in \mathcal{S}, i' \in \mathcal{A}, j' \in \mathcal{A}: N_{k',i',j'} > 0} \frac{3|w_{k',i',j'}^{(k,i_k^*,i)}|}{(1-\beta)^4\sqrt{n \cdot N_{k',i',j'}}}. \tag{68}$$

That is,

$$\left| \mathbb{E}_v^{\mathrm{RL\text{-}LOW}}[\hat{r}_{k,i_k^*,i}] - r_{k,i_k^*,i} \right| \leq \sum_{k' \in \mathcal{S}, i' \in \mathcal{A}, j' \in \mathcal{A}: N_{k',i',j'} > 0} \frac{3|w_{k',i',j'}^{(k,i_k^*,i)}|}{(1-\beta)^4\sqrt{n \cdot N_{k',i',j'}}} \tag{69}$$

$$= \frac{3\tilde{\gamma}_{k,i}}{(1-\beta)^4\sqrt{n}}, \tag{70}$$

where we denote $r_{k,i,j} := r_{k,i} - r_{k,j}$.

Note that by the definition of $R_n$ we have

$$R_n \leq \sum_{(k,i) \in \mathcal{S} \times \mathcal{A}: i \neq i_k^*} \mathbb{1}_{\{\hat{r}_{k,i_k^*,i} \leq 0\}} \cdot \rho_k \Delta_{k,i} \tag{71}$$

$$\leq \sum_{(k,i) \in \mathcal{S} \times \mathcal{A}: i \neq i_k^*} \left[ \left( \mathbb{1}_{\{\hat{r}_{k,i_k^*,i} \leq \mathbb{E}_v^{\text{RL-LOW}}[\hat{r}_{k,i,i_k^*}] - \Delta_{k,i}/2\}} \right. \right.$$

$$\left. \left. \vee \mathbb{1}_{\{|\mathbb{E}_v^{\text{RL-LOW}}[\hat{r}_{k,i_k^*,i}] - \Delta_{k,i}| \geq \Delta_{k,i}/2\}} \right) \cdot \rho_k \Delta_{k,i} \right] \tag{72}$$

By Lemma C.4 and Eqn. (67), we get that for any $(k,i) \in \mathcal{S} \times \mathcal{A}$ with $i \neq i_k^*$,

$$\mathbb{P}_v^{\text{RL-LOW}} \left( \hat{r}_{k,i_k^*,i} \leq \mathbb{E}_v^{\text{RL-LOW}}[\hat{r}_{k,i_k^*,i}] - \Delta_{k,i}/2 \right) \leq \exp\left( -\frac{2n\Delta_{k,i}^2}{C\gamma_{k,i}} \right), \tag{73}$$

and by Eqn. (70) we get that

$$\left| \mathbb{E}_v^{\text{RL-LOW}}[\hat{r}_{k,i_k^*,i}] - r_{k,i_k^*,i} \right| \leq \Delta_{k,i}/2$$

for all $n > \frac{18\tilde{\gamma}_{k,i}^2}{(1-\beta)^8 \Delta_{k,i}}$. That is, for all sufficiently large $n$, we have

$$\mathbb{E}_v^{\text{RL-LOW}}[R_n] \leq \sum_{k \in \mathcal{S}, i \in \mathcal{A}: i \neq i_k^*} \rho_k \Delta_{k,i} \exp\left( -\frac{2n\Delta_{k,i}^2}{C\gamma_{k,i}} \right),$$

which further implies that for all sufficiently large $n$, we have

$$\mathbb{E}_v^{\text{RL-LOW}}[R_n] \leq \exp\left( -\frac{n}{C_{\text{up}} \cdot H(v)} \right).$$

This completes the proof of Theorem 3.3. $\qquad \square$

In addition, we present the proof of Proposition 3.4 as follows.

*Proof of Proposition 3.4.* Note that by the definition of $R_n$ we have

$$R_n \leq \sum_{(k,i) \in \mathcal{S} \times \mathcal{A}: i \neq i_k^*} \mathbb{1}_{\{\hat{r}_{k,i_k^*,i} \leq 0\}} \cdot \rho_k \Delta_{k,i}$$

$$\overset{(a)}{\leq} \sum_{(k,i) \in \mathcal{S} \times \mathcal{A}: i \neq i_k^*} \rho_k \left[ \mathbb{1}_{\{\Delta_{k,i} < \frac{6\tilde{\gamma}_{k,i}}{(1-\beta)^4 \sqrt{n}}\}} \cdot \frac{6\tilde{\gamma}_{k,i}}{(1-\beta)^4 \sqrt{n}} \right.$$

$$\left. + \mathbb{1}_{\{|\mathbb{E}_v^{\text{RL-LOW}}[\hat{r}_{k,i,i_k^*}] - r_{k,i,i_k^*}| \leq \Delta_{k,i}/2\}} \cdot \Delta_{k,i} \right]$$

where $\beta = \frac{\exp(2L)}{1+\exp(2L)}$, and (a) follows from Eqn. (70).

Hence, by Eqn. (73), we have

$$\mathbb{E}_v^{\text{RL-LOW}}[R_n] \leq \sum_{(k,i) \in \mathcal{S} \times \mathcal{A}: i \neq i_k^*} \rho_k \left[ \frac{6\tilde{\gamma}_{k,i}}{(1-\beta)^4 \sqrt{n}} + \exp\left( -\frac{2n\Delta_{k,i}^2}{C\gamma_{k,i}} \right) \Delta_{k,i} \right] \tag{74}$$

$$\leq \sum_{(k,i) \in \mathcal{S} \times \mathcal{A}: i \neq i_k^*} \rho_k \left[ \frac{6\tilde{\gamma}_{k,i}}{(1-\beta)^4 \sqrt{n}} + \sqrt{\frac{C\gamma_{k,i}}{2n}} \right] \tag{75}$$

$$= \frac{1}{\sqrt{n}} \sum_{(k,i) \in \mathcal{S} \times \mathcal{A}: i \neq i_k^*} \rho_k \left[ \frac{6\tilde{\gamma}_{k,i}}{(1-\beta)^4} + \sqrt{\frac{C\gamma_{k,i}}{2}} \right] \tag{76}$$

where $C \leq (6\sqrt{2e} \cdot (3\sqrt{\log 2} + 1))^2 \cdot \frac{3}{2(1-\beta)^4}$.

This completes the desired proof. $\qquad \square$

*Proof of Theorem 6.1.* Fix any consistent instance $v$ under RL-LOW-MDP. By Lemma G.2 and Lemma C.3, we get that $\hat{r}_{k,i,j}$ is subGaussian with variance proxy as

$$\|\hat{r}_{k,i,j}\|_{\text{vp}}^2 \le \frac{C\gamma_{k,i,j}}{n}, \tag{77}$$

where $C \le (6\sqrt{2e} \cdot (3\sqrt{\log 2} + 1))^2 \cdot \frac{3}{2(1-\beta)^4}$ and $\beta = \frac{\exp(2L)}{1+\exp(2L)}$, and $\gamma_{k,i,j}$ is defined as

$$\gamma_{k,i,j} := \sum_{k' \in \mathcal{S}, i',j' \in \mathcal{A}: N_{k',i',j'} > 0} \frac{(w_{k',i',j'}^{(k,i,j)})^2}{N_{k',i',j'}}$$

In addition, by Lemma G.1, we get that for any $(k', i', j') \in \mathcal{S} \times \mathcal{A} \times \mathcal{A}$ with $N_{k',i',j'} > 0$,

$$|\mathbb{E}_v^{\text{RL-LOW-MDP}}[f(B_{k',i',j'})] - f(\mathbb{E}_v^{\text{RL-LOW-MDP}}[B_{k',i',j'}])| \le \frac{3}{(1-\beta)^4\sqrt{n \cdot N_{k',i',j'}}},$$

which implies for any $(k, i, j) \in \mathcal{S} \times \mathcal{A}^2$ with $i \ne j$,

$$\left| \sum_{k' \in \mathcal{S}, i' \in \mathcal{A}, j' \in \mathcal{A}} w_{k',i',j'}^{(k,i,j)} \mathbb{E}_v^{\text{RL-LOW-MDP}}[f(B_{k',i',j'})] - f(\mathbb{E}_v^{\text{RL-LOW-MDP}}[\hat{r}_{k,i,j}]) \right|$$

$$\le \sum_{k' \in \mathcal{S}, i' \in \mathcal{A}, j' \in \mathcal{A}: N_{k',i',j'} > 0} \frac{3|w_{k',i',j'}^{(k,i,j)}|}{(1-\beta)^4\sqrt{n \cdot N_{k',i',j'}}}. \tag{78}$$

That is,

$$\left| \mathbb{E}_v^{\text{RL-LOW-MDP}}[\hat{r}_{k,i,j}] - r_{k,i,j} \right| \le \sum_{k' \in \mathcal{S}, i' \in \mathcal{A}, j' \in \mathcal{A}: N_{k',i',j'} > 0} \frac{3|w_{k',i',j'}^{(k,i,j)}|}{(1-\beta)^4\sqrt{n \cdot N_{k',i',j'}}} \tag{79}$$

$$= \frac{3\tilde{\gamma}_{k,i,j}}{(1-\beta)^4\sqrt{n}}, \tag{80}$$

where we denote

$$\tilde{\gamma}_{k,i,j} := \sum_{k' \in \mathcal{S}, i' \in \mathcal{A}, j' \in \mathcal{A}: N_{k',i',j'} > 0} \frac{|w_{k',i',j'}^{(k,i,j)}|}{\sqrt{N_{k',i',j'}}}$$

and recall that $r_{k,i,j} := r_{k,i} - r_{k,j}$. In addition, we denote $r(\pi) := \mathbb{E}_{k \sim d^\pi}[r_{k,\pi(k)}]$ for any MDP policy $\pi$.

Note that by the definition of $\hat{\pi}_{\text{out}}$ under RL-LOW-MDP, we have

$$R^{\text{MDP}}(\hat{\pi}_{\text{out}}) \le \sum_{\pi \ne \hat{\pi}_{\text{out}}} \mathbb{1}_{\{r(\hat{\pi}_{\text{out}}) < r(\pi)\}} \cdot R^{\text{MDP}}(\pi)$$

$$\le \sum_{\pi \ne \pi^*} R^{\text{MDP}}(\pi) \cdot \mathbb{1}_{\{\bigcup_{k \in \mathcal{S}}\{\hat{r}_{k,\pi(k),\pi^*(k))} - r_{k,\pi(k),\pi^*(k)} \ge R^{\text{MDP}}(\pi)\}\}} \tag{81}$$

By Lemma C.4 and Eqn. (77), we get that for any $\pi \ne \pi^*$ and $k \in \mathcal{S}$ with $\pi(k) \ne \pi^*(k)$,

$$\mathbb{P}_v^{\text{RL-LOW-MDP}}\left( \hat{r}_{k,\pi(k),\pi^*(k)} \ge \mathbb{E}_v^{\text{RL-LOW-MDP}}[\hat{r}_{k,\pi(k),\pi^*(k)}] + R^{\text{MDP}}(\pi)/2 \right)$$

$$\le \exp\left( -\frac{2n(R^{\text{MDP}}(\pi))^2}{C\gamma_{k,\pi(k),\pi^*(k)}} \right), \tag{82}$$

and by Eqn. (80) we get that

$$\left| \mathbb{E}_v^{\text{RL-LOW-MDP}}[\hat{r}_{k,i,j}] - r_{k,i,j} \right| \le R^{\text{MDP}}(\pi)/2$$

for all $n > \frac{18\tilde{\gamma}_{k,i,j}^2}{(1-\beta)^8 R^{\mathrm{MDP}}(\pi)/2}$. That is, for all sufficiently large $n$, we have

$$\mathbb{E}_v^{\mathrm{RL\text{-}LOW\text{-}MDP}}\left[R^{\mathrm{MDP}}(\hat{\pi}_{\mathrm{out}})\right] \leq \sum_{\pi \neq \pi^*} R^{\mathrm{MDP}}(\pi) \sum_{k \in \mathcal{S}: \pi(k) \neq \pi^*(k)} \exp\left(-\frac{2n(R^{\mathrm{MDP}}(\pi))^2}{C\gamma_{k,\pi(k),\pi^*(k)}}\right)$$

which further implies that for all sufficiently large $n$, we have

$$\mathbb{E}_v^{\mathrm{RL\text{-}LOW\text{-}MDP}}\left[R^{\mathrm{MDP}}(\hat{\pi}_{\mathrm{out}})\right] \leq \exp\left(-\frac{n}{C_{\mathrm{MDP}} \cdot H^{\mathrm{MDP}}(v)}\right).$$

This completes the proof of Theorem 6.1.

$\square$

# H  THEORETICAL ANALYSIS OF DP-RL-LOW

## H.1  INSTANCE-DEPENDENT UPPER BOUND

**Lemma H.1.** *Fix any $\varepsilon > 0$ and $\delta > 0$. Let $Y_n$ be a random variable sampled from the binomial distribution with number of trials $n \in \mathbb{N}$ and probability of success $p \in [1-\beta, \beta]$ for $\beta \in (\frac{1}{2}, 1)$. Then,*

$$|\mathbb{E}(f(\tilde{X}_n)) - f(\mathbb{E}(\tilde{X}_n))| \leq \frac{3}{(1-\beta)^4 \sqrt{2n}} + \frac{4\sqrt{\log(1.25/\delta)}}{(1-\beta)^4 (\varepsilon n)},$$

*where $\tilde{X}_n = \frac{Y_n}{n} + \tilde{\xi}_n$, and $\tilde{\xi}_n$ is an independent Gaussian noise with zero mean and variance of $\frac{2\log(1.25/\delta)}{(\varepsilon n)^2}$, and $f(\cdot)$ is defined in (59).*

*Proof of Lemma H.1.* For simplicity of notation, we let $x_0 := \mathbb{E}(\tilde{X}_n)$, which implies $x_0 = p \in [1-\beta, \beta]$. Again, by the smoothness of $f(\cdot)$ on $[1-\beta, \beta]$, we get the Taylor expansion of $f(\cdot)$ on $[1-\beta, \beta]$,

$$f(x) = \begin{cases} f(x_0) + f'(x_0) \cdot (x - x_0) + \frac{1}{2}f''(\xi_x) \cdot (x - x_0)^2 & \text{if } x \in [1-\beta, \beta] \\ f(x_0) + f'(x_0) \cdot (\beta - x_0) + \frac{1}{2}f''(\xi_\beta) \cdot (\beta - x_0)^2 & \text{if } x > \beta \\ f(x_0) + f'(x_0) \cdot (1 - \beta - x_0) + \frac{1}{2}f''(\xi_\beta) \cdot (1 - \beta - x_0)^2 & \text{if } x < 1 - \beta, \end{cases} \tag{83}$$

where $\xi_x \in (\min(x, x_0), \max(x, x_0))$ that only depends on $x$ and $x_0$ in the Talor expansion.

Similarly, by the fact that $f'(x_0) \cdot (\beta - x_0) = f'(x_0) \cdot (\beta - x) + f'(x_0) \cdot (x - x_0)$ and $f'(x_0) \cdot (1 - \beta - x_0) = f'(x_0) \cdot (1 - \beta - x) + f'(x_0) \cdot (x - x_0)$, we can get from (83),

$$|\mathbb{E}(f(\tilde{X}_n)) - f(\mathbb{E}(\tilde{X}_n))|$$
$$\leq \sup_{x \in (1-\beta, \beta)} |f''(x)| \mathrm{Var}(\tilde{X}_n) + |f'(x_0)| \cdot \mathbb{E}([\tilde{X}_n - \beta]_+) + |f'(x_0)| \cdot \mathbb{E}([1 - \beta - \tilde{X}_n]_+) \tag{84}$$

where $\mathrm{Var}(\cdot)$ represents the variance and $[x]_+ = \max\{x, 0\}$.

In addition, we note that

$$\mathbb{E}([\tilde{X}_n - \beta]_+) \leq \int_\beta^1 (x - \beta)\mathbb{P}(\tilde{X}_n \geq x)\,\mathrm{d}x$$
$$\overset{(a)}{\leq} \int_\beta^1 (x - \beta)\exp\left(-\frac{(x - x_0)^2}{2(\frac{1}{4n} + \frac{2\log(1.25/\delta)}{(\varepsilon n)^2})}\right)\mathrm{d}x$$
$$\leq \int_\beta^1 (x - \beta)\exp\left(-\left(2n(x - x_0)^2 \wedge \frac{(x - x_0)^2(\varepsilon n)^2}{4\log(1.25/\delta)}\right)\right)\mathrm{d}x$$
$$= \int_\beta^1 (x - \beta)\exp\left(-(x - x_0)^2\left(2n \wedge \frac{(\varepsilon n)^2}{4\log(1.25/\delta)}\right)\right)\mathrm{d}x$$

$$\leq \int_\beta^1 (x-\beta) \exp\left(-(x-\beta)^2 \left(2n \wedge \frac{(\varepsilon n)^2}{4\log(1.25/\delta)}\right)\right) \, dx$$

$$\overset{(b)}{\leq} \int_\beta^1 \sqrt{\frac{1}{2n \wedge \frac{(\varepsilon n)^2}{4\log(1.25/\delta)}}} \, dx$$

$$= \frac{1-\beta}{\sqrt{2n \wedge \frac{(\varepsilon n)^2}{4\log(1.25/\delta)}}}, \tag{85}$$

where (a) follows from the fact that $\tilde{X}_n$ is subGaussian with variance proxy $\frac{1}{4n} + \frac{2\log(1.25/\delta)}{(\varepsilon n)^2}$, and (b) follows from the fact that $x\exp(-x^2 y) < \sqrt{\frac{1}{y}}$ for any $y > 0$. Similarly, we also can get that

$$\mathbb{E}([1-\beta-X_n]_+) \leq \frac{1-\beta}{\sqrt{2n \wedge \frac{(\varepsilon n)^2}{4\log(1.25/\delta)}}}. \tag{86}$$

Finally,

$$|\mathbb{E}(f(\tilde{X}_n)) - f(\mathbb{E}(\tilde{X}_n))|$$

$$\overset{(a)}{\leq} \sup_{x\in(1-\beta,\beta)} |f''(x)|\mathrm{Var}(\tilde{X}_n) + |f'(x_0)| \cdot \mathbb{E}([\tilde{X}_n - \beta]_+) + |f'(x_0)| \cdot \mathbb{E}([1-\beta-\tilde{X}_n]_+)$$

$$\overset{(b)}{\leq} \frac{1}{(1-\beta)^4} \cdot \left(\frac{1}{n} + \frac{2\log(1.25/\delta)}{(\varepsilon n)^2}\right) + f'(x_0) \cdot \mathbb{E}([X_n - \beta]_+) + f'(x_0) \cdot \mathbb{E}([1-\beta-X_n]_+)$$

$$\overset{(c)}{\leq} \frac{1}{(1-\beta)^4} \cdot \left(\frac{1}{n} + \frac{2\log(1.25/\delta)}{(\varepsilon n)^2}\right) + f'(x_0) \cdot \frac{2}{\sqrt{2n \wedge \frac{(\varepsilon n)^2}{4\log(1.25/\delta)}}}$$

$$\overset{(d)}{\leq} \frac{1}{(1-\beta)^4} \cdot \left(\frac{1}{n} + \frac{2\log(1.25/\delta)}{(\varepsilon n)^2}\right) + \frac{1}{(1-\beta)^2} \cdot \frac{2}{\sqrt{2n \wedge \frac{(\varepsilon n)^2}{4\log(1.25/\delta)}}}$$

$$\leq \frac{3}{(1-\beta)^4\sqrt{2n}} + \frac{4\sqrt{\log(1.25/\delta)}}{(1-\beta)^2(\varepsilon n)} + \frac{2\log(1.25/\delta)}{(1-\beta)^4(\varepsilon n)^2} \tag{87}$$

$$\leq \frac{3}{(1-\beta)^4\sqrt{2n}} + \frac{4}{(1-\beta)^4} \cdot \left(\frac{\sqrt{\log(1.25/\delta)}}{(\varepsilon n)} \vee \frac{\log(1.25/\delta)}{(\varepsilon n)^2}\right)$$

$$\overset{(e)}{\leq} \frac{3}{(1-\beta)^4\sqrt{2n}} + \frac{4\sqrt{\log(1.25/\delta)}}{(1-\beta)^4(\varepsilon n)}$$

where (a) follows from Eqn. (84), (b) follows from the fact that $\mathrm{Var}(\tilde{X}_n) \leq \frac{1}{n} + \frac{2\log(1.25/\delta)}{(\varepsilon n)^2}$ and that $f''(x) = \frac{2x-1}{(x-1)^2 x^2}$, (c) follows from Eqn. (85) and Eqn. (86), (d) follows from the fact that $f'(x) = \frac{1}{x-x^2}$, and (e) follows the fact that $|\mathbb{E}(f(\tilde{X}_n)) - f(\mathbb{E}(\tilde{X}_n))| \leq \frac{4}{(1-\beta)^4}$ and that $\sqrt{x} \geq x$ for any $x \in (0,1]$ $\qquad\square$

**Lemma H.2.** *Fix any $\varepsilon > 0$ and $\delta > 0$. Let $Y_n$ be a random variable sampled from the binomial distribution with number of trials $\in \mathbb{N}$ and probability of success $p \in [1-\beta, \beta]$ for $\beta \in (\frac{1}{2}, 1)$. Let $\tilde{X}_n = \frac{Y_n}{n} + \tilde{\xi}_n$, and $\tilde{\xi}_n$ is an independent Gaussian noise with zero mean and variance of $\frac{2\log(1.25/\delta)}{(\varepsilon n)^2}$. Then, $f(\tilde{X}_n)$ is subGaussian with*

$$\|f(\tilde{X}_n)\|_{vp}^2 \leq C \cdot \left(\frac{1}{n} + \frac{8\log(1.25/\delta)}{(\varepsilon n)^2}\right),$$

*where $C \leq (6\sqrt{2e} \cdot (3\sqrt{\log 2} + 1))^2 \cdot \frac{3}{2(1-\beta)^4}$ and $f(\cdot)$ is defined in (59).*

*Proof.* By the definition of $X_n$, we get that

$$\|\tilde{X}_n - p\|_{\text{vp}}^2 \le \frac{1}{4n} + \frac{2\log(1.25/\delta)}{(\varepsilon n)^2}.$$

Then, by Lemma C.5, we get that

$$\|\tilde{X}_n - p\|_{\phi_2}^2 \le \frac{3}{2n} + \frac{12\log(1.25/\delta)}{(\varepsilon n)^2}.$$

By the fact that $|f'(x)| \le \frac{1}{(1-\beta)^2}$, we get that

$$\|f(\tilde{X}_n) - f(p)\|_{\phi_2}^2 \le \left( \frac{3}{2n} + \frac{12\log(1.25/\delta)}{(\varepsilon n)^2} \right) \cdot \frac{1}{(1-\beta)^4}. \tag{88}$$

Similarly, from Lemma C.5, we further get that

$$\|f(\tilde{X}_n) - f(p)\|_{\text{vp}}^2 \le (6\sqrt{2e} \cdot (3\sqrt{\log 2} + 1))^2 \cdot \frac{3}{2(1-\beta)^4} \left( \frac{1}{n} + \frac{8\log(1.25/\delta)}{(\varepsilon n)^2} \right). \tag{89}$$

Note that by definition, we have $\|f(\tilde{X}_n) - f(p)\|_{\text{vp}}^2 = \|f(\tilde{X}_n)\|_{\text{vp}}^2$, which completes the proof of Lemma G.2. $\qquad\square$

*Proof of Theorem 5.3.* Fix any consistent instance $v$ and $n > 0$ under DP-RL-LOW. By lemma G.2 and lemma C.3, we get that $\tilde{r}_{k,i_k^*,i}$ is subGaussian with variance proxy to be,

$$\|\tilde{r}_{k,i_k^*,i}\|_{\text{vp}}^2 \le C \cdot \left( \frac{\gamma_{k,i}}{n} + \frac{8\gamma_{k,i}^{\text{DP}}\log(1.25/\delta)}{(\varepsilon n)^2} \right), \tag{90}$$

where $C \le (6\sqrt{2e} \cdot (3\sqrt{\log 2} + 1))^2 \cdot \frac{3}{2(1-\beta)^4}$ and $\beta = \frac{\exp(2L)}{1+\exp(2L)}$.

In addition, by Lemma H.1, we get that for any $(k', i', j') \in \mathcal{S} \times \mathcal{A}^2$ with $N_{k',i',j'} > 0$,

$$\left| \mathbb{E}_v^{\text{DP-RL-LOW}} \left[ f(\tilde{B}_{k',i',j'}) \right] - f\left( \mathbb{E}_v^{\text{DP-RL-LOW}} \left[ \tilde{B}_{k',i',j'} \right] \right) \right|$$

$$\le \frac{3}{(1-\beta)^4\sqrt{2nN_{k',i',j'}}} + \frac{4\sqrt{\log(1.25/\delta)}}{(1-\beta)^4(\varepsilon nN_{k',i',j'})}, \tag{91}$$

which implies for any $(k,i) \in \mathcal{S} \times \mathcal{A}$ with $i \ne i_k^*$,

$$\left| \sum_{k'\in\mathcal{S},i'\in\mathcal{A},j'\in\mathcal{A}} w_{k',i',j'}^{(k,i_k^*,i)} \mathbb{E}_v^{\text{DP-RL-LOW}} \left[ f(B_{k',i',j'}) \right] - f(\mathbb{E}_v^{\text{DP-RL-LOW}} \left[ B_{k',i',j'} \right]) \right|$$

$$\le \sum_{k'\in\mathcal{S},i'\in\mathcal{A},j'\in\mathcal{A}:N_{k',i',j'}\ne 0} |w_{k',i',j'}^{(k,i_k^*,i)}| \left( \frac{3}{(1-\beta)^4\sqrt{2nN_{k',i',j'}}} + \frac{4\sqrt{\log(1.25/\delta)}}{(1-\beta)^4(\varepsilon nN_{k',i',j'})} \right). \tag{92}$$

Recall that we denote $r_{k,i^*k,i} = r_{k,i_k^*} - r_{k,i}$. Then, from Eqn. (92), we get

$$\left| \mathbb{E}_v^{\text{DP-RL-LOW}} \left[ \tilde{r}_{k,i_k^*,i} \right] - r_{k,i_k^*,i} \right|$$

$$\le \sum_{k'\in\mathcal{S},i'\in\mathcal{A},j'\in\mathcal{A}:N_{k',i',j'}\ne 0} |w_{k',i',j'}^{(k,i,j)}| \left( \frac{3}{(1-\beta)^4\sqrt{2nN_{k',i',j'}}} + \frac{4\sqrt{\log(1.25/\delta)}}{(1-\beta)^4(\varepsilon nN_{k',i',j'})} \right) \tag{93}$$

$$\le \frac{3\tilde{\gamma}_{k,i}}{(1-\beta)^4\sqrt{n}} + \frac{4\tilde{\gamma}_{k,i}^{DP}\sqrt{\log(1.25/\delta)}}{(1-\beta)^4(\varepsilon n)}, \tag{94}$$

where $\tilde{\gamma}_{k,i}^{\text{DP}} := \sum_{k'\in\mathcal{S},i'\in\mathcal{A},j'\in\mathcal{A}:N_{k',i',j'}\ne 0} \frac{|w_{k',i',j'}^{(k,i_k^*,i)}|}{N_{k',i',j'}}$

Note that by the definition of $R_n$, under DP-RL-LOW we have

$$R_n \le \sum_{(k,i)\in\mathcal{S}\times\mathcal{A}:i\ne i_k^*} \mathbb{1}_{\{\tilde{r}_{k,i_k^*,i}<0\}} \cdot \rho_k \Delta_{k,i} \tag{95}$$

$$\leq \sum_{(k,i) \in \mathcal{S} \times \mathcal{A}: i \neq i_k^*} \left[ \left( \mathbb{1}_{\{\tilde{r}_{k,i_k^*,i} \leq \mathbb{E}_v^{\text{DP-RL-LOW}}[\tilde{r}_{k,i_k^*,i}] - \Delta_{k,i}/2\}} \right. \right.$$

$$\left. \left. \vee \mathbb{1}_{\{|\mathbb{E}_v^{\text{DP-RL-LOW}}[\tilde{r}_{k,i_k^*,i}] - r_{k,i_k^*,i}| \leq \Delta_{k,i}/2\}} \right) \cdot \rho_k \Delta_{k,i} \right] \tag{96}$$

By Lemma C.4 and (90), we get that for any $(k,i) \in \mathcal{S} \times \mathcal{A}$ with $i \neq i_k^*$,

$$\mathbb{P}_v^{\text{DP-RL-LOW}} \left( \tilde{r}_{k,i_k^*i} \leq \mathbb{E}_v^{\text{DP-RL-LOW}}[\tilde{r}_{k,i_k^*,i}] - \Delta_{k,i}/2 \right)$$

$$\leq \exp\left( -\frac{2\Delta_{k,i}^2}{C \cdot (\gamma_{k,i}/n + 8\gamma_{k,i}^{\text{DP}} \log(1.25/\delta)/(\varepsilon n)^2)} \right), \tag{97}$$

and by (94) we get that

$$|\mathbb{E}_v^{\text{DP-RL-LOW}}[\tilde{r}_{k,i_k^*,i}] - r_{k,i_k^*,i}| \leq \Delta_{k,i}/2$$

for all $n > \frac{12\tilde{\gamma}_{k,i}^2}{(1-\beta)^8 \Delta_{k,i}^2} + \frac{8\tilde{\gamma}_{k,i}^{DP} \sqrt{\log(1.25/\delta)}}{(1-\beta)^4 (\varepsilon \Delta_{k,i})}$. That is, for all sufficiently large $n$, we have

$$\mathbb{E}_v^{\text{RL-LOW}}(R_n) \leq \sum_{k \in \mathcal{S}, i \in \mathcal{A}: i \neq i_k^*} \rho_k \Delta_{k,i} \exp\left( -\frac{2\Delta_{k,i}^2}{C(\gamma_{k,i}/n + 8\gamma_{k,i}^{\text{DP}} \log(1.25/\delta)/(\varepsilon n)^2)} \right),$$

which further implies that for all sufficiently large $n$, there exists a global constant $C_{\text{DP}}$, we have

$$\mathbb{E}_v^{\text{RL-LOW}}(R_n) \leq \exp\left( -C_{\text{DP}} \cdot \left( \frac{n}{H(v)} \wedge \left( \frac{n}{H_{\text{DP}}^{(\varepsilon,\delta))}(v)} \right)^2 \right) \right).$$

This completes the proof of Theorem 5.3 $\qquad \square$

## H.2 WORST-CASE UPPER BOUND

**Proposition H.3.** *(Worst-Case Upper Bound for DP-RL-LOW) For any consistent instance $v$ and for all $n \geq 1$,*

$$\mathbb{E}_v^{\text{DP-RL-LOW}}[R_n] \leq C_{\text{WDP}} \cdot \left( \frac{\sum_{k,i:i \neq i_k^*} \rho_k(\sqrt{\gamma_{k,i}} + \tilde{\gamma}_{k,i})}{\sqrt{n}} + \frac{\sum_{k,i:i \neq i_k^*} \rho_k(\sqrt{\gamma_{k,i}^{\text{DP}}} + \tilde{\gamma}_{k,i}^{\text{DP}})\sqrt{\log(1.25/\delta)}}{\epsilon n} \right) \tag{98}$$

*where $C_{\text{WDP}} > 0$ is a universal constant.*

*Proof of Proposition H.3.* Note that by the definition of $R_n$ we have

$$R_n \leq \sum_{(k,i) \in \mathcal{S} \times \mathcal{A}: i \neq i_k^*} \mathbb{1}_{\{\tilde{r}_{k,i_k^*,i} \leq 0\}} \cdot \rho_k \Delta_{k,i}$$

$$\overset{(a)}{\leq} \sum_{(k,i) \in \mathcal{S} \times \mathcal{A}: i \neq i_k^*} \rho_k \left[ \mathbb{1}_{\{\Delta_{k,i} < \frac{6\tilde{\gamma}_{k,i}}{(1-\beta)^4 \sqrt{n}} + \frac{48\tilde{\gamma}_{k,i}^{\text{DP}} \sqrt{\log(1.25/\delta)}}{(1-\beta)^4(\varepsilon n)}\}} \cdot \left( \frac{6\tilde{\gamma}_{k,i}}{(1-\beta)^4 \sqrt{n}} + \frac{8\tilde{\gamma}_{k,i}^{DP} \sqrt{\log(1.25/\delta)}}{(1-\beta)^4 (\varepsilon n)} \right) \right.$$

$$\left. + \mathbb{1}_{\{|\mathbb{E}_v^{\text{DP-RL-LOW}}[\tilde{r}_{k,i,i_k^*}] - r_{k,i,i_k^*}| \leq \Delta_{k,i}/2\}} \cdot \Delta_{k,i} \right] \tag{99}$$

where $\beta = \frac{\exp(2L)}{1+\exp(2L)}$, and (a) follows from Eqn. (94).

Hence, by Eqn. (99) and (97), we have

$$\mathbb{E}_v^{\text{DP-RL-LOW}}[R_n] \leq \sum_{(k,i) \in \mathcal{S} \times \mathcal{A}: i \neq i_k^*} \rho_k \left[ \left( \frac{6\tilde{\gamma}_{k,i}}{(1-\beta)^4 \sqrt{n}} + \frac{8\tilde{\gamma}_{k,i}^{DP} \sqrt{\log(1.25/\delta)}}{(1-\beta)^4 (\varepsilon n)} \right) \right.$$

$$+ \exp\left(-\frac{2\Delta_{k,i}^2}{C \cdot (\gamma_{k,i}/n + 8\gamma_{k,i}^{\mathrm{DP}} \log(1.25/\delta)/(\varepsilon n)^2)}\right) \Delta_{k,i}\Bigg]$$

$$\leq \sum_{(k,i)\in\mathcal{S}\times\mathcal{A}:i\neq i_k^*} \rho_k \Bigg[ \left(\frac{6\tilde{\gamma}_{k,i}}{(1-\beta)^4\sqrt{n}} + \frac{8\tilde{\gamma}_{k,i}^{DP}\sqrt{\log(1.25/\delta)}}{(1-\beta)^4(\varepsilon n)}\right)$$

$$+ \sqrt{\frac{C\cdot\gamma_{k,i}}{2n}} + \sqrt{\frac{C\cdot 8\gamma_{k,i}^{\mathrm{DP}}\log(1.25/\delta)}{2(\varepsilon n)^2}}\Bigg]$$

$$= \sum_{(k,i)\in\mathcal{S}\times\mathcal{A}:i\neq i_k^*} \rho_k \Bigg[ \frac{1}{\sqrt{n}} \cdot \left(\frac{6\tilde{\gamma}_{k,i}}{(1-\beta)^4} + \sqrt{\frac{C\cdot\gamma_{k,i}}{2}}\right)$$

$$+ \frac{1}{\varepsilon n} \cdot \left(\frac{8\tilde{\gamma}_{k,i}^{\mathrm{DP}}\sqrt{\log(1.25/\delta)}}{(1-\beta)^4} + \sqrt{\frac{C\cdot 8\gamma_{k,i}^{\mathrm{DP}}\log(1.25/\delta)}{2}}\right)\Bigg]$$

$$(100)$$

where $C \leq (6\sqrt{2e}\cdot(3\sqrt{\log 2}+1))^2 \cdot \frac{3}{2(1-\beta)^4}$.

This completes the desired proof. $\qquad\square$