# OpenReview forum: "Order-Optimal Instance-Dependent Bounds for  Offline Reinforcement Learning with Preference Feedback"
_ICLR.cc/2025/Conference — Submitted to ICLR 2025_

### Official Review · Reviewer_7eX6 · 2024-10-25

**Soundness:** 3
**Presentation:** 3
**Contribution:** 3
**Rating:** 6
**Confidence:** 4

**Summary:**

In this paper, the authors consider offline reinforcement learning (RL) with preference feedback in which the implicit reward is a linear function of an unknown parameter. They propose an algorithm, RL with Locally Optimal Weights (RL-LOW), which yields a simple regret which decays exponentially as n goes large. Furthermore, they derive a matching instance-dependent lower bound in offline RL with preference feedback. Moreover, they also extend RL-LOW to the setting of differentially private RL and show that the hardness parameter
 is unchanged in the asymptotic regime as n tends to infinity.

**Strengths:**

1. The problem of deriving an instance dependent sub-optimality bound for offline RLHF is well-motivated and interesting.

2. The paper presents the first instance dependent bounds for the setting, while the upper bound matches the lower bound in the order. The paper is solid and the proof looks correct to me.

3. The authors also extend the result to DP-RL setting and the MDP setting, which makes this paper a comprehensive work.

4. The simulation supporting the theoretical findings is convincing.

5. The paper is well-written in general.

**Weaknesses:**

1. The setting is kind of simple. For the bandit setting, although the reward is linear, the setting is actually tabular (finite states & actions). For the MDP setting, the transition kernel is known. While I understand to derive a gap-dependent analysis as in the paper, the setting is required to be tabular, it seems that the locally optimal weights heavily depend on the tabular nature of the problem.

2. The improvement on the worst-case bound is limited. Meanwhile, the instance-dependent upper bound seems to be well-expected, since given the minimum gap, the failure probability of identifying the optimal arm will decay exponentially as n converges to $\infty$.

3. Previous work [1] derived an instance-dependent sub-optimality upper bound $O(\frac{1}{\sqrt{n}}+\frac{1}{n\epsilon})$ for standard offline RL (explicit reward) with DP. The two terms in [1] seem to match the order of the two terms in (22) of Theorem 5.3. It will be interesting if the author could provide a worst-case version of Theorem 5.3 and compare with the result in [1] to further showcase the cost of privacy.

[1] Offline Reinforcement Learning with Differential Privacy, Dan Qiao and Yu-Xiang Wang.

Overall, I think the instance-dependent sub-optimality bound for offline RLHF is meaningful and the strengths outperform the weaknesses. Therefore, I lean towards acceptance. I will be happy to raise my score if my comments are solved.

**Questions:**

Please refer to the weaknesses above.

---

> ### Author Response · Authors · 2024-11-20
>
> **Q1**: "For the bandit setting, although the reward is linear, the setting is actually tabular (finite states \& actions). For the MDP setting, the transition kernel is known."
>
> **A1**:  The reviewer is right in their understanding of our setting. However, note that there is still no research on the instance-dependent analysis within the field of offline PbRL, regardless of whether the transition kernel is known or not and whether the number of states is finite or not. Our work is the first to provide instance-dependent (lower and upper) bounds for offline PbRL, and, we wish, it can inspire further research of instance-dependent analysis within the domain.
>
> ---
>
> **Q2**: The improvement on the worst-case bound is limited. Meanwhile, the instance-dependent upper bound seems to be well-expected, since given the minimum gap, the failure probability of identifying the optimal arm will decay exponentially as n converges to $+\infty$
>
> **A2**:  Indeed, the improvement of the worst-case bound is by an albeit small factor of $\sqrt{\log n}$. Note that the improvement of the worst-case bound is not our focus in this paper -- it is merely a by-product of our analyses. Our focus is on deriving *instance-dependent* bounds and showing their tightness. Yes, it decays exponentially in $n$, i.e, $\mathrm{Reg}\le\exp(-\Omega(n))$. Most importantly, we formally nail down the hardness constant $H(v)$ in the exponent (i.e.,  $\mathrm{Reg}\le\exp(- cn/H(v))$) and show, via a matching lower bound, that it is tight.
>
> ---
>
> **Q3**: Previous work [1] derived an instance-dependent sub-optimality upper bound  $\frac{1}{\sqrt{n}} + \frac{1}{\epsilon n}$ for standard offline RL (explicit reward) with DP. The two terms in [1] seem to match the order of the two terms in (22) of Theorem 5.3. It will be interesting if the author could provide a worst-case version of Theorem 5.3 and compare with the result in [1] to further showcase the cost of privacy.
>
> **A3**: Thank you very much for providing the interesting related work. We have provided the worst-case version of our Theorem 5.3 in the newly added Appendix H.2 of the revised manuscript. In summary, we obtain that the worst-case regret under $(\epsilon,\delta)$-DP yields the form of $\frac{1}{\sqrt{n}} + \frac{1}{\epsilon n}$ for the dependency of $n$ and $\varepsilon$, which is exactly what the reviewer expects. That is, it indeed aligns with the bounds of [1] without preference feedback. We have added these comments and cited the paper of [1] in Section 5 of our revised version. Thanks again for raising this study.
>
>
> [1] Offline Reinforcement Learning with Differential Privacy, Dan Qiao and Yu-Xiang Wang.
>
> ---
>
> Finally, we would like to express our heartfelt thanks again to the reviewer for recognizing our technical contributions, and we are welcome any further questions if the reviewer may have.
>
> (edit: update the appendix letter for the latest revised paper)

---

> > ### Comment · Reviewer_7eX6 · 2024-11-23
> >
> > Thanks for you detailed explanation. I will keep my positive evaluation of the paper.

---

### Official Review · Reviewer_dCJ6 · 2024-11-02

**Soundness:** 4
**Presentation:** 3
**Contribution:** 3
**Rating:** 8
**Confidence:** 4

**Summary:**

The work studies simple regret bounds of offline reinforcement learning with preference feedback. Formally, we are given a dataset generated as follows:
1. A state $k$ is drawn from some fixed distribution $\rho$.
2. Two actions $a_0, a_1$ are picked.
3. A Bernoulli variable $\sigma$ is generated based on the soft-max of the rewards associated with $(k, a_0), (k, a_1)$.

We want to assign each state $k$ a candidate action $a(k)$ so as to minimize the simple regret defined as $  \mathbf E_{ k \sim \rho } [  \text{reward}(k, a^*(k)) - \text{reward} (k, a(k))  ) ] $, where $a^*(k)$ is the action that maximizes the reward of state $k$.

The main contribution is an algorithm that achieves a simple regret bound of $\exp(- O(n/H))$, where $H$ is some complexity parameter that depends on the instance, and a lower bound construction showing that $\exp(- \Omega(n/H) )$ is necessary.

**Strengths:**

The main novelty of the algorithm is the design of Locally Optimal Weights. At a high level, the weights are used to weight to what extend one should take into account the information about a datapoint about state $k’$, action $i’$, and action $j’$ while comparing the rewards of action $i$ and action $j$ on state $k$. Moreover, the weights are constructed to minimize a variance proxy for the comparison result so as to enhance the estimation precision. The technique is sophisticated, and shown in experiment to have non-trivially improved the average simple regret bound.

**Weaknesses:**

The extension to the MDP setting, where one tries to find optimal policies, rather than just actions that maximize per-state rewards, is rather superficial. First, the authors assume that the transition probabilities are known a priori (though this is the case in prior work as well), which significantly weakens the algorithm’s applicability. Besides, there is likely no mention of the computation complexity of their algorithm in this setting. It seems like the algorithm is just doing brute-force to find a policy $\pi$ that maximizes $\mathbf E_{k \sim d^{\pi}}[ \hat r_{k, \pi(k),1} ]$, which is prohibitively expensive.

**Questions:**

1. How is $d^{\pi}$ formally defined? Is it the stationary distribution under the policy $\pi$?
2. It seems like the consistency requirement is mainly to ensure that the dataset at least contains enough information to compare each pair of actions $i,j$ on a state $k$. Is there a setting where such assumption is relaxed and one instead assumes that there is a joint distribution over state and action tuples so that one can ignore the highly unlikely state-action tuples in the final regret bound?

---

> ### Author Response · Authors · 2024-11-20
>
> **Q1**: On the computation complexity of finding a policy  that maximizes $\pi$ that maximize $\arg \max_{\pi} \\mathbb{E}\_{k \sim d^\pi} [\\hat{r}\_{k,\pi(k),1}]$
>
> **A1**: Thanks for the question. To calculate $\arg \max_{\pi} \\mathbb{E}\_{k \sim d^\pi} [\\hat{r}\_{k,\pi(k),1}]$,
> there are two steps.
>
> First, we calculate the empirical reward of $\hat{r}_{k,i,1}$ for each $(k,i)\in \mathcal{S}\times \mathcal{A}$, which takes computational complexity of $\mathcal{O}(SAd + nd^2 + d^3)$.
>
> Second, after we obtain the empirical reward of $\hat{r}\_{k,i,1}$, the next step is exactly the classical RL problem of finding the optimal MDP policy under the condition that the transition probabilities and rewards are known. This classical problem can be solved by using asynchronous dynamic programming [1], which typically takes $O(\kappa SA)$ time complexity, where $\kappa$ is a hyperparameter that represents the average number of iteration, which controls the solution precision. Hence, the overall computational complexity is $\mathcal{O}(SAd + nd^2 + d^3+\kappa SA)$; we have added more details in Appendix E.2.2 of our revised version to address this question. Thanks again for the precious question.
>
> ---
>
> **Q2**: How is $d^\pi$ formally defined? Is it the stationary distribution under the policy?
>
> **A2**: Yes, it bears high similarities with the stationary distribution to some extent. Generally, we adopt the notion of state distribution as is in Zhu et al [1], which is standardly defined in Sutton and Barto [2018, Section 9.2]. We have cited this in our revised version to avoid any ambiguity. Thank you very much for the question.
>
> Here we quote the paragraph of Section 9.2 of Sutton and Barto in the following, where they call it "on-policy distributioin":
>
>
> > Let $h(s)$ denote the probability that an episode begins in each state $s$, and let $\eta(s)$ denote the number of time steps spent, on average, in state $s$ in a single episode. Time is spent in a state $s$ if episodes start in $s$, or if transitions are made into $s$ from a preceding state $\bar{s}$ in which time is spent:
> $$
> \eta(s)=h(s)+\sum_{\bar{s}} \eta(\bar{s}) \sum_a \pi(a \mid \bar{s}) p(s \mid \bar{s}, a), \text { for all } s \in \mathcal{S}
> $$
> This system of equations can be solved for the expected number of visits $\eta(s)$. The on-policy distribution is then the fraction of time spent in each state normalized to sum to one:
> $$
> \mu(s)=\frac{\eta(s)}{\sum_{s^{\prime}} \eta\left(s^{\prime}\right)}, \text { for all } s \in \mathcal{S}
> $$
> This is the natural choice without discounting. If there is discounting $(\gamma<1)$ it should be treated as a form of termination, which can be done simply by including a factor of $\gamma$ in the second term of $\eta(s)$.
>
>
> ---
>
> **Q3**: It seems like the consistency requirement is mainly .... in the final regret bound?
>
> **A3**: Yes, the review is correct in noting that consistency ensures the dataset contains enough information to compare each pair of actions for a given state. This is formally supported by our Proposition 2.3, which shows that achieving vanishing regret is not possible for inconsistent instances. Therefore, our focus in this paper is on the investigation with consistent instances. We also appreciate the reviewer’s insightful idea regarding the idea of 'ignoring highly unlikely state-action tuples.' This is an interesting direction, and we leave it for future study.
>
> [1] Zhu et al. (2023). Principled reinforcement learning with human feedback from pairwise or k-wise comparisons. ICML 2023.
>
> [2] Sutton, R. S. and Barto, A. G. (2018). Reinforcement learning: An introduction. MIT press.
>
> ---
>
> Finally, we would like to extend our sincere gratitude to the reviewer for acknowledging our technical contributions, particularly on the design of Locally Optimal Weights
>
> (edit: update the appendix letter for the latest revised paper)

---

### Official Review · Reviewer_tkFf · 2024-11-03

**Soundness:** 2
**Presentation:** 1
**Contribution:** 1
**Rating:** 3
**Confidence:** 4

**Summary:**

The paper studies an offline preference-based policy learning problem. This work utilizes the locally optimal weights to construct an instance-dependent algorithm. In theory, the instance-dependent upper bound and lower bound are derived. Further, an extension to the data-privacy setting is studied.

**Strengths:**

1.  The preference-based problem is important in the offline RL community.
2.  Some theoretical guarantees are established for the proposed algorithm.

**Weaknesses:**

1. This work considers a linear reward setting, which is more stringent compared to the general reward function setting in the existing literature: a. Chen, Xiaoyu, et al. "Human-in-the-loop: Provably efficient preference-based reinforcement learning with general function approximation." ICML b. Zhan, Wenhao, et al. "Provable offline preference-based reinforcement learning." ICLR. On the other hand, the reward functions are complex and usually require non-linear function approximation.

2. The reward function is not trajectory-based defined but is just defined on the state-action pairs. The state-action pair setting is inconsistent with the practical applications, particularly natural language processing. See Ramamurthy, Rajkumar, et al. "Is reinforcement learning (not) for natural language processing: Benchmarks, baselines, and building blocks for natural language policy optimization." ICLR.

3. This paper mentions the underlying problem is a PbRL. However, following careful reading of the setting in Section 2, the problem considered in the paper is essentially a contextual bandit problem. This greatly limits the contribution of the work.

3. The instance-dependent setup is not well explained in the preference-based settings. Particularly, the coverage of the offline data is not discussed in the work.

4. Definition 2.2: The condition on the consistent instance is strong, which is usually not true in general offline settings.

5. Equation (6): The clip on the implicit reward is difficult. That is, the bound in the implicit reward is not easy to measure in practice.

6. The estimation of the locally optimal weight seems very sensitive over the low-coverage samples. Also, the computational complexity is high for estimating the local optimal weight.

7. Equation (7): the condition of the consistent instance essentially implies good coverage following the definition of the locally optimal weight.

8. The instance-dependent upper bound seems superficial as compared to the setting in the general offline RL. For example, the dimension of the feature mapping is not in the bound. See Nguyen-Tang, Thanh, et al. "On instance-dependent bounds for offline reinforcement learning with linear function approximation." AAAI.  Some further explanation and discussion on this point are required.

9. The extension to a privacy-preserve setting is trivial, but the motivation for ensuring privacy in the PbRL problem is not well discussed.

10. There is a lack of comprehensive empirical studies on the algorithm. The evaluation of the offline PbRL benchmark is necessary. This can better position the paper in the existing literature.

11. The ablation studies on the instance-dependent upper bound and lower bound in terms of the key parameters are needed.

12. The presentation of the paper is not clear. There are many notations and some of them are not necessary.

**Questions:**

See weakness.

---

> ### Author Response · Authors · 2024-11-20
>
> **Q1**: "This work considers a linear reward setting, which is more stringent compared to the general reward function setting in the existing literature".
>
> **A1**: We agree that considering general reward functions is indeed more flexible. However, prior studies have not considered the derivation of instance-dependent upper and lower bounds even for the *linear reward setting*. Given that the linear case has not been done, we need to tackle it before considering more general reward settings, which subsume the linear rewards as a special case. We aim for this contribution to serve as a foundational step, offering valuable insights that may guide future research on instance-dependent bounds in more general reward settings.
>
> ---
>
> **Q2**: "The reward function is not trajectory-based defined but is just defined on the state-action pairs.
>
> **A2**: Thank you for highlighting this point. Indeed, our reward function is defined based on state-action pairs. We would like to emphasize that **both trajectory-based and state-action-based frameworks are valuable** but complementary directions in PbRL. For example, in a recent technical report of Ahmadian et al. [1], which is adopted by the TRL library of Huggingface, the authors modelled the entire response as a single action (see in their Section 3.3) rather than use it as a trajectory, which exactly aligns with our Section 2.
>
> ---
>
> **Q3**: "This paper mentions the underlying problem is a PbRL. However, following careful reading of the setting in Section 2, the problem considered in the paper is essentially a contextual bandit problem. This greatly limits the contribution of the work."
>
> **A3**: Thank you for your insightful comments. The reviewer is right that the setting described in Section 2 bears similarities with contextual bandit problems to some extent. However, we wish to draw the reviewer's attention to Section 6, in which the results are extended to include MDPs, thus bringing more realism to the PbRL problem.
>
> ---
>
> **Q4**: "The instance-dependent setup is not well explained in the preference-based settings. Particularly, the coverage of the offline data is not discussed in the work.
>
> **A4**: We are not sure if "the coverage of the offline data" mentioned by the reviewer is exactly the same as that Zhu et al. [2] under offline PbRL. We acknowledge that the coverage of offline data plays a significant role in the worst-case analysis presented by Zhu et al. However, we must admit that our instance-dependent analysis does not currently establish an explicit connection to the concept of "coverage of offline data."
>
> We remain open to further exploration of this topic and would greatly appreciate any insights the reviewer might provide regarding how "coverage of offline data" could be integrated or related to our instance-dependent analysis.
>
> ---
>
> **Q5**: "Definition 2.2: The condition on the consistent instance is strong, which is usually not true in general offline settings."
>
> **A5**:  Our focus on the consistent instance is supported by Proposition 2.3, which states that it is not possible to get vanishing regret in inconsistent instances. Thus, the notion of consistency is required in our analyses, without which vanishing regret cannot be obtained by any algorithm. Thanks again for the observation.
>
> ---
>
> **Q6**: "Equation (6): The clip on the implicit reward is difficult. That is, the bound in the implicit reward is not easy to measure in practice."
>
>
> **A6**: Thanks for the comment. We would like to clarify that our lower and upper bounds do not rely on the CLIP function defined in Equation (6). The CLIP function is utilized solely as a component within our proposed algorithm; this CLIP operation in RL-LOW is efficient and easy to implement.
>
> ---
>
> **Q7**: The estimation of the locally optimal weight seems very sensitive over the low-coverage samples. Also, the computational complexity is high for estimating the local optimal weight.
>
> **A7**: The overall computational complexity of our algorithm is $O(SAd + nd^2 + d^3)$, where the term "$SAd$" corresponds to basic requirement to scan through the feature vectors for all state-action pairs, and the term "$nd^2 + d^3$" corresponds to the complexity of calculating the inverse of a matrix.
>
> In addition, we wish to clarify that the computational complexity's dependency "$SAd$" is inevitable, as we require to output $\hat{i}_k$ for all  $k \in \mathcal{S}$ in the setting of our manuscript. Nonetheless, if we remove this output's requirement, the term "$SAd$" indeed be removed in our algorithm, and the overall computational complexity beccomes $O(nd^2+d^3)$.
> We have added more details in Appendix E.2 of our revised version to address this issue. Thanks again for the comment of computational complexity.
>
> ---
> (edit: update the appendix letter for the latest revised paper)

---

> ### Author Response · Authors · 2024-11-20
>
> **Q8**: Equation (7): the condition of the consistent instance essentially implies good coverage following the definition of the locally optimal weight.
>
> **A8**: We are thankful to the reviewer for providing an insightful observation. In our setting, we wish to highlight that the  consistency is required for us to obtain non-vacuous instance-dependent bounds as shown in Proposition 2.3.
>
> ---
>
> **Q9**: "The instance-dependent upper bound seems superficial as compared to the setting in the general offline RL. For example, the dimension of the feature mapping is not in the bound". See Nguyen-Tang, Thanh, et al.
>
> **A9**: We thank the reviewer for providing a thoughtful opinion. The dimension is indeed not explicitly in our instance-dependent bound.
>
> For example, let's say $S=1, A=2$, and $d$ is sufficiently large. In addition $\phi(1,1)=[0,1,0,0,\cdots]$, $\phi(1,2)=[1,0,0,0,\cdots]$, and $\theta=[0,1,0,0,\cdots]$. Then, we have $r_{1,1}=0$ and $r_{1,2}=1$. Given the offline dataset with size $n$ (i.e., these are the $n$ historical records for "action $1$ wins action $2$" or "action $2$ wins action $1$" ) under our setting, if the learner picks the action with the most winning records (as it does in our RL-LOW), then by Hoeffding's inequality it will suffer an expected regret of $\exp(-\Omega(n))$ which does not depend on the dimension $d$, and it explains why our upper bound does not explicitly depend on $d$.
>
> In addition, we would like to draw the reviewers' attention that our upper bound yields an exponential decay while Nguyen et al. [3] does not, due to different settings that necessitate different proof techniques.
>
> Thanks again for the careful observation, which enforces the novelty of our upper bound and shows that the setting and techniques differ significantly from Nguyen et al.
>
> ---
>
> **Q10**: The extension to a privacy-preserve setting is trivial, but the motivation for ensuring privacy in the PbRL problem is not well discussed.
>
> **A10**: Our motivation aligns closely with that of Chowdhury et al. [4]. For example, in certain labelling procedures for question-answering (QA) systems, users provide preference labels between two responses for a given question rather than directly submitting their own responses to the question. Consequently, our work prioritizes safeguarding the privacy of these preference labels rather than the privacy of the questions or responses themselves.
>
> ---
>
> **Q11, Q12**: The issues in empirical study.
>
> **A11,A12**: We sincerely appreciate the reviewer's thorough and comprehensive evaluation of the empirical studies. While we fully acknowledge and respect these observations, we would like to kindly note that the primary focus of our paper is theoretical, i.e., the establishment of the instance-dependent upper and lower bounds that are {\em tight} in the hardness parameter. In light of this, we hope the reviewer might consider adopting an optimistic perspective on this aspect. Thanks again for the detailed and thoughtful comment.
>
> ---
>
> **Q13**: "The presentation of the paper is not clear. There are many notations and some of them are not necessary."
>
> **A13**:  We attach great importance to simplified and clear notations in article writing. We would be very grateful if the reviewer could provide more details on this point.
>
>
> [1] Ahmadian, et al. (2024). Back to basics: Revisiting reinforce style optimization for learning from human feedback in llms. arXiv preprint arXiv:2402.14740
>
> [2] Zhu et al. (2023). Principled reinforcement learning with human feedback from pairwise or k-wise comparisons. ICML 2023.
>
> [3]  Nguyen-Tang, Thanh, et al. "On instance-dependent bounds for offline reinforcement learning with linear function approximation." AAAI.
>
> [4] Chowdhury et al. (2024). Differentially private reward estimation with preference feedback. In International Conference on Artificial Intelligence and Statistics (pp. 4843-4851). PMLR.
>
> ---
>
> Finally, we would like to extend our sincere gratitude to the reviewer for providing the detailed review. We hope that the above replies to the esteemed reviewer are satisfactory. We remain open and happy to answer any follow-up questions the reviewer may have.

---

> > ### Author Response · Authors · 2024-11-25
> >
> > Dear Reviewer,
> >
> > We'd like to express our gratitude again for your dedication and commitment in reviewing our paper. We were wondering whether our responses to your questions are satisfactory. In particular, we would like to summarize some key points of our responses (to you and other reviewers) for your convenience of assessment:
> >
> > 1) We presented a simple example to explain why our bounds do not depend on $d$.
> >
> > 2) We clarified that the overall computational complexity is $O(nd^2+d^3)$ when removing the output's requirement, and we added more details of the computational complexity on our revised Appendix E.2.
> >
> > 3) We clarified that our setting is extended to MDPs in Section 6.
> >
> > 4) We clarified the motivation for our extension to label differential privacy in the newly added Appendix A.2.
> >
> > 5) We provided more clarification on the intuitive meaning of the instance-dependent hardness parameter $H(v)$ in the newly added Appendix B.
> >
> > We hope that our clarifications align satisfactorily with your expectations. Your assessment would be immensely influential in shaping the trajectory of our paper, and we hope to engage with you in the remaining time for the author-reviewer discussion period. Thanks again for your effort.
> >
> > Best Regards,
> >
> > The authors

---

> > ### Comment · Reviewer_tkFf · 2024-11-26
> > **Thank you for the explanations and rebuttal!**
> >
> > I sincerely appreciate the authors' thoughtful rebuttal and detailed explanations addressing the comments. While I am satisfied with most of the responses, there remain several major concerns, particularly regarding the fundamental settings upon which the work is based. These issues remain unresolved, and thus, I maintain my original score.

---

> > > ### Author Response · Authors · 2024-11-26
> > >
> > > We are happy to hear that you are satisfied with most of our responses, and we appreciate the reviewer's continued feedback.
> > >
> > > ---
> > >
> > > **Q14**: "There remain several major concerns, particularly regarding the fundamental settings upon which the work is based."
> > >
> > > **A14**: We would be greatly grateful if you could kindly specify the details of your remaining concerns, which would help us improve the paper.
> > >
> > > Our setting is consistent with Zhu et al. [2] (which has been well-cited as a landmark work in theoretical PbRL), although our results complement those of Zhu et al.'s. Besides the motivations described by Zhu et al.,  we have clarified that:
> > >
> > > - **The motivation that the reward is unknown but the transition is known.** This is pertinent in most language models because the states are represented by prompts and the actions are represented by the next response (or next token). For example, let's say the current state $s=$"the capital of America is". If we take action $a$="Washington", then next state $s'$ is deterministic and known to be $s'$="the capital of America is Washington". That is, the rewards are unknown, but the transitions are indeed known in this context. We have mentioned this motivation in the introduction of the paper.
> > >
> > > - **The motivation for using data of state-action pairs.**
> > > Similarly, in practical applications of RLHF for LLMs, the types of data vary. Some utilize trajectory-based data, while others rely on state-action pair data, underscoring the significance of both types. For instance, the recent technical report by Ahmadian et al. [1], integrated into Huggingface's TRL library, uses the entire response as a singular action rather than using it as a trajectory, as detailed in Section 3.3 of their report, which resembles the setting in Section 2 of our paper. We have cited Ahmadian et al.\ in the introduction of our revised paper.
> > >
> > > ---
> > >
> > > We hope our above motivations are satisfactory. If not,  we would deeply appreciate it if you could kindly provide more details about your concerns (e.g., about the "fundamental settings"), which would be very helpful for improving the manuscript.

---

### Official Review · Reviewer_1qP9 · 2024-11-03

**Soundness:** 3
**Presentation:** 2
**Contribution:** 2
**Rating:** 5
**Confidence:** 3

**Summary:**

This paper introduces RL-LOW, an algorithm aimed at minimizing simple regret in offline reinforcement learning (RL) with preference feedback.

The paper claims that RL-LOW achieves tight instance-dependent bounds, offering theoretical guarantees beyond existing minimax approaches.

Additionally, the algorithm is extended to a differential privacy setting, maintaining its instance-dependent guarantees.

**Strengths:**

1. **Instance-Dependent Bounds**:

The algorithm’s ability to achieve instance-dependent error bounds is a major strength. This focus provides finer theoretical guarantees compared to minimax bounds.

2. **Theoretical Contributions**:

The paper does a thorough job of proving both upper and lower bounds for the regret, providing order-optimality guarantees in terms of the hardness parameter $H$.

**Weaknesses:**

1. **Limited Practical Relevance of Instance-Dependent Bounds**:

While the instance-dependent bounds are novel, I feel they are disconnected from practical applications. In many real-world scenarios, problem instances aren’t clearly characterized in terms of hardness, which limits the utility of these bounds. It is also unclear how people could use these refined bounds to benefit practice.

2. **Some parts Hard to Understand**:

The paper suffers from clarity issues throughout. Key concepts, such as the distinction between the instance-dependent and worst-case bounds, are not well-explained. For instance, the introduction of the hardness parameter $H$ is not intuitive, and sections like Definition 2.2 on consistent instances are particularly difficult to digest. More explanation or discussions would benefit understanding.

3. **Experiments Lack Validity**:

The experimental validation is weak. The simulations are based on a synthetic dataset, and while this is standard for theoretical work, the lack of real-world testing makes the results less convincing. Furthermore, the comparisons with PESSIMISTIC MLE are only on narrow setups, leaving questions about the general applicability of RL-LOW.

4. **Unclear Motivation for Differential Privacy**:

The introduction of differential privacy seems tacked on and not well-motivated. There’s little discussion on why privacy is critical for this particular problem setting, and the paper introduces the concept late in the related work section rather than making it part of the core problem setup. This reduces the relevance of the extension. Furthermore, it is also unclear whether the bound under different privacy (in Theorem 5.3) is tight or not in terms of $\epsilon$ and $\delta$.

**Questions:**

1. **BTL Model Assumptions**:

The Bradley–Terry–Luce (BTL) model used for preference feedback is quite limited in terms of modeling real human preferences. To what extent could the assumptions of this model be relaxed, and what impact would this have on the results?

2. **Algorithmic Novelty vs. Analytical Refinement**:

Is the prior theoretical guarantee of RL-LOW due to its algorithmic design or a more refined analysis? A comparison of RL-LOW’s performance with that of PESSIMISTIC MLE under the same conditions would help clarify this point.

3. **Clarification on Assumptions**:

Definition 2.2 needs further explanation, which appears somewhat confusing. This definition relies on the collected data $N_{k,i,j}$, which is random, while the concept of a consistent instance seems to be a property of the feature $\phi(k, i)$. This suggests that for a given feature $\phi(k, i)$, it may sometimes be consistent and other times not, depending on the data. Could the authors clarify why this assumption is necessary and provide more insight into its implications? Additionally, it would be helpful to understand how consistency is determined in practice.

4. **Clarification on "Instance-Dependent" Bound**:

The so-called instance-dependent bound appears more like a "gap-dependent" bound, as it heavily relies on the suboptimality gaps between actions. This raises the question of how the results would change if these gaps did not exist or were negligible. In such cases, would the proposed algorithm still achieve meaningful bounds, or would its performance degrade? Could the authors explain how the proposed method would handle instances where the suboptimality gaps are small or non-existent?

---

> ### Author Response · Authors · 2024-11-20
>
> **Q1**: "Limited Practical Relevance of Instance-Dependent Bounds."
>
> **A1**:  Thanks for the comment. We wish to highlight that the significance of instance-dependent bounds is to comprehensively justify that the regret is in the rate of $\exp(-\Omega(n))$ when it is dependent on the sub-optimality gap instead of $\frac{1}{\sqrt{n}}$ (which is only applicable in the worst case). In the study of multi-armed bandits, both worst-case and instance-dependent bounds are useful as they provide complementary insights into the difficulty of the problem. We hope our conclusions provide some insights for real-world applications, as is in many other theoretical papers.
>
> ---
>
> **Q2**: "Some parts Hard to Understand"
>
> **A2**: There are essentially two types of regret bounds in the literature -- worst-case and instance-dependent bounds. As the name suggests, the former involves deriving regret over worst-case instances over a certain class of instances. The latter, however, involves deriving regret for a fixed instance. Our derived hardness parameter in both the upper and lower bounds is also rather intuitive. It depends on various suboptimality gaps in an inverse manner. Essentially, the smaller the suboptimality gap, the larger of the hardness, which implies the instance is harder to learn. Furthermore, the introduction of a notion of consistency in Definition 2.2 is inspired by our Proposition 2.3, i.e., it is necessary for deriving non-vacuous regret bounds.  Thanks again for the question.
>
> ---
>
> **Q3**: Experiments Lack Validity:
>
> **A3**: We are thankful for the comment in terms of the empirical studies. We do not disagree with this point. However, given that the main focus of our paper is theoretical, we hope that the reviewer might consider the issue of the empirical study with an optimistic perspective. Thanks again for the valuable feedback.
>
> ---
>
> **Q4**: Unclear Motivation for Differential Privacy.it is also unclear whether the bound under different privacy (in Theorem 5.3) is tight or not in terms of $\delta$ and $\varepsilon$.
>
>
> **A4**:  Our motivation aligns closely with that of Chowdhury et al. [2]. For example, in certain labelling procedures for question-answering (QA) systems, users provide preference labels between two responses for a given question, rather than directly submitting their own responses to the question. Consequently, our work prioritizes safeguarding the privacy of these preference labels rather than the privacy of the questions or responses themselves.
>
> Furthermore, the tightness of the Theorem 5.3 is guaranteed when $n$ is larger than ${(H_{\rm DP}^{(\varepsilon,\delta)}(v))^2}/H(v)$ as explained at the end of Section 5. Thank you very much for submitting this point.
>
> ---
>
> **Q5**: On the BTL Model Assumptions
>
> **A5**: Note that the BTL model is very popular/prevalent for modeling human preference feedback. The reviewer is right that the real-world application may not exactly meet the BTL model, we leave this for future research, e.g., considering corruption, mis-specification in the BTL model or the presence of adversaries.

---

> > ### Author Response · Authors · 2024-11-20
> >
> > **Q6**: A comparison of RL-LOW’s performance with that of PESSIMISTIC MLE under the same conditions would help clarify this point.
> >
> > **A6**: The focus of PESSIMISTIC MLE is different from our paper, as they first provide a *worst-case* bound, while our paper is the first provide an *instance-dependent* bound. Nonetheless, we provide a comparison in the dependency of $n$ at the end of Section 3. That is, our worst-case bound is of the form $O(\frac{1}{\sqrt{n}})$, which is slightly superior to those of the form $O(\sqrt{\frac{\log(n)}{{n}})}$ in PESSIMISTIC MLE for the dependency of $n$. Note that this comparison is indeed under the same conditions.
> >
> > ---
> >
> > **Q7**: Clarification on Assumptions of Def 2.2
> >
> > **A7**: Thanks for the comment. We wish to clarify that given an instance, the proportion $N_{k,i,j}\in [0,1]$ is non-random and the label $\sigma_i$ is random. Given the features $\phi(k,i)$, whether or not an instance is consistent depends not only on $\phi(k,i)$ but also the proportions $\{N_{k,i,j}\}$, i.e., the proportion of times $i$ and $j$ are compared under state $k$. Hence, it is not true that "it may sometimes be consistent and other times not, depending on the data".
> >
> > We hope this reply could adequately address the concern on this question, and we remain open if the reviewer has a follow-up question at this point.
> >
> > ---
> >
> > **Q8**: Clarification on "Instance-Dependent" Bound.
> >
> > **A8**: The reviewer is right that some researchers may call these as "gap-dependent" bounds. When this gap is small, the hardness parameter $H$ is commensurately large, i.e., we may have $\exp(-n/H) \gg {1/\sqrt{n}}$ when $n$ is small. That is to say, the instance-dependent bounds are superior to the worst-case bound only when it $n$ is large, and it may be worse when $n$ is small. Hence, we claim that our research is complementary to that of Zhu et al [1].
> >
> > In light of this issue, we also provide a worst-case bound of our proposed algorithm in Proposition 3.4, which yields the form of $O(\frac{1}{\sqrt{n}})$ and it is not dependent on suboptimality gaps. We hope that our Proposition 3.4 can adequately answer the reviewer's question of "Could the authors explain how the proposed method would handle instances where the suboptimality gaps are small or non-existent?"
> >
> > [1] Zhu et al. (2023). Principled reinforcement learning with human feedback from pairwise or k-wise comparisons. ICML 2023.
> >
> > [2] Chowdhury et al. (2024). Differentially private reward estimation with preference feedback. In International Conference on Artificial Intelligence and Statistics (pp. 4843-4851). PMLR.
> >
> > ---
> >
> > Finally, we are grateful to the reviewer for their valuable feedback. Have we addressed your concern? Please let us know if there are any further questions.

---

> > > ### Comment · Reviewer_1qP9 · 2024-11-23
> > > **Thank you for the detailed explanations and rebuttals.**
> > >
> > > Thank you for the detailed explanations and rebuttals. Most of my questions have been addressed, but I have a few further thoughts:
> > >
> > > 1. Regarding Q2: I now understand that $H$ represents an instance-dependent term related to a type of gap. I recommend adding a concise yet impactful explanation of what $H$ represents when it is first introduced. For example, you could include an intuitive description or a small example to clarify its meaning. This would help readers quickly grasp the significance of $H$ and its role in highlighting the paper's key contributions.
> > >
> > > 2. Regarding Q4: While I see the alignment with the motivation in Chowdhury et al., I believe it’s crucial to frame this aspect in a way that feels distinct and necessary in your paper. Currently, the rationale for incorporating differential privacy (DP) feels like a reactionary decision (“others have considered it, so we should too”). Instead, you could establish the need for DP by explicitly illustrating potential risks or negative consequences of not using DP in RL methods. For instance, without DP, could there be challenges like vulnerability to adversarial attacks or loss of fairness? If so, presenting these tangible concerns would make the motivation for DP feel indispensable and elevate the discussion beyond a mere comparison to prior work.

---

> ### Author Response · Authors · 2024-11-24
>
> We would like to extend our sincere gratitude for the review's follow-up comments and helpful suggestions.
>
> ---
>
> **Q9**: I recommend adding a concise yet impactful explanation of what $H$ represents when it is first introduced. For example, you could include an intuitive description or a small example to clarify its meaning.
>
> **A9**:
> We are greatly thankful for the advice. We agree that it would be good to present an additional intuitive explanation for $H$. In light of this point, we have provided a newly added Appendix B in our revised version, and we also provide a link to this appendix in Section 3 of the main text. For the convenience of the reviewer's reading, we quote the text in our newly added Appendix B in the following:
>
>
> >The hardness parameter $H(v)$ is inversely proportional to the square of the suboptimality gaps across various states and actions. Specifically, a smaller suboptimality gap implies an increased hardness parameter, indicating that the instance is more challenging to learn. This relationship underscores the significance of the suboptimality gap as a critical measure in evaluating the complexity and learning difficulty of each instance.
>
> >For example, let's suppose $S=1, A=2$, $r_{1,1}=0$ and $r_{1,2}=1$. Given offline data with of size $n$ (i.e., these are $n$ history records for "action $1$ wins action $2$" or "action $2$ wins action $1$"), if the learner picks the action with the most winning records (as it does in our algorithm RL-LOW), then by Hoeffding's inequality, it will suffer an upper bound of expected simple regret of $\exp(-C \cdot \frac{n}{ {(r_{1,2}-r_{1,1})^{-2}}})$ for a constant $C$ that does not depend on $r_{1,2}-r_{1,1}$ (in fact this upper bound is also tight in the hardness parameter according to our lower bound). Notice the exponent is $-\Theta(\frac{n}{ {(r_{1,2}-r_{1,1})^{-2}}})$, and the hardness parameter $H(v)$ is exactly $ (r_{1,2}-r_{1,1})^{-2}$ under this instance.
>
>
> ---
>
> **Q10**: On the motivation of Label-DP
>
> **A10**: Thanks a lot for the advice. We agree that it is important to present more details for the motivation of why we consider Label-DP. We have provided a newly added Appendix A.2 to present more details on the motivation of label-DP to address this issue.  For the convenience of the reviewer, we quote our newly added Appendix A.2 in the following:
>
> >In the development of question-answering (QA) systems, a common approach for improving response quality involves engaging users in a labeling process where they are asked to provide preference labels. Specifically, users evaluate pairs of system-generated responses to a given question and indicate which response they prefer. This method, often referred to as pairwise preference labeling, is integral to training RLHF algorithms that aim to optimize the relevance and utility of answers provided by QA systems.
>
> >Given our understanding of the nature of this process, our research emphasizes the importance of protecting the confidentiality of user-submitted preference labels. Without any concerted attempt to protect privacy, these labels, which directly reflect individual opinions or biases toward specific types of responses, can potentially reveal sensitive information, e.g., their preferences for some specific political parties. Therefore, we augment our RL-LOW with a label-DP protection mechanism to mitigate the risk of privacy breaches from the labels.

---

### Official Review · Reviewer_cPuL · 2024-11-03

**Soundness:** 4
**Presentation:** 3
**Contribution:** 2
**Rating:** 5
**Confidence:** 4

**Summary:**

The paper studies offline reinforcement learning with preference feedback under the Bradley-Terry-Luce Model. There is a known feature function \phi that maps state-action pairs to a vector embedding such that the preference of two actions is determined by a linear function of their feature through a sigmoid transform.

The offline dataset have fixed s,a1, a2, but the label is random (from the BTL model). The target state distribution is \rho.  The offline dataset appears to be fixed for s, a1,a2 but have random Boolean label (preference).

The main claims are: (1) An algorithm that enjoys a gap-dependent bound that decays exponentially in the number of samples n. (2)  A lower bound showing that the rate of decay (on the exponent) is optimal up to a constant. (3) A label-differentially private algorithm with appropriate privacy and utility bound.

**Strengths:**

Strengths

-	The results appear to be new.
-	The presentation of the technical results is clear.
-	The logarithm of the upper and lower bounds of the suboptimality match up to a universal constant. The weak notion of optimality (on the exponent) may appear to be esoteric, but it is appropriate in cases when the error bound is exponentially small.
-	In a specialized setting, the result knocks out a log n factor in the worst-case expected “simple regret” of Zhu et al. (2023).

**Weaknesses:**

Weaknesses:

-	The model of offline RL being considered in the paper is crudely simplified. The target state distribution \rho is fixed rather than controlled by the chosen policy.  The policy only controls the actions.  This setting should be called “Offline Contextual Bandits under distribution-shift” rather than offline RL.  A generalization is given in Section 6 but still requires the transition dynamics to be known (still quite far from a typical offline RL setting).

-	The whole point of assuming a linear feature model in BTL model is that the learning bound does not depend explicitly on the cardinality of the state space and that the model can learn without seeing all states, i.e., the learner can generalize to unseen states.  The setting in this paper assumes linear features, but also requires state/action spaces to be finite and that the distribution samples each state with non-zero probability.  For example,  Zhu et al's bound depends polynomially in d.  When S = {0,1}^d,  this is log S. I personally think that it is more natural if the authors just focus on the tabular case (with \phi being the trivial one-hot feature).

-	Related to the above, the claim on computational efficiency is only valid when S and A are finite and small.  Typical settings in RL with function approximation require dependence on log S to be considered efficient.

-	The fourth strength above (Proposition 3.4 compared to Theorem 1.1 of Zhu et al 2023) is not a fair comparison.  In the setting of Zhu et al, there is no dependence on the cardinality of the state space.  The “coverage” is described by the properties of the feature covariance matrix.  The results of this paper require all states (in the support of target distribution \rho) to have many observations to be non-vacuous.

**Questions:**

*These are not quite questions, but comments that I think may help the authors.*

-	I would suggest that the authors not just focus on the dependence on n, but more thoroughly inspect the “coverage”, i.e., how the discrepancy between N and \rho affects the learning bounds.  This is well-studied in offline RL (non-preference feedback), e.g., https://proceedings.mlr.press/v97/chen19e.html  https://arxiv.org/abs/2106.04895  https://arxiv.org/abs/2203.05804


-	Since the setting is label-DP, e.g., in the “Our Contributions” setting, it should be appropriately referred to “label-DP”.  This is in contrast to existing studies on differentially private reinforcement learning that considered individual trajectories to be the contribution of a user (to be protected).  I agree that label-DP seems natural in the RLHF setting.

-	More discussion on the following body of work will help the paper to situation its novel contribution in a broader context:

1.	Gap-dependent analysis for Dueling Bandits, e.g.,  https://proceedings.mlr.press/v130/yang21a/yang21a.pdf

2.	Contextual Dueling Bandits (and many follow-up work, e.g., https://proceedings.neurips.cc/paper/2021/file/fc3cf452d3da8402bebb765225ce8c0e-Paper.pdf who talked about large-gap as an open problem) https://proceedings.mlr.press/v40/Dudik15.html

3.	Gap-dependent bound for offline RL is well-studied but not cited (a).	Tabular setting:  https://arxiv.org/abs/2206.00177  (b).	Linear function approximation https://arxiv.org/abs/2211.13208

Note that in those cases even if the rewards are known, a 1/n rate is required in the lower bound. The main difference is that the current paper assumes away any planning and does not measure the regret in terms of the value function.

---

> ### Author Response · Authors · 2024-11-20
>
> **Q1**: " The target state distribution $\rho$ is fixed rather than controlled by the chosen policy. A generalization is given in Section 6 but still requires the transition dynamics to be known."
>
> **A1**: The reviewer is right. Our RL setting indeed requires the transition dynamics to be known in both Sections 2 and 6. Notably, these assumptions align with those made by Zhu et al. [1] in their Sections 1 and 5. More importantly, we wish to highlight that whether or not the transition dynamics are known, prior to our work, there were no instance-dependent bounds in the offline PbRL setting. Our paper thus provides a first step to derive the instance-dependent bounds in the offline PbRL setting for bridging this research gap in the literature.
>
> ---
>
> **Q2**: The setting in this paper assumes linear features, but also requires state/action spaces to be finite and that the distribution samples each state with non-zero probability. For example, Zhu et al's bound depends polynomially in $d$. When $S = \\{0,1\\}^d$, this is $\log S$.  Typical settings in RL with function approximation require dependence on $log S$ to be considered efficient."
>
> **A2**: Yes, the state/action spaces are assumed to be finite
> in our current manuscript. However, we wish to highlight that Zhu et al.'s bound not only depends polynomially in $d$, but also depends on a term $\mathbb{E}_{s \sim \rho} (\cdot )$ that is related to state distribution. Hence, their upper bound implicitly depends on the number of states; see Theorem 3.2 in Zhu et al [1].
>
> In addition, it is worth noting that the proposed algorithm of Zhu et al.  also requires the computational complexity depending on $S$ rather than $\log(S)$, as they need to calculate a term "$\mathbb{E}_{s \sim q} (\cdot )$" in their algorithm.
>
> Most importantly, we wish to clarify that **the computational complexity's dependency $S$ is inevitable**, as we require to output $\hat{i}_k$ for all  $k \in \mathcal{S}$ in the current setting. Nonetheless, if we remove this output's requirement, the dependency "$S$" is indeed avoidable in our algorithm, and the overall computational complexity becomes $O(nd^2+d^3)$, aligning with the reviewer. We have added more details in our modified Appendix E.2 of our revised version to address this issue. Thank you very much for the precious comment. We hope this reply could adequately address this comment.
>
> ---
> (edit: update the appendix letter for the latest revised paper)

---

> > ### Author Response · Authors · 2024-11-20
> >
> > **Q3**: "Proposition 3.4 compared to Theorem 1.1 of Zhu et al. 2023 is not a fair comparison.".
> >
> > **A3**: We agree with the reviewer. Apart for $n$, the dependencies of other parameters are not comparable between our worst-case upper bound and those in Zhu et al. [1]. Thank you very much for raising this opinion. We have made this clear in the Section 3 of the revised version, and we highlighted that our superiority to Zhu et al. is *only for the dependency of $n$*. Given our main contribution is to provide the first instance-dependent bounds, we hope the reviewer could view this point of worst-case bound with optimism.
> >
> > ---
> >
> > **Q4**: Since the setting is label-DP, e.g., in the “Our Contributions” setting, it should be appropriately referred to “label-DP”.
> >
> > **A4**: Thank you very much for the valuable suggestion. We have appropriately highlighted "label" in "our contribution" as well as in Section 5 of our revised paper. Thank you again for the advice.
> >
> > ---
> >
> > **Q5**: On the other related works.
> >
> > **A5**: We are very grateful for the reviewers to provide the details of the other related works. We have cited all the papers that are mentioned by the reviewers in our newly added Appendix A of the revised manuscript.
> >
> > ---
> >
> > **Q6**: "Note that in those cases even if the rewards are known, a $1/n$ rate is required in the lower bound. The main difference is that the current paper assumes away any planning and does not measure the regret in terms of the value function."
> >
> > **A6**: Thanks for the comment. We agree that $1/n$ rate may be required in the lower bound under other settings.
> > In our setting, we wish to highlight that it is straightforward to verify that this $1/n$ (or $\frac{1}{\sqrt{n}}$) rate cannot be the form of the instance-dependent lower bounds.
> >
> > For example, let's say $S=1, K=2$, $r_{1,1}=0$ and $r_{1,2}=1$. Given the offline data with size $n$ (i.e., these are $n$ history records for "action $1$ wins action $2$" or "action $2$ wins action $1$"  ), if the learner picks the action with the most winning records (as it does in our RL-LOW), then by Hoeffding's inequality it will suffer an upper bound of expected regret of $\exp(-\Omega(n))$ instead of $O(\sqrt{\frac{1}{n}})$ (or $O({\frac{1}{n}})$) in this instance. Hence, $O(\sqrt{\frac{1}{n}})$ (or $O({\frac{1}{n}})$) can not be the form of an instance-dependent lower bound of our setting, as it will violate the upper bound of the above instance.
> >
> > Thanks again for this comment, which reinforces the novelty of our derived lower bound.
> >
> > [1] Zhu et al. (2023). Principled reinforcement learning with human feedback from pairwise or k-wise comparisons. ICML 2023.
> >
> > ---
> >
> > Finally, we really appreciate the reviewer's careful reading and feedback (particularly for provding the details of other related stuies), and we welcome any further questions if the reviewer may have.

---

> > > ### Comment · Reviewer_cPuL · 2024-11-23
> > >
> > > Thanks for your response to these questions and comments.
> > >
> > > Re: A6.  Thanks for the example. I think I understand it.  Though I was pointing out that in the settings where people study "full-fledged" RL that requires not only estimating the reward, but also solving exploration,  even if you assume for every state s, there is a superior action that is substantially better than other actions,  you still cannot get logarithmic sample complexity of learning.  The challenge is not from learning reward (as I said, the reward function is assumed to be known and fixed),  but rather from learning the dynamics sufficiently so you know how to reach that high-reward state.
> > >
> > > It is a very interesting question when one can solve RL with logarithmic sample complexity (and what does it mean by "large gap" in the RL setting with pairwise comparison).    e.g., maybe one should compare not just action but states and require one state to be easily reachable, even if the transition kernel is estimated very crudely.

---

> > ### Comment · Reviewer_cPuL · 2024-11-23
> >
> > Thanks! The answer to Q1 is well-taken.
> >
> > I don't follow the logic in the answer A2.  Why would E_{s \sim \rho}[ . ] necessarily depend on the number of states?
> > Let's say s ~ Normal(0,1).   The cardinality of s is unbounded.   But the expected value of most function of s should be well-behaved.
> >
> > Similarly, in terms of computation, one can estimate an expected value to alpha accuracy with 1/alpha^2 samples?    So it would be poly(n) rather than linear in |State space|?

---

> ### Author Response · Authors · 2024-11-23
>
> We are deeply grateful for the prompt follow-up feedback.
>
> ---
>
> **Q7**: "Why would $E_{s \sim \rho}[ . ]$ necessarily depend on the number of states?"
>
> **A7**: The reviewer is right in their special example. Hence, we did not claim $E_{s \sim \rho}[ . ]$ of Theorem 3.2 in Zhu et al. explicitly depend on $S$ but in an implicit way, which is similar to our worst-case bound of Prop 3.4.
>
> The reviewer is correct in noting that we can reduce computation by calculating $\mathbb{E}_{s \sim q}()$ approximately rather than exactly in Zhu et al.'s algorithm; however, it is unclear (for us) how the performance would be affected by doing so for Zhu et al.'s algorithm. Hence, we claim the computation of their algorithm depends on $S$ at least for their original version in **A2**.
>
> Nonetheless, we would like to highlight that our RL-LOW's computational complexity indeed does not depend on
> $S$ if we remove an output's requirement, as detailed in our revised Appendix E.2 as well as **A2**. We hope this could satisfactorily align with the reviewer's expectation for the computational complexity.
>
> Thank you very much for the question, which gives us the chance for further clarification.
>
> ---
>
> **Q8:** "Though I was pointing out that ....You still cannot get logarithmic sample complexity of learning.".
>
> **A8**: We wish to highlight that we indeed get a logarithmic sample complexity of learning in our setting.
>
> Specifically, in our settings (in both Sections 2 and 6), the upper bounds of the expected regret are of the form $\exp(-\Omega(n/H(v)))$ in terms of the dependency on $n$. That is to say, to obtain the regret of $\varepsilon$, one needs to take no more than $n =O(\log(1/\varepsilon))$ samples when taking $H(v)$ as a contant, which is indeed a *logarithmic function* of in $1/\varepsilon$.
>
> More importantly, we wish to emphasize that our upper bound matches our lower bound in Theorem 4.3, elucidating the important role of the hardness parameter $H(v)$. That is, our algorithm is order-optimal and cannot be improved in general in terms of instance-dependent analysis, which is beyond whether it can yield a logarithmic sample complexity.
>
> Admittedly, our result is not applicable to other settings (e.g., those settings that the reviewer considers), and we wish to emphasize the significance of our setting in the following **"A9"**.
>
> ---
>
> **Q9**: On the significance of the setting.
>
>  **A9**: We agree that it is interesting to consider other various settings, e.g., when the rewards are known and the transitions are unknown.
>
> However, given that our particular setting (i.e., the rewards are unknown, but the transitions are known), which was proposed by Zhu et al. has hitherto not been studied well, we feel it is meaningful to provide a complementary instance-dependent analysis to Zhu et al., elucidating the dependence of the regret on the instance. This setting is also pertinent in LLMs, because the states are represented by prompts and the actions are represented by the next response (or next token). For example, let's say the current state $s=$"the capital of America is". If we take action $a$="Washington", then next state $s'$ is deterministic and known to be $s'$="the capital of America is Washington".  That is, the rewards are unknown, but the transitions are indeed known in this application.
>
> We hope that the esteemed reviewer appreciates our contributions to this practically-relevant setting in which we have established the first instance-dependent theoretical results. We kindly wish that the reviewer could adopt optimistic views to the various settings. Thank you so much again for the very detailed follow-up comment.
>
> ---
>
> Update: We simplify "A7" for the ease and efficiency of communication in the rebuttal stage.

---

> > ### Comment · Reviewer_cPuL · 2024-11-25
> >
> > I think we are on the same page.
> >
> > Re: A7
> > > Hence, we did not claim of Theorem 3.2 in Zhu et al. explicitly depend on but in an implicit way, which is similar to our worst-case bound of Prop 3.4.
> >
> > I do not see any (implicitly or explicitly) dependence in the cardinality of the state space (even logarithmic dependence) in Theorem 3.2 of Zhu et al.   Their result appears to work even if most (nearly all) states are not visited in the data.
> >
> > > however, it is unclear (for us) how the performance would be affected by doing so for Zhu et al.'s algorithm.
> >
> > Based on their arguments, it doesn't seem hard (a basic perturbation analysis). I am surprised to learn that they used exact expectation in the algorithm (rather than the empirical average, as in their use of the sample complexity).
> >
> > Re: A8.
> > > we indeed get a logarithmic sample complexity of learning in our setting.
> >
> > I never say you didn't. I was sayng that the "full-fledged" RL with preference feedback setting is exponentially harder than the setting (Bandits with Preference Feedback) you guys considered.

---

> ### Author Response · Authors · 2024-11-26
> **Thanks a lot again for the engagement!!!**
>
> Thanks a lot again for the engagement. We apologize for the use of "implicitly" that is ambiguous in A2.
>
>  ---
>
> **Q10**: Their result appears to work even if most (nearly all) states are not visited in the data.
>
> **A10**: Thanks for pointing it out in Zhu et al.'s results. We would like to highlight that **our results also work if most states are not visited in the data**.
>
> Note that in our setting, we do not assume any relation between the offline data's proportions $N_{k,i,j}$ and the distribution of states $\rho_k$.
> For example, let $S=10000$, $d=2$,$A=3$, $N_{1,1,2}=0.5$, and $N_{1,2,3}=0.5$ (i.e., $N_{1,1,3}=N_{k,i,j}=0$ for $k>1$ and $i \neq j$). In addition, let $\rho_k = \frac{1}{10000}$ for all $k \in \mathcal{S}$. In state $1$, let $\phi(1,1)=[0,1]^\top$,$\phi(1,2)=[1,0]^\top$ and $\phi(1,3)=[1,1]^\top$. Furthermore, for any state $k$, suppose $i^*_k$ is unique and $\phi(k,i)\neq \phi(k,j)$ for $i \neq j$.
>
>
> That is, in the above instance, we only have data in state $1$ and no data for other states, and the $\rho$ is uniform in $\mathcal{S}$. Note that the above instance is a consistent instance from our Def 2.2. Hence, the upper bounds of Theorem 3.3 and Prop 3.4 are valid under this instance.
>
> ---
>
> **Q11**: "the "full-fledged" RL with preference feedback setting is exponentially harder than the setting" (of Zhu et al. and ours)
>
> **A11**: We agree. In view of the fact that the present setting is not studied well in the literature, we hope our results can provide some insights for future study of much harder settings in the community.

---

### Meta-Review · Area_Chair_FTXz · 2024-12-20

**Metareview:**

The reviewers are unfortunately not excited about the results in the paper, both before and after the rebuttal discussion. While some of the issues raised are superficial (for instance, I do not see any problem in this paper's notation in the first place), the paper is below the bar from a technical merit standpoint. The main issue is that the paper is only studying the much simpler problem of contextual bandits, rather than RL, as the title suggests (as also pointed out by the reviewer). Bandits are much simpler, and there are many more instance-dependent bounds investigated in the bandits literature than in RL. There's a lack of comparison to bandits literature (studied not just in ML/CS, but in operations research and statistics). As such, I view this paper as a warm-up to the ultimate worthwhile pursuit for RL. I would suggest work out the full RL version and submit to ICML.

**Additional Comments On Reviewer Discussion:**

NA

---

### Decision · Program_Chairs · 2025-01-22

Reject